# Global methane footprints growth and drivers 1990-2023

Yuli Shan [1,2,9] ✉, Kailan Tian [3,4,9] ✉, Ruoqi Li [1,5], Yuru Guan [1], Jiamin Ou [6], Dabo Guan [7,8] & Klaus Hubacek [5,9] ✉

Methane has been identified as the second-largest contributor to climate change, accounting for approximately 30% of global warming. Countries have established targets and are implementing various measures to curb methane emissions. However, our understanding of the trends in methane emissions and their drivers remains limited, particularly from a consumption perspective (i.e. accounting for all emissions along the entire global supply chain). This study investigates the most recent dynamics of methane emissions across 120 sectors from both production and consumption viewpoints in 164 countries. It also discusses the status of decoupling of production- and consumption-based methane emissions from economic growth. Our results indicate that there is no foreseeable slowdown in the momentum of global methane emissions growth. Only a few developed countries have managed to reduce both production- and consumption-based emissions while maintaining economic growth (i.e., strong decoupling) during the observed period (1990-2023). Global trade accounts for approximately 30% of global methane emissions, but major trade patterns are shifting from North-North and North-South to South-South countries, reflecting the increasing participating of developing countries in global supply chains. The study further reveals the changing drivers of global methane emissions from 1998 to 2023 in five-year intervals. It identifies that the reduction in emission coefficient (i.e., emissions per unit of output), driven by advancements in improved energy efficiency and cleaner production technologies, is the main determinant for reducing emissions over the observation period, partly offsetting the increasing effects from growth of final demand. Changes in demand structure have played a considerable role in the increase of emissions since 2008. This study enhances our understanding of the changes and drivers of methane emissions and supports countries in incorporating methane emissions into their climate mitigation strategies.

Climate change is one of the most challenging threats we are facing in this century. Scientists have confirmed that anthropogenic Greenhouse Gas (GHG) emissions are the main drivers of increasing global temperature and accelerated climate change since the industrial revolution[1]. In contrast to $CO_2$, which has been the main focus of carbon mitigation policies, methane has a much shorter lifetime in the atmosphere[2], and has contributed about 30% to global warming since pre-industrial times due to its high Global Warming Potential (GWP)[3]. The GWP of methane is 80 times higher than that of $CO_2$ over the first 20 years; thus, methane contributes to increasing temperature in the

short term. This implies that effective mitigation of methane could offer a quick path to slowing down global temperature rise in the foreseeable future, which is not readily achievable by reducing $CO_2$ emissions alone. In addition to the climate forcing effect, methane is an indispensable precursor for hazardous air pollutants such as tropospheric ozone, which is responsible for about one million premature deaths per year, globally[4]. As a result, awareness of the important role of methane emissions in climate change mitigation has been steadily increasing, especially after the 26th United Nations Climate Change Conference (COP26) in 2021. During the COP28 in 2023, the United States and China released the "Sunnylands Statement on Enhancing Cooperation to Address the Climate Crisis," in which both countries agreed to incorporate methane reduction commitments into their NDCs by 2035. Additionally, the EU proposed monitoring and reducing the methane emission intensity of imported energy starting in 2030, with the aim of decreasing methane emissions and mitigating leaks. Over 130 countries have committed to reducing methane emissions by at least 30% by 2030, but more actions are needed[5].

Global methane emissions have been rising for decades, with the average (since 2001) annual growth rate reaching 1.22%, slightly below the growth rate of $CO_2$ emissions (1.80%)[6]. Existing emission datasets and studies have presented long-term methane trends[6-10]. For example, He et al.[11] employed a top-down approach to match methane emissions transported through a global chemical transport model with observation data. Saunois et al.[7] combined multiple existing datasets to compare methane estimated derived from both bottom-up and top-down approaches. Some studies further looked into methane emission patterns of selected countries (e.g., China[12], US[13], UK[14]) and several major sectors, including agricultural activities[15], such as livestock[16], the food system[17]; wastewater treatment[18-20] and waste management in landfills[21]; and energy sectors such as oil and gas production[22], and coal mining[23]. Sun et al.[24] and Ari and Şentürk[25] also discussed the extent of decoupling between methane emissions and economic growth.

However, these studies focused on production-based methane emissions, ignoring consumption-based emissions. Production-based emissions (PBE) refer to emissions emitted within a specific country or region during the production of goods and services, encompassing all economic activities and sectors of a country (to answer the question of where emissions arise). Meanwhile, the consumption-based emissions (CBE) allocate the emissions to the countries where the goods are consumed (to answer why we have these emissions or the ultimate drivers of emissions). It offers a different insight into understanding countries' emissions along the entire global supply chain. It is essential to consider both PBE and CBE when designing policies to reduce emissions to prevent emission leakage and the outsourcing of emission-intensive production to other locations[26], which would result in a misleading decoupling of PBE from economic growth.

Previous studies have employed the input-output approach to estimate methane CBE, emissions embodied in global supply chains, as well as their drivers. However, research lacks comprehensive coverage of a wide range of countries and high sectoral resolution. For example, Zhang et al.[27] accounted for methane CBE using the Eora Multi-Region Input-Output (MRIO) database[28] and Emissions Database for Global Atmospheric Research (EDGAR). While their analysis covered a wide range of sectors and economies (26 sectors across 181 economies), it was limited to a narrow temporal scope, spanning 2000 to 2012. Liu et al.[29] analyzed the trade embodied emissions of 56 sectors in 44 countries from 2000 to 2014 using the World Input–Output Dataset[30]. Fernández-Amador et al.[31] analyzed the consumption-based methane emissions for 78 countries/regions in 1997, 2001, 2004, 2007, 2011 and 2014 using the Global Trade Analysis Project MRIO tables. Sun et al.[32] investigated the drivers of methane CBE in 43 countries from 2000 to 2014. Ma et al.[33] analyzed Chinese methane CBE and drivers for four years (2005, 2007, 2010 and 2012) and 20 sectors. Similar studies include Zhang et al.[34] and Wang et al.[35]. In addition to the analysis of the

whole economic system and supply chains, some studies have specifically focused on the Agriculture, Forestry and Other Land Use activities[36,37] and food production[38], which constitute the major sources of methane emissions. Although sector-specific analyses offer detailed insights into emission patterns, they fall short of presenting the overall landscape of changes in emissions across the entire economic system influenced by human activities and interactions with other sectors.

Aiming to address the research gaps concerning the lack of up-to-date analysis on methane emissions from a consumption-based perspective, covering a wide range of countries and with high sectoral resolution, this study uses the latest GLORIA input-output dataset (Global Resource Input-Output Assessment)[39] to examine methane CBE of 164 countries/regions (accounting for 98% of global methane emissions) for the long-term period 1990–2023[6]. We further compare the degree of decoupling of both PBE and CBE from economic growth in each country, reveal the recent changes in trade embodied emissions, and investigate the drivers of changes in CBE with structure decomposition analysis (SDA). We compare our methane emissions results with $CO_2$ emissions and conclude the study with a discussion of solutions to reduce methane emissions. For regional comparisons, we follow the Intergovernmental Panel on Climate Change (IPCC) Sixth Assessment Report (AR6)[40] classification and group countries into five categories: Developed Countries, Eastern Europe and West-Central Asia, Asia and Developing Pacific, Latin America and the Caribbean, and Africa and the Middle East (see Supplementary Fig. 1 for details). The latter four groups are considered developing countries. More details about the input–output model, decoupling analysis, SDA, and input data are described in the Methods section.

## Results

### Methane emission trends

Based on our calculations, global methane emissions have increased slightly from 266.4 million tons/yr (equivalent to 7.5 gigatons of $CO_2$ emissions/yr, Gt $CO_2$-eq/yr) in 1990 to 292.3 million tons/yr (8.2 Gt $CO_2$-eq/yr) in 2002[6], with an annual growth rate of 0.8%/yr. Since 2002, global emissions increased at a faster rate of 2.0%/yr, reaching 329.5 million tons/yr (9.2 Gt $CO_2$-eq/yr) in 2008. Almost one third (or 32.5%) of the global increase (i.e., 37.2 million tons) over the period can be attributed to China, whose emissions increased by 12.1 million tons due to increasing production for both domestic use and exports. Other Asian and Developing Pacific countries (as classified by the Intergovernmental Panel on Climate Change (IPCC) Sixth Assessment Report (AR6)[40], see details in Supplementary Fig. 1) contributed another 27.1% of the global increase while Africa and Middle East contributed 23.1%. After a decline in 2009 due to the global financial crisis, global methane emission growth resumed at 1.1%/yr, reaching 383.5 million tons/yr (10.7 Gt $CO_2$-eq/yr) in 2023. Compared to previous studies focusing on emission trends before 2014[31,41], our result further highlights a resurgence in methane emissions after 2015, particularly in Asia and rapidly developing countries, driven by ongoing economic and population growth. That is to say, global methane emissions have been growing faster in recent years[42], albeit with some fluctuations, which is in contrast with the recent slowdown observed in $CO_2$ emission growth[43]. We derive the robustness of our results by comparing them with estimates from other methane inventories and MRIO tables. GLORIA methane emissions are 0.9% lower than EDGAR estimates in 2022 and 9.7% higher than IEA Methane Tracker in 2023. For CBE, comparisons with estimates based on an alternative GTAP MRIO and the above PBE sources yield Pearson $R$ values of 0.952–0.998, confirming the reliability of our results (see the "Methods" and Supplementary Information for details).

Figure 1a shows the emission trends by country group in terms of production (solid lines) and consumption (dashed lines). Since the mid-1990s, Asia and the developing Pacific have been dominating

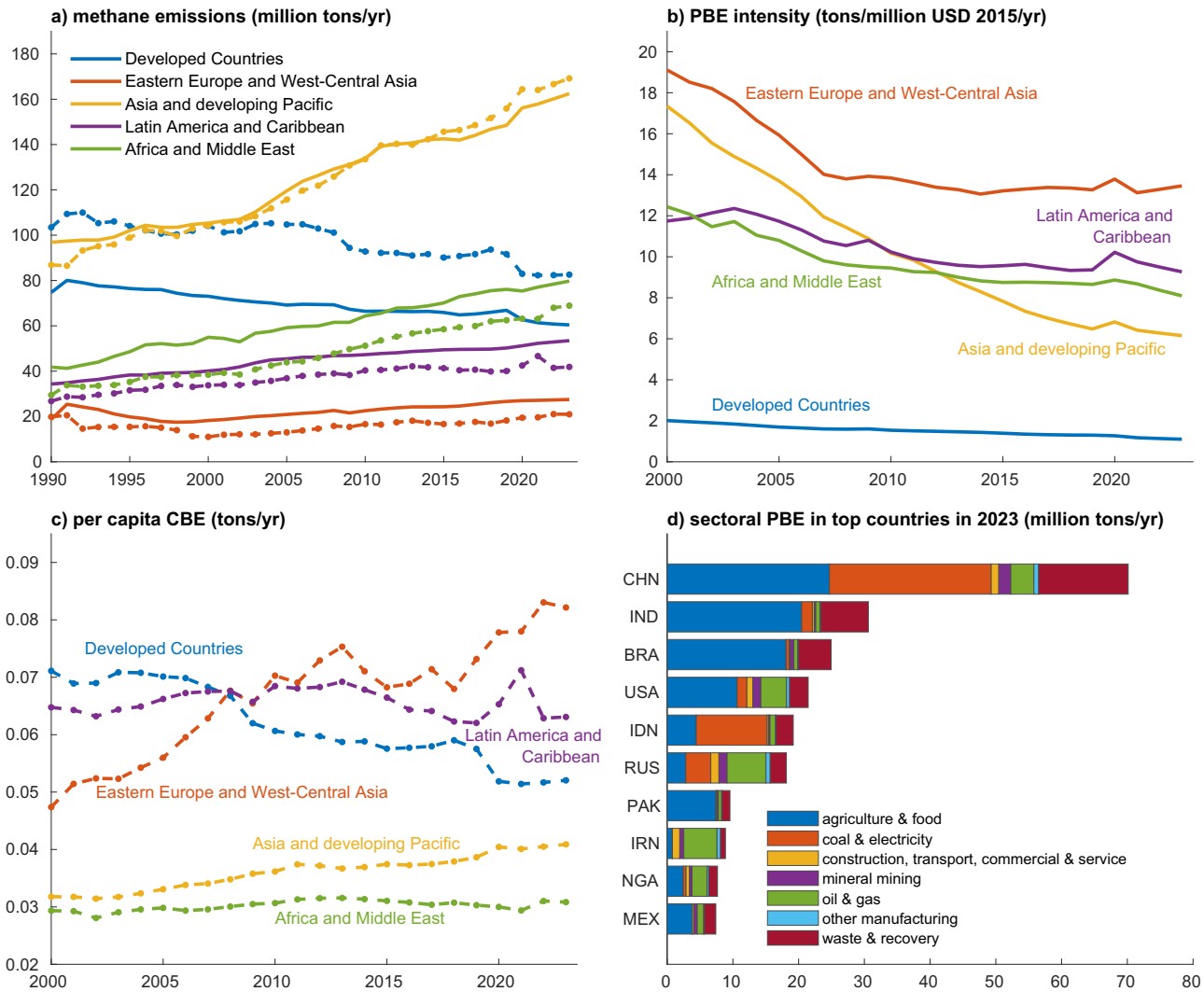

**Fig. 1 | Methane emissions by country income group and the top ten countries.**
**a** PBE (in solid lines) and CBE (in dashed lines) of country groups during 1990–2023;
**b**, **c** PBE intensity (i.e., PBE per unit of GDP) and per capita CBE of country groups,
respectively; and **d** Sectoral PBE in the top ten countries in 2023. Data source: the
PBE and CBE of countries used in **a** are calculated based on GLORIA environmental
accounts and MRIO tables[39]; PBE intensity and per capita CBE in **b** are calculated
with the World Bank dataset; the sectoral emissions in **d** are collected from EDGAR[6].
Detailed PBE and CBE of countries and aggregated groups are provided in Sup-
plementary Data 1. A global map for the five aggregated regions is shown in the
Supplementary Fig. 1.

global methane PBE, increasing from 96.9 million tons/yr (36.4% of
global emissions) in 1990 to 162.4 (42.4%) in 2023. Meanwhile, the PBE
of Developed Countries, globally the 2nd largest contributor,
decreased from 74.7 million tons/yr (28.0%) in 1990 to 60.4 (15.8%) in
2023. The PBE from other country groups kept slowly increasing over
the past decades and reached 79.7 million tons/yr (20.8% of global
emissions, Africa and Middle East), 53.5 (13.9%, Latin America and
Caribbean) and 27.5 (7.2%, Eastern Europe and West-Central Asia) in
2023. The top ten emitting countries shown in Fig. 1d) contributed
56.8% of global PBE in 2023, reflecting roughly their share on global
population (56.0%) and GDP (54.2%). In terms of CBE, Asia and the
developing Pacific surpassed the Developed Countries after 1999 and
became the world's top CBE emitter. The CBE of Asia and developing
Pacific increased from 32.6% (86.9 million tons/yr) in 1990 to 44.1%
(169.2) in 2023. Developed Countries have declined CBE from 38.8%
(103.5 million tons/yr) to 21.5% (82.6) over the same period.

When comparing a country's PBE and CBE, the gap between them
is known as net emissions embodied in trade. A larger PBE than CBE
implies that the country is a net exporter of emissions embodied in
goods and services, while a net emission importer (i.e., CBE higher than
PBE) causes more emissions embodied in goods and services than it

produces. Developed Countries are the only global net importers over
the whole observed period. Their net emissions embodied in trade
varied between 20.2 and 35.6 million tons/yr over the period, which
accounted for 24.4–34.0% of their CBE. Asia and the Developing Pacific
have seen a change in the role from a net exporter to an importer
between 2010 and 2014. After 2014, their imported embodied emis-
sions increased from 0.2 (0.1%) to 6.8 million tons/yr (4.0% of CBE). All
the other country groups are net emission exporters, with 12.1–65.7%
of their national PBE caused by export production. Our findings align
with prior research[31,32,34], underscoring the increasing complexity of
methane emissions driven by global supply chains. More details will be
discussed in the section below.

The trends of methane emissions are considerably different from
those of $CO_2$ emissions, for which the Developed Countries were the
major contributors until 2011 for PBE and 2015 for CBE[44]. The reason
could be that $CO_2$ is mainly emitted from fossil fuel consumption, in
particular the manufacturing and power sectors, while methane
emissions are mainly from agriculture and energy extraction. Devel-
oped countries are usually highly industrialized with more consump-
tion and production of industrial products. In contrast, developing
countries are at early industrialization stages with a higher share of

energy extraction, mining and agriculture in their economic structure of the economy, and thus emit a huge amount of methane emissions. Therefore, increasing attention to emission reduction of methane should be paid to the developing countries, especially those in Asia and developing Pacific.

Figure 1b, c presents the average PBE intensity (i.e., PBE per unit of Gross Domestic Product (GDP) in USD 2015 constant prices) and per capita CBE of country groups. All country groups show a decline in PBE intensity over the past two decades. Developed Countries have the lowest PBE intensity (1.1 tons/million US dollar (USD) 2015/yr) in 2023, while Eastern Europe and West-Central Asia have the highest value (13.5). Asian and developing Pacific countries had the most considerable drop in their PBE intensity from 17.3 tons/million USD2015/yr in 2000 to 6.2 in 2023. In contrast, when looking at the per capita CBE, only Developed Countries show a continuous decline in their per capita CBE, from 0.07 tons in 2000 to 0.05 tons in 2023. Asia, the developing Pacific, Africa, and the Middle East have relatively low levels of per capita CBE compared to the other three groups due to lower levels of consumption per capita.

To illustrate the sectoral distribution of methane PBE, we aggregated 120 sectors into seven categories, while the underlying calculations are done at the 120-sector resolution. These seven sectors are shown in Fig. 1d for the ten nations with the highest PBE. Globally, agriculture and food production are the largest contributors to methane emissions, accounting for 47% in 2023, followed by waste and recovery (17%), coal and electricity (14%), and oil and gas (13%). Over time, the proportions of sectoral emissions have shifted. Agriculture and food production decreased from 54% in 1990 to 47% in 2023, and oil and gas declined from 18% in 2006 to 13% in 2023. In contrast, emissions from coal and electricity production rose from 9% in 2000 to 14% in 2023, reflecting sectoral structure changes in global methane emissions. Countries have huge differences in the sectoral structure of methane emissions, as shown in Fig. 1d. For example, coal and electricity production, especially the extraction of hard coal, is the largest emission source in China and Indonesia, contributing to 35% and 56% of their PBE, respectively. The two countries account for 59.3% of global coal production in 2023 and therefore emit a huge amount of methane from coal mining[45]. In contrast, Russia and Iran have higher emissions from oil and gas production (33% and 57%, respectively). India and Brazil emit large shares of emissions from agriculture (67% and 73%, respectively), e.g., pigs and cattle, due to livestock farming.

## Decoupling of methane emissions and economic growth

The trends of methane emissions show that a number of countries have reduced emissions in recent years. Between 2013 and 2023, 55 (33.5% of 164) countries reduced their PBE, 57 (34.8%) countries reduced their CBE, and 30 (18.3%) reduced both. 17 of the 30 countries that reduced both PBE and CBE are high-income countries. Among the five country groups (shown in solid dots in Fig. 2a, b), Developed Countries is the only group that have achieved both PBE and CBE decline and GDP growth. We also notice a decline in emissions not only in Developed Countries, but also in some developing countries. Thirteen developing countries have reduced both PBE and CBE. This implies that emission reduction can be achieved at any level of economic development. However, such emission reductions might be caused by economic recessions, as economic growth has been recognized as a highly relevant indicator and an important driver of GHG emissions[46–48].

We use the Tapio Decoupling Index to describe the dynamic relationship between emissions (both PBE and CBE) and economic growth (i.e., GDP) in each country and aggregated group during the years 2013 to 2023. Strong decoupling occurs when a country's emissions decrease while its GDP continues to grow. Weak decoupling happens when both emissions and GDP increase, but GDP grows at a

faster rate than emissions. In contrast, no decoupling indicates that emissions are rising at a faster rate than GDP.

Figure 2c, d shows the results for the period 2013–2023. We find that among the 55 countries that have reduced their PBE, 48 countries have achieved strong decoupling of PBE and GDP, indicating that there are 7 countries reducing PBE due to economic recessions (i.e., belonging to the group of recessive-no-decoupling or recessive-weak-decoupling). Forty-seven countries have achieved strong decoupling of CBE and GDP, among which 23 countries also achieved PBE decoupling.

We find that Developed Countries are the only group that achieved strong decoupling in both CBE and PBE from economic growth. Asia and the developing Pacific show weak decoupling of emissions and GDP over the period, implying a slower emission growth than economic growth. For example, China's PBE and CBE increased by 11.2% and 27.6% (2013–2023), respectively, meanwhile its GDP increased by 74.6%. The Africa and Middle East group has a similar trend to Asia and the developing Pacific. For example, Uganda increased its PBE and CBE by 37.1% and 44.8% respectively, while its GDP increased by 57.3% from 2013 to 2023. Latin America and the Caribbean have achieved strong decoupling of CBE and GDP, but their PBE is only weakly decoupled from GDP. For example, Colombia reduced its CBE by 8.4% while PBE increased by 12.1% from 2013 to 2023. Eastern Europe and West-Central Asia show no evidence of decoupling of PBE nor CBE versus GDP, which means this country group has faster emission increase than economic growth. For example, Russia had increased PBE by 18.7% and CBE by 14.7% while GDP increased by only 4.3%. However, the group as a whole does not necessarily reflect the individual decoupling status of each country within it. Some developing countries have already achieved strong decoupling in terms of both PBE and CBE. For example, Armenia has decreased both PBE (by −10.4%) and CBE (−19.2%) from 2013 to 2023, while GDP kept growing.

It is important to point out that the status of decoupling is temporary and can change over time. Thirty countries have achieved strong decoupling of both PBE and CBE from 2003 to 2013, but only 12 of them stayed strongly decoupled from 2013 to 2023. Thus, even if a country has decoupled its emissions from economic growth over a period, they should be wary of future rebounds. A full list of our decoupling results of all countries from 1993 to 2023 at 10 year intervals can be found in Supplementary Data 2.

There is no significant difference between strongly decoupled countries (mostly developed) and non-decoupled countries (mostly developing) in terms of per capita CBE ($p$-value = 0.999 for a two-sample $t$-test), as shown in Table 1. This suggests that people in different countries have a similar level of methane footprints to meet their consumption needs across the entire supply chain. This finding contrasts with $CO_2$ emissions, where the average per capita CBE of strongly decoupled countries (10.27 tons/yr in 2018) is considerably higher than that of non-decoupled countries (4.47 tons/yr)[44]. In contrast, the PBE intensities of strongly decoupled countries are significantly lower than those of weakly or non-decoupled countries ($p$-value = 0.000), mirroring the pattern seen in $CO_2$ emissions[44]. Among the top 10 emission-intensive sectors (e.g., agriculture, energy, mining, and wastewater treatment), the average sectoral emission intensity in weakly/non-decoupled countries is 35–460% higher across nine sectors than in strongly decoupled countries.

## Emissions embodied in trade and outsourcing

Although a number of developed countries have decoupled emissions from economic growth, this might be achieved by outsourcing emission-intensive industries to less developed regions[26]. Previous studies have noted this trend in methane emissions, despite domestic reductions being achieved[31,34]. Such outsourcing will largely increase

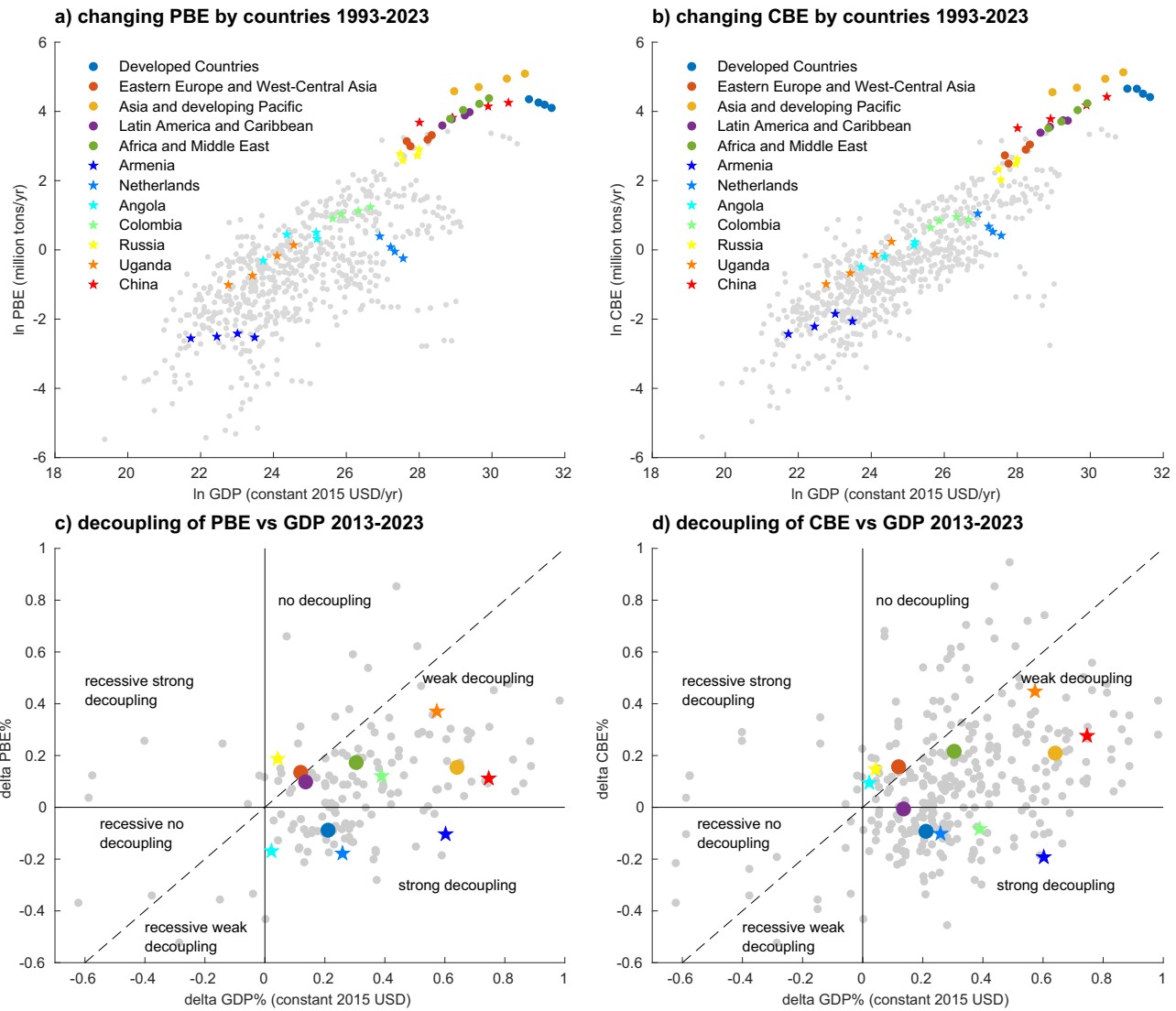

**Fig. 2 | Decoupling of methane emissions and economic growth in selected countries at 10-year intervals. a**, **b** The relationship between PBE, CBE and GDP during the time period 1993–2023 at 10-year intervals, the colored dots present five country groups, and the colored stars present selected seven countries; the gray dots present the remaining countries. **c**, **d** The decoupling results of countries/groups in coordinate systems. Delta GDP and emissions refer to the changes in GDP and emissions over the given period. Detailed results of decoupling are shown in Supplementary Data 2.

the emissions from less developed regions with lagging levels of technology and lower production efficiency, which will lead to a possible rise in global emissions[44]. This is especially the case for countries that have achieved strong decoupling of PBE and GDP, where no such decoupling is observed between CBE and GDP. For example, the PBE of New Zealand decreased by 2.8% from 2013 to 2023 while its CBE increased by 49.3%. Sun et al.[32] also noted emissions transfer from developed countries to emerging countries in agriculture and food manufacturing sectors align with global trade. Therefore, we investigate the emissions embodied in international trade flows, as shown in Fig. 3a, b.

Methane emissions embodied in trade increased from 84.1 to 118.9 million tons/yr from 1998 (Fig. 3a) to 2023 (Fig. 3b) globally, but as a proportion of global total emissions remained unchanged (29.4% in 1998 and 31.0% in 2023). In 2023, Developed Countries were the overwhelming largest emission importer, accounting for 42.6% (50.6 million tons/yr) of total emissions embodied in trade, followed by Asia and Developing Pacific countries (35.0%, 41.6). In terms of export, Asia and Developing Pacific are the world top manufacturing hubs with the largest contribution of 29.3% (34.8 million tons/yr) of global export

related emissions. In comparison, Developed Countries accounted for 23.9% or 28.4 million tons/yr and Africa and Middle-East countries for 21.9% or 26.1 million tons/yr export emissions.

The patterns of emissions embodied in trade have changed considerably from 1998 to 2023. With the rapid growth of developing countries, trade within the Global South (encompassing four groups of developing countries) has considerably increased, now playing a dominant role in the world trade patterns. More and more developing countries are now considerably participating in global supply chains and importing larger shares of goods, rather than solely serving as global manufacturing exporters. Emissions embodied in trade from Global North (i.e., Developed Countries) to Global South, also known as North–South trade, have declined from 49.4% (41.6 million tons/yr) of global trade embodied emissions in 1998 to 39.6% (47.1) in 2023. Meanwhile, emissions embodied in South-South trade have increased from 21.2 million tons/yr (25.2%) to 55.8 (47.0%) over the same period. The increase in trade-related emissions associated with the global South is mainly allocated to Asia and the developing Pacific. Their imported embodied emissions increased from 13.6 million tons/yr (16.1% of the global total trade embodied emissions) in 1998 to

**Table 1 | Average CBE per capita, PBE intensity, and GDP per capita of different decoupling groups in 2023**

| Country group | No. | Average CBE per capita (tons/yr) | Average PBE intensity (tons/million USD2015/yr) | Average GDP per capita (thousand USD2015) |
|---|---|---|---|---|
| Strong decoupling (CBE) | 47 | 0.0678 | n/a | 28.5 |
| Strong decoupling (PBE) | 48 | n/a | 5.5 | 26.2 |
| Strong decoupling (CBE & PBE) | 23 | 0.0655 | 2.5 | 35.7 |
| No/week decoupling (CBE) | 117 | 0.0678 | n/a | 10.3 |
| No/week decoupling (PBE) | 116 | n/a | 21.5 | 11.1 |
| No/week decoupling (CBE & PBE) | 141 | 0.0681 | 19.0 | 12.3 |

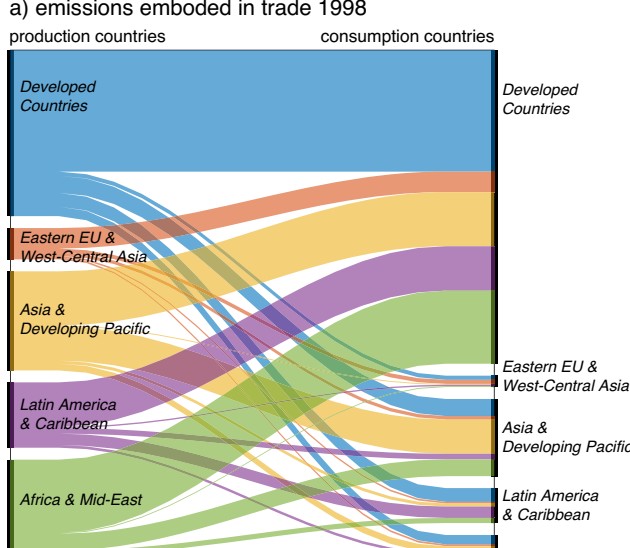

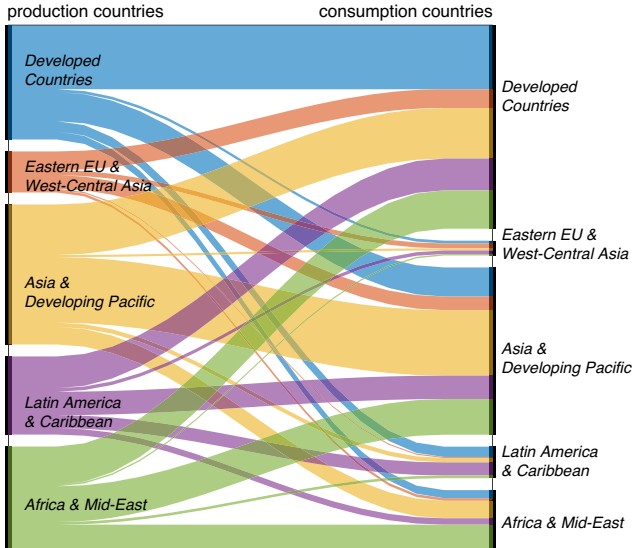

**Fig. 3 | Emissions embodied in trade 1998 and 2023. a, b** Emission flows in 1998 and 2023. The figures include the emission flows from one country to another (i.e., emissions associated with production in one country which are then consumed in another country), and exclude the emissions where production and consumption happen in the same country. The country groups on the left *y*-axis are producers, while country groups on the right *y*-axis are consumers. The flows between the same country group present emissions embodied in trade between different countries within the group.

41.6 million tons/yr (35.0%) in 2023. Similar trends have been found for $CO_2$ emissions embodied in trade as well[49].

When examining emission flows by sector, we observe considerable differences across countries. For example, in 2023, Developed Countries have the largest share of emissions embodied in their agricultural and food-related exports (57.6%), while the major sectors for Africa and the Middle East's emissions embodied in exports are oil and gas (27.2%). Agricultural and food products account for 20.5% of Africa and the Middle East's exported emissions. Latin America and the Caribbean export substantial amounts of agricultural and food products, accounting for 73.4% of their exported embodied methane emissions. In contrast, Asia and the Developing Pacific region have 47.4% of their emissions embodied in exports associated with coal and electricity production. Figure 4a, b shows detailed trade embodied emissions by sectors in top-emitting countries in 1998 and 2023.

We observe that emissions embodied in North-to-South trade primarily originate from agriculture (63.6% in 2023). In contrast, South-to-North trade is predominantly focused on resource extraction (i.e., coal, oil, gas and mineral mining), accounting for 45.4% of emissions embodied in exports. A possible explanation of these patterns is that the global North outsources mining industries to the global South, with a flow of resources back through global supply chains[50,51]. More importantly, these trade patterns have intensified over the past decades. Detailed sectoral emissions embodied in the trade of countries are provided in Supplementary Data 3.

## Drivers of consumption-based methane emissions

We employed a structural decomposition analysis (SDA) to investigate the contributions of five drivers behind the changes in global emissions (as shown in Fig. 5a) and each country group (as shown in Fig. 5b–f) at 5-year intervals from 1998 to 2023. We also presented the contributions of drivers for the five top-emitting countries (i.e., China, India, USA, Brazil, and Russia), as described in the Supplementary Table 1.

The global results indicate that economic growth has been identified as a key driver of increased emissions, which has been discussed previously[31]. In addition, we find that changes in the structure of final demand play an important role in the increase of emissions, particularly after 2008. Between 2008–2013 and 2018–2023, this factor contributed to a staggering 155 and 234 million tons (or 47% and 64%) of emission growth, respectively. This suggests that there is an increasing share of methane-intensive products in the final demand. For example, the share of red meats (beef, lamb, and pork), which are methane-intensive products[17,52], in total global demand has increased from 0.6% to 0.8% between 2008 and 2023.

Changes in emission coefficients (i.e., emissions per unit of output) are the main determinant to reduce emissions over the period, offsetting the increasing effects from demand growth and structure changes, which is consistent with previous studies[31,32,53]. This driver reflects considerable improvements in the emission efficiency, driven by advancements in improved energy efficiency and cleaner production technologies. Our results show that global average emission

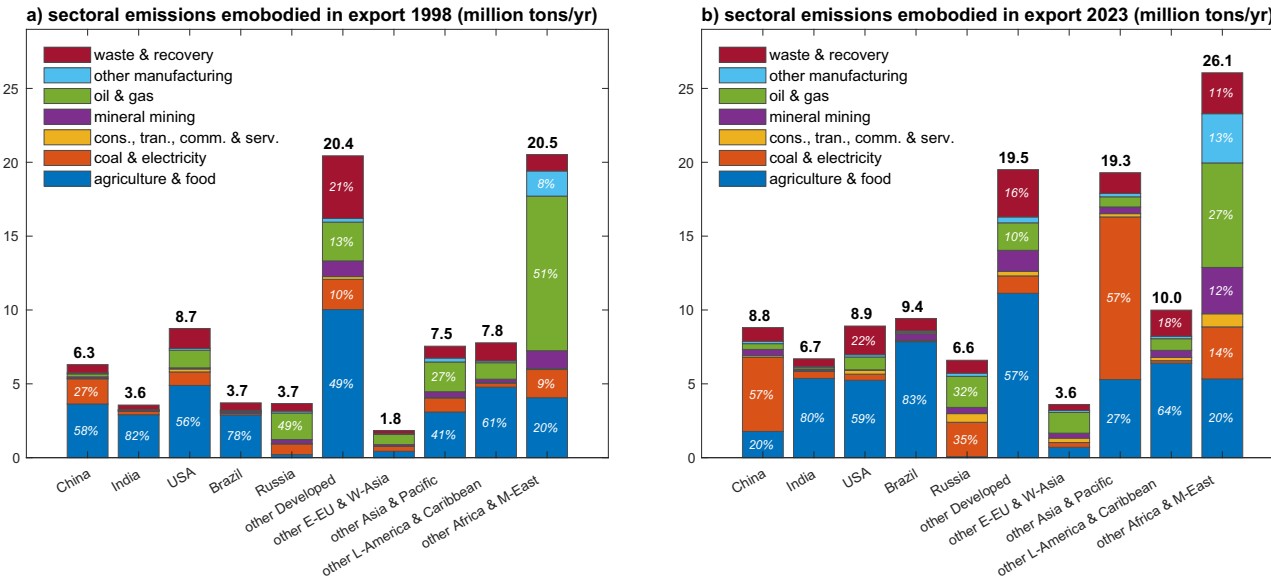

**Fig. 4 | Sectoral emissions embodied in the export of countries 1998 and 2023. a, b** The sectoral emissions by countries in 1998 and 2023, respectively. Sector distributions in the figure are aggregated based on GLORIA sectors according to their emissions.

coefficients decreased by 66.7% between 1998 and 2023. Such decreases in emission coefficients have been observed across sectors. For example, between 2018 and 2023, 97 out of 120 sectors reduced their emission coefficient. In particular, three sectors with the highest emission coefficient—raising of animals and agricultural services; water collection, treatment, and supply, sewerage; and raising of cattle—reduced their emission coefficient by 42%, 32%, and 20%, respectively, during this period. These reductions can be attributed to various advances in production practices, such as new irrigation control systems and improved fertilization management in the agricultural sector[54]; enhanced livestock, manure management, and breeding efficiency in animal husbandry[54,55]; the recycling and reuse of waste, along with emerging anaerobic digestion technologies in sewage treatment[56,57].

Changes in production technology reflect transformations in global production structures, which encompass both domestic economic structure and inter-country input-output industrial linkages. Between 2003 and 2013, changes in production technology increased global methane emissions by 208 million tons (31%), indicating an increased reliance on intermediate inputs from methane-intensive sectors. However, during 2018–2023, this trend reversed, with production technology contributing to a reduction in emissions. This shift suggests that global production is gradually transitioning to a cleaner (in terms of methane emission) intermediate input structure. Nevertheless, progress remains slow, with emissions decreasing by only 56 million tons (15%). Population growth has a stable but limited increasing effect on emissions, contributing to 13 (4%) to 19 million tons (6%) every 5 years.

In addition to the global results, we observe high heterogeneity across country groups, as illustrated in Fig. 5b–f. Although the influencing directions (positive or negative) of drivers across regions and periods are the same, the size of the contribution differs. We calculated the standard deviation (SD) of every driver using their share of contributions across countries. The results show that the population has the least degree of dispersion compared to the other four drivers, with a standard deviation (SD) of 0.06. The SD of contributing effects of emission coefficient to total emission changes in each country is 1.02, while for production technology the SD is 0.40, final demand structure is 0.87, and the SD for the size of final demand is 0.43.

We further compared the heterogeneity of each driver between country groups by calculating the SD of the average contribution of the drivers in each group. We find that the emission coefficient, final demand per capita, and demand structure show the greatest differences between groups; the SDs of these three drivers are 0.71, 0.56, and 0.41, respectively, compared to 0.21 for production technology and 0.04 for population. The wide range of SDs can be attributed to the heterogeneity among country groups. Taking the emission coefficient as an example, it increased emissions by 75% in Eastern Europe and West-Central Asia from 1998 to 2003, while reducing emissions in the other four regions (−57% to −38%). Similarly, the driver of demand structure increased emissions by 242% in Eastern Europe and West-Central Asia from 2018 to 2023, which is much higher than the other four country groups (44–95%). This indicates that the dynamics of emissions and drivers are fluctuating across regions and periods, similar to the status of decoupling from GDP, as discussed above. Achieving emission reduction in one region/period does not promise long-lasting or stable reduction in the future.

## Discussion

This study employs an environmentally extended global MRIO approach, decoupling analysis, and SDA, to analyze methane emission changes globally and for 164 countries' from both production- and consumption-based perspectives from 1990 to 2023.

We find that Developed Countries stand out as the sole group that has consistently achieved emission reductions and strong decoupling between emissions (both PBE and CBE) and GDP. Their decoupling is mainly caused by a decline in their emission coefficient, and to a lesser degree, due to outsourcing methane emissions to less developed regions. One hundred and twenty sectors, 100 of which are from Developed Countries, reduced their emission coefficient between 2003 and 2013. This is evident from the analysis of the drivers, where the emission coefficient is the primary determinant in reducing emissions. Therefore, many sectors in Developed Countries provide successful examples in methane emission control for other countries around the world.

Asia and developing countries, particularly due to the rapid growth of China, have been major contributors to the increase in methane emissions over the past decades, establishing themselves as

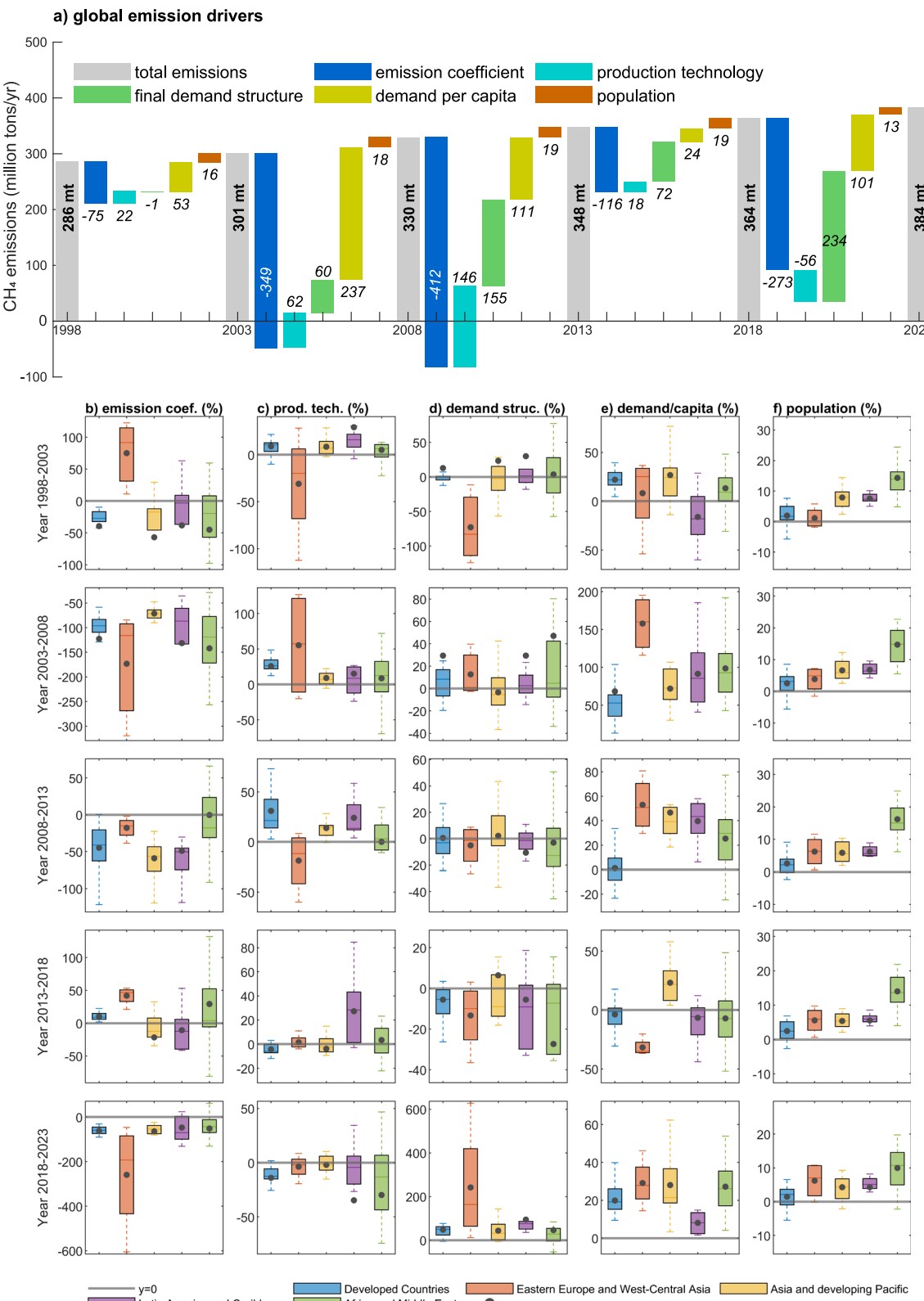

**Fig. 5 | Emission drivers of consumption-based methane emissions at 5-year intervals. a** The contributing effects of each driver to global emissions in absolute values, **b–f** Contributing effects of each driver to five country groups in percentages relative to the group's total emissions.

leading global methane emitters. Consequently, we recommend directing our focus on addressing methane emissions in Asia and the developing Pacific countries. While the joint announcement by the United States and China at COP28 is a positive initiative, further efforts are necessary to facilitate developed countries' support for developing

countries in reducing emissions through financial and technical assistance. It is essential to recognize that, although emerging and small countries currently emit lower levels of methane, they may become key methane emitters in the future due to economic growth. Therefore, they should not be overlooked when designing global goals for

GHG reduction, and policies need to be designed to prevent emissions from increasing rapidly at this stage.

We find that global trade accounts for approximately 30% of methane emissions, which underscores the key role of global trade in influencing emission patterns, providing valuable insights for shaping effective reduction policies. As sectors across countries differ in technological endowment, production efficiency, and emission coefficient of products produced in different countries vary. This provides an opportunity to reduce a sector's methane footprint by supply chain management of upstream production through carefully selecting import partners with lower emission coefficients. Taking the production and trade of fertilizers as an example, the production of which is one of the top methane sources, Russia is the world's leading exporter of fertilizers. It exported 18.7 billion USD worth in 2022, accounting for 13.1% of global total exports, followed by Canada (14.5 billion USD), China (12.7), the US (8.19), and Morocco (7.92)[58]. Our results show that the emission coefficient for Russia's fertilizer production is 0.07 tons/ million USD/yr in 2022. However, the emission coefficients of fertilizer production in Morocco (with 0.002 tons/million USD/yr), Canada (0.05), the US (0.04), and China (0.06) are lower than in Russia, indicating these countries have more advanced production technologies with higher efficiency or alternative inputs for fertilizer production, reducing associated methane emissions. Countries that currently import fertilizer from Russia can reduce their imported embodied emissions by shifting to other exporting countries with lower emission coefficients. However, it is important to consider that this shift could lead to other environmental problems (e.g., in the case of Morocco, which has a larger share of phosphate fertilizers with lower methane emissions but higher local environmental impacts). It is encouraging to see the EU making efforts to reduce the import of high-emitting products from non-European countries by setting up the Carbon Border Adjustment Mechanism (CBAM), which will invite EU countries to choose cleaner upstream trade partners to reduce their emissions to stay competitive. CBAM includes fertilizers whose production is emission-intensive and high risk of carbon leakage. The EU accounts for 23% of global fertilizer imports. Future actions of CBAM should focus on expanding to a larger range of products, including methane-intensive ones[59].

Our analysis of driving forces identifies that a reduction in emission coefficients, via advancements in improved energy efficiency and cleaner production technologies, is the main determinant for reducing emissions and achieving strong decoupling over the observation period. Reducing methane emission coefficients requires targeted strategies across sectors. For example, the energy sector can apply advanced detection technologies, such as infrared imaging and drones, in oil and gas extraction, storage, and transportation to identify and repair emission leaks[59,60]; replace coal and oil with clean energy across the entire energy system[61]; improve ventilation systems in oil and coal mines to reduce uncontrolled methane emissions[62]; and deploy equipment to capture associated methane emissions for recycling rather than flaring. In the agricultural sector, we can improve feed formulations to reduce methane production in ruminants by incorporating additives like oils, enzymes, or methane inhibitors into livestock diets[54,55]; increase biogas recovery by processing manure and organic waste into energy[57]; and improve rice farming through irrigation control (e.g., alternate wetting and drying methods) and better fertilization management[63]. In the waste management sector, we can enhance landfill management by sealing landfills and installing methane recovery systems to recycle methane for power generation or heating; promote composting technologies by replacing landfilling with composting to effectively reduce methane emissions from organic waste; and upgrade wastewater treatment by adopting advanced anaerobic digestion technologies to capture methane from wastewater and use it as an energy source[56].

The measures above can reduce emission coefficients from a production perspective. Unlike $CO_2$ emissions, which mainly stem from fossil fuels, a major source of methane emissions is biological activities, such as biodegradation in wastewater treatment, fertilizers production, and effusion from coal mining, and livestock farming. We further call for smart demand-side emission reduction strategies to supplement the aforementioned reduction measures in mitigating methane emissions, particularly from biological activities. For example, we can promote a shift towards consuming less red meat in our diets[52,64]. Since methane has a relatively short lifespan of 7–12 years in the atmosphere, our demand-side efforts today will have an immediate impact on global warming.

## Methods
### Global multi-regional input–output (MRIO) analysis
This study adopts the widely used environmental extended Global MRIO method to calculate the methane CBE and also emissions embodied in trade, as shown in Eq. (1).

$$CBE_{i,j} = diag(int_{i,j}) \times (I - A)^{-1} \times FD_k \quad (1)$$

In Eq. (1), $CBE_{i,j}$ is the CBE in sector $i$ of country $j$, which are generated by the final demand of country $k$; $int_{i,j}$ is the emission coefficient in sector $i$ of country $j$, which is PBE per unit of output in each country sector; $A$ is the direct input coefficient matrix, which shows the sectoral inputs per unit of output; $(I - A)^{-1}$ is known as Leontief inverse matrix, which reflects the technical level of production in a given economic system; $FD_k$ i is the final demand of country $k$.

In this study, we use the GLORIA dataset[39] which is a global MRIO dataset that integrates global economic flows with environmental data to assess the impacts of consumption and production patterns across industries and regions, relying on a range of input data including sectoral economic accounts, international trade statistics, and environmental resource use data (e.g., energy, water, materials). GLORIA includes 120 sectors, 164 countries/regions, and four final demand categories (i.e., household final consumption, non-profit institutions serving households, government final consumption, gross fixed capital formation).

### Decoupling analysis
Tapio decoupling index[65] is utilized to gauge the extent of decoupling between methane emissions and economic growth in individual countries, as shown in (2). Such relationships between emissions and economic growth detected by the decoupling index are descriptive and no causality can be extracted.

This index has previously been extensively employed for $CO_2$ emissions[44,47,66]. Sun et al.[24] applied this method to assess the decoupling of methane PBE and GDP across 83 countries. However, there have been no existing studies addressing the decoupling status of methane CBE from GDP. This analysis can help identify the risk of emissions outsourcing when compared to decoupling results based on PBE.

$$DI = 1 - \frac{Em_1 - Em_0}{Em_0} / \frac{GDP_1 - GDP_0}{GDP_0} \quad (2)$$

In Eq. (2), $DI$ represents the decoupling index for a specific period and country; $Em$ denotes methane emissions, with consideration given to both PBE and CBE; constant-price $GDP$ (in 2015 USD) is used to reflect the economic level of a country; the subscript 0 and 1 indicate the base year and reporting year, respectively.

A degree of strong decoupling ($DI \geq 1$ & $GDP_1 \geq GDP_0$) implies that the country's emissions decrease but the GDP keeps growing, weak

decoupling ($0 \leq DI < 1 \, \& \, GDP_1 \geq GDP_0$) has both emissions and GDP increasing, but GDP is growing faster than emissions, while no decoupling ($DI < 0 \, \& \, GDP_1 \geq GDP_0$) means the country's emissions are growing faster than its GDP. Similarly, countries with a recessionary economy ($GDP_1 < GDP_0$) can be categories into three groups as well.

## Structural decomposition analysis (SDA)

We employed the SDA to quantify the driving effects of certain determinants of changes in methane emissions. Compared to Index Decomposition Analysis, SDA has higher data requirements but can better explain the contribution from structure change from both production and consumption perspectives. SDA has been used in research on $CO_2$ emissions[67–69], energy consumptions[70,71], and material footprints[72]. However, few studies have applied these methods to methane CBE. Sun et al.[32] investigated the drivers of methane emissions in 43 countries during 2001–2014, while Cheng et al.[53] explored the drivers of income-based methane emissions during the same period but using Ghosh MRIO and SDA models. There is still limited knowledge regarding the drivers of methane CBE across a wide range of countries in the most recent decade.

Based on the equation, we calculate the CBE, and the changes in emissions can be decomposed as (3). Instead of using first-order decompositions, which has $5! = 120$, we use the average of two polar decompositions, which can adequately approximate the average of all decompositions[73], as shown in Eq. (4).

$$
\begin{aligned}
\triangle CBE = &\triangle int \times L \times FS \times FDper \times P + int \times \triangle L \times FS \times FDper \times P \\
&+ int \times L \times \triangle FS \times FDper \times P + int \times L \times FS \times \triangle FDper \\
&\times P + int \times L \times FS \times FDper \times \triangle P
\end{aligned} \quad (3)
$$

$$
\begin{aligned}
\triangle CBE_{int} = &\tfrac{1}{2}(\triangle int \times L_0 \times FS_0 \times FDper_0 \times P_0 + \triangle int \times L_1 \times FS_1 \\
&\times FDper_1 \times P_1) \\
\triangle CBE_L = &\tfrac{1}{2}(int_1 \times \triangle L \times FS_0 \times FDper_0 \times P_0 + int_0 \times \triangle L \times FS_1 \\
&\times FDper_1 \times P_1) \\
\triangle CBE_{FS} = &\tfrac{1}{2}(int_1 \times L_1 \times \triangle FS \times FDper_0 \times P_0 + int_0 \times L_0 \times \triangle FS \\
&\times FDper_1 \times P_1) \\
\triangle CBE_{FDper} = &\tfrac{1}{2}(int_1 \times L_1 \times FS_1 \times \triangle FDper \times P_0 + int_0 \times L_0 \times FS_0 \\
&\times \triangle FDper \times P_1) \\
\triangle CBE_P = &\tfrac{1}{2}(int_1 \times L_1 \times FS_1 \times FDper_1 \times \triangle P + int_0 \times L_0 \times FS_0 \\
&\times FDper_0 \times \triangle P)
\end{aligned} \quad (4)
$$

In Eq. (3), int and $L$ represent sectoral methane emission coefficient and the Leontief inverse matrix, respectively, as described in Eq. (1); FS represents the sectoral structure of final demand, calculated as sectoral final demand divided by the total volume, and shows the consumption structure; FDper represents the per capita final demand, reflecting the economic level of a country; $P$ represents the population. The subscript 0 and 1 indicate the base year and reporting year, respectively; variables with a $\triangle$ before them denote changes from the base year to the reporting year.

The idea of an SDA is to consider the change in emissions due to the change in each driver, assuming the other drivers remain unchanged. Therefore, the five items in Eq. (4) give the respective contribution of each driver to the change in CBE. $\triangle CBE_{int}$ gives the impact of the emission coefficient, often interpreted as the contribution of emission technology. As emission technology improves, emissions per unit of output decrease directly. $\triangle CBE_L$ provides the contribution of the changes in the global Leontief inverse, often interpreted as changes in production technology. It reflects

transformations in global production structures which encompass both domestic economic structure and inter-country input–output industrial linkages[74]. $\triangle CBE_{FS}$ captures the impact of the final demand structure. For instance, a shift in final demand towards emission-intensive products will lead to an increase in overall emissions. $\triangle CBE_{FDper}$ and $\triangle CBE_P$ provide the contribution of final demand per capita and population, respectively. Final demand per capita often reflects a country's economic level, as wealthier nations tend to consume more.

## Data source

We collect methane PBE from the environmental accounts of GLORIA[39], which provides emissions data by countries and products consistent with the GLORIA MRIO table. The emission data includes both biogenic methane, produced and released from living plants and animals, and emissions from fossil fuel use. GLORIA environmental accounts combine EDGAR, the Organization for Economic Cooperation and Development, and Eurostat emission datasets.

The global MRIO tables used in this study are sourced from the GLORIA dataset. In comparison to other MRIO datasets, GLORIA stands out for providing the longest time-series MRIO table, covering 164 countries and 120 products from 1990 to 2027 in version 057. Tables from 2022 to 2027 are forecasted based on GDP projections from the International Monetary Fund. We specifically utilize the table spanning from 1990 to 2023 to capture long-term changes and the most up-to-date dynamics in global methane emissions.

The GDP and population data by countries are collected from the World Bank dataset.

## Uncertainties and limitations

We acknowledge that there are uncertainties in the accounting of methane emissions, from both production- and consumption-based approaches. The uncertainties of PBE come from emission factors and activity data (e.g., energy consumption and industrial outputs)[75]. Top-down approaches based on satellite and ground-based observations can be used to enhance the accuracy of PBE accounts. For example, Sadavarte et al.[76] found that the methane emissions from coal mines estimated using satellite observations are considerably higher than those estimated by bottom-up approaches. Meanwhile, the uncertainties of CBE arise from the PBE coefficients and also from input–output data[40,77].

We have performed uncertainty and robust analysis of countries' PBE and CBE as shown in the Supplementary Information and Supplementary Data 3. In brief, we compare our PBE (i.e., GLORIA environmental accounts) with emission data from EDGAR v8.0_GHG 1970-2022 ($CO_2$, $CH_4$, $N_2O$, F-gases)[6] and IEA Methane Tracker[78]. Comparisons reveal that GLORIA methane emissions (379.5 million tons in 2022 and 383.5 in 2023) are 0.9% lower than EDGAR estimates in 2022 (388.3 million tons) and 9.7% higher than IEA Methane Tracker in 2023 (349.5 million tons). Regarding CBE, we compare our estimates with countries' CBE calculated using an alternative GTAP MRIO dataset[79] and abovementioned PBE sources. The Pearson $R$ value ranges from 0.952 to 0.998 for different comparisons, indicating the robustness of our calculations.

## Data availability

All results data supporting the findings of this study are available in the Supplementary Data 1–4. The source data underlying figures can be extracted from the Supplementary Data files. Country group classifications are based on the regional definitions outlined in the Intergovernmental Panel on Climate Change (IPCC) Sixth Assessment Report (AR6)[40]. All the data generated in this study are available in our open-access dataset "Carbon Emission Accounts and Datasets for emerging economies (CEADs)" (https://www.ceads.net/data/) for free download.

## Code availability

The MATLAB code for the GMRIO and SDA modeling can be provided upon request.

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

## Acknowledgements

We thank the Carbon Neutrality and Energy System Transformation programme (D.G.) and the anonymous reviewers. This research is funded by the National Natural Science Foundation of China (72373081 (D.G.), 72242105 (D.G.), 72311530105 (Y.S.), 72373141 (K.T.), 71988101 (K.T.), 72361137002 (K.H. & Y.S.). We acknowledge supports from the Royal Society International Exchanges (IEC\NSFC\223059) (Y.S.), the Research Grants Council of the Hong Kong Special Administrative Region, China (AoE/P-601/23-N) (D.G.), the Horizon Europe Projects PANTHEON (101137905) (K.H.) and EU-CHINA-BRIDGE (101137971) (Y.S.), which is supported by UKRI grant (10132630) at the University of Birmingham (Y.S.). D.G. acknowledges the support by the New Cornerstone Science Foundation through the Xplorer Prize and the AXA Chair Grant.

## Author contributions

Y.S. and K.H. designed the research. Y.S. led the study, calculated the results and wrote the paper. K.T. provided technical support for the input–output analysis. R.L. and Y.G. revised the paper with inputs from K.H., J.O., and D.G. All authors reviewed the paper.

## Competing interests

The authors declare no competing interests.

## Additional information

[1]School of Geography, Earth and Environmental Sciences, University of Birmingham, Birmingham B15 2TT, UK. [2]Birmingham Institute for Sustainability and Climate Action (BISCA), University of Birmingham, Birmingham B15 2TT, UK. [3]State Key Laboratory of Mathematical Sciences, Academy of Mathematics and Systems Science, Chinese Academy of Sciences, Beijing 100190, China. [4]Department of Land Economy, University of Cambridge, Cambridge CB2 1RX, UK. [5]Integrated Research on Energy, Environment and Society (IREES), Energy and Sustainability Research Institute Groningen, University of Groningen, Groningen 9747 AG, the Netherlands. [6]Department of Sociology, Utrecht University, Utrecht 3584 CC, the Netherlands. [7]Department of Earth System Science, Ministry of Education Key Laboratory for Earth System Modelling, Institute for Global Change Studies, Tsinghua University, Beijing 100084, China. [8]The Bartlett School of Sustainable Construction, University College London, London WC1E 6BT, UK. [9]These authors jointly supervised this work: Yuli Shan, Kailan Tian, Klaus Hubacek. ✉e-mail: y.shan@bham.ac.uk; k.tian@amss.ac.cn; k.hubacek@rug.nl

