## [Transparent Peer Review File · Nature Communications]

Global methane footprints growth and drivers 1990-2023

Corresponding Author: Professor Yuli Shan

Version 0:

Reviewer comments:

Reviewer #1

(Remarks to the Author)

the manuscript titled 'Global methane footprints growth and drivers 1990-2023' presents an assessment of global methane emissions based on production and consumption for 164 countries. The study uses a mixture of satellite and inventory (EDGAR and World Bank) data for the analysis and finds that there is a relationship between consumption and production to methane emissions and decoupling methane from economic growth is tricky. The paper offers an interesting insight into methane emissions and a different approach to emissions accounting but the paper requires major revisions before it can be considered for publication in Nature Communications. I outline my concerns below:

- the paper is too long and does not fit the requirements specified for articles published in Nature Communications
- the GLORIA data used in the study, what resolution and frequency are the data? Satellite measurements of methane are traditionally used on a regional or site specific basis as the resolution of satellites, bar GHGSat and Sentinel-5P (when paired with data from Sentinel-2), is too large for useful studies and the data is often patchy.
- the emission coefficient is stated as an important factor in reducing emissions but this term is never explained
- the paper talks in great detail about agriculture and waste methane emissions but does not mention oil and gas methane emissions. As many initiatives to tackle methane are related to the oil and gas sector, and globally the oil and gas sector is estimated to account for around 20% of anthropogenic emissions, this seems like an oversight
- the International Energy Agency (IEA) have a methane tracker that is often used for methane data. Have the results of the study been compared to this (for 2022 to 2023)?
- there are some minor grammatical issues in the paper which can be addressed from detailed proof reading (but overall writing is of high quality but some minor issues with tenses)
- the authors explain the general reasons for the changes in PBE and CBE, but more detailed explanations could be given to better explain why some countries and regions experience such large growths in emissions e.g. is China's big increase in methane entirely due to coal or did the improvement in diet (more meat and access to food) also have an impact?
- it is not clear from that paper what actions should be taken to reduce methane emissions and how countries can effectively decouple emissions from growth (especially in a permanent or long term way).

Reviewer #2

(Remarks to the Author)

General comments

The consumption/production analysis outlined within this paper is an interesting and pertinent approach to investigating the trends and drivers in anthropogenic methane emissions. Considering that earlier studies of this type are now over a decade old this will be a topic of interest to many and useful to the wider scientific community. However, I have a number of concerns which must be addressed before considering publication.

The comparison of consumption/production emissions is a very interesting approach and useful to better understand the global scale flows and drivers of emissions. But its benefits need to be better emphasised and articulated within the paper. What does this approach tell you that the standard (typically production) doesn't? This should be your focus.

There is a distinct lack of context. While earlier studies are mentioned (paragraph 4 of the introduction) at no point are the results of this study compared to these studies. Is there the same general pattern of emissions distribution between sectors

(e.g. compared to the GCP CH₄ budget)? Are the trends over time similar to earlier studies? The discussion does not need to be extensive but the results need context.

There are no uncertainty estimates given on any of the data. How can a reader be sure that these are actual trends/differences and not just noise?

While the main paper only requires minimal methodological information there are a number of key details and definitions which should be included to ensure that a general scientific audience can understand the paper, specifically:

A clear definition of a production based emission vs. a consumption based emission

A brief (one sentence) description of how the GLORIA model works and what input data it relies on (UNFCCC? EDGAR? Country specific economic data?)

A map showing which countries are in which region colour coded to match Fig. 1

A definition of PBE intensity

One of the key drivers of decreasing emissions is specifically identified as “improvement in emission coefficient[s]” in both the Abstract and Drivers of consumption-based methane emissions sections. But there’s no discussion as to whether these emissions coefficients are “improving” because of changes in production methods which are reducing methane emissions (i.e. they changes reflect an actual improvement/reduction in methane emissions for a given sector/process) or whether the emissions coefficients are being updated to reflect emissions from the given sector more accurately (i.e. processes within the given sector have not changed just our estimate of the emissions). If it’s the later than this isn’t a true reduction in emissions merely a methodological artefact. If it’s the former than more detail as to exactly which processes and which sectors would be of interest to the reader. This is touched on briefly in the final paragraph of the Conclusions but needs to be explored in further detail in the main body of the paper and the abstract.

The language used is at times overly wordy, confusing and ambiguous. E.g. Brazil, India and Russia are described as “the top three emitting countries of fertilizers”. Does this mean that these countries have the highest methane emissions due to fertilizer production (or consumption?) or is it that they produce the most fertilizer? Similarly, methane is described as “the world’s second largest greenhouse gas emission source”. Is this perhaps meant to be the second most important contributor to radiative forcing? Specific suggestions for improvements are given below.

Some acronyms are not defined while other are defined and then not used.

Units – emissions are an amount per time. E.g. t/yr. Please correct throughout. Also, the text discusses emissions in terms of Gt CO₂-eq while the figures are in million tons, tons and kilotons of methane (I think – it is unclear). Please be consistent and clear to allow easier comparisons.

Figure descriptions are lacking in key details.

Specific comments

Abstract

“It also discusses the status of decoupling from economic growth for both methane emission accounts.” This is confusing. The decoupling of what from economic growth? What methane emission accounts? Do you mean “It also discusses the status of decoupling both production and consumption driven methane emissions from economic growth.”?

Please change “Developed countries are the only group that has managed to reduce both production- and consumption-based emissions...” to “Only developed countries have managed to reduce both production- and consumption-based emissions ...”

Please change “It identifies that the improvement in emission coefficients is the main determinant for reducing emissions...” to “It identifies that changes in emission coefficients, reflecting [insert reason here] are the main determinant for reducing emissions...”

Introduction

In light of the recent global methane pledge please change “...CO₂, which is the main focus...” to “...CO₂, which has been the main focus...”

Please change “Global CH₄ emissions have been rising for decades and since 2001 the annual growth rate reached 1.22% on average, which was slightly below the growth rate of CO₂ emissions (1.80%)” to “Global CH₄ emissions have been rising for decades, with the average (since 2001) annual growth rate reaching 1.22%, slightly below the growth rate of CO₂ emissions (1.80%)”

Please change “... specific global studies on methane are still scant...” to “... specific global studies on methane are not common...”. Considering the recent GCP methane budgets (e.g. Saunio et al. 2020) this seems a fairer assessment.

“Some emission datasets include countries’ CH₄ emissions as a part of GHG emissions (Saunio et al., 2020; He et al., 2020; European Commission, 2024; Jones et al., 2023) ...” Of the studies given in these references two are focuses solely on methane while the other two while investigating multiple GHG’s do consider methane as distinct from other GHGs. In contrast, the sentence suggests that that methane emissions are provided as a mixed component of total GHGs emissions. Secondly the sentence implies that each of these studies outline a dataset. This is not true. While references 6 and 13 outline datasets the Saunio et al. paper (reference 11) pulls together multiple existing datasets (including that described in ref 6) and compares them and He et al. (reference 12) uses an optimisation (top-down) approach to match emissions transported via a global chemical transport model to global methane observations. It relies on the CEDS and SSP2. Please rewrite this sentence and include further examples of bottom-up CH₄ emission datasets (e.g. CEDS).

“Most of the existing studies focused on selected countries and a small number of sectors such as agricultural outputs in EU countries...” Does this refer only to studies of decoupling? If it refers to studies of methane emissions in general than I’d disagree e.g. the papers listed in above take a global approach. I’d also suggest Stavert et al. (Global Change Biology, 28, 182-200, 10.1111/gcb.15901, 2022).

“... lack comprehensive coverage of a wide range of countries and high sectoral resolution...” followed by “...accounted for consumption-based CH₄ emissions of 26 sectors in 181 economies...”. 26 sectors and 181 economies doesn’t sound lacking in comprehensive coverage. Perhaps reword? E.g. “However, none of these studies were conducted in the last decade and many lack a comprehensive coverage of a wide range of countries and high sectoral resolution. For example,

Zhang et al. (2018) while having a broad sectoral and economic focus (26 sectors in 181 economies) had only a narrow temporal range (2000 – 2012) ...”

Please change “...to account for the most up-to-date time-series (1990 to 2023) consumption-based methane emissions of 164 countries/regions, which account for 98% of global methane emissions” to “...to examine consumption-based methane emissions of 164 countries/regions (accounting for 98% of global methane emissions) for the period 1990-2023.”

Please expand on how the GLORIA model works and what input data is used. Also what are MRIO tables? How were they used?

I would also suggest including here a sentence on where the production emissions estimates are coming from.

Please change “..discussion on..” to “...discussion of...”.

Methane emission trends

“Global methane emissions...” are these production or consumption emissions? Or both? Where have they come from? Are you referring to those from this study or earlier studies? Are these annual emissions? Please specify.

Please change “... at an annual growth rate...” to “... with an annual growth rate...”.

Please change “Since 2002, global emissions increased at a faster rate of 2.0%/yr and reached 329.5 million tons (9.2 Gt CO₂-eq) in 2008. One major reason is that China joined the World Trade Organization (WTO) in 2002, and its emissions surged due to increasing production for both domestic use and exports. The increased emissions in China from 2002 to 2008 (12.1 million tons) accounted for 32.5% of the global total increment (37.2 million tons).” To “Since 2002, global emissions increased at the faster rate of 2.0%/yr reaching 329.5 million tons (9.2 Gt CO₂-eq) in 2008. Following its joining of the World Trade Organization (WTO) in 2002 Chinese methane emissions surged, with its emissions increasing by 12.1 million tons between 2002 to 2008 accounting for 32.5% of the global total increment (37.2 million tons).”

Please change “After a decline in 2009 due to the global financial crisis, global methane emissions resumed its growth after 2009 at 1.1%/yr and finally reached 383.5 million tons (10.7 Gt CO₂-eq) in 2023. Unfortunately, the momentum of global methane emission growth shows no sign of slowing down, which contrasts with the recent slowdown observed in CO₂ emission growth” to “After a decline in 2009 due to the global financial crisis, global methane emission growth resumed at 1.1%/yr reaching 383.5 million tons (10.7 Gt CO₂-eq) in 2023. Unfortunately, the momentum of global methane emission growth shows no sign of slowing down, contrasting with the recent slowdown observed in CO₂ emission growth”

Please change “The gaps between countries’ PBE and CBE are known as net-emissions embodied in trade.” to “The gap between a country’s PBE and CBE is known as its net-emissions embodied in trade.”

“Asia and Developing Pacific has seen a change in the role from a net exporter to importer between 2010 to 2014.” The lines shown in Figure 1 for this region are very close together. Considering that there is no uncertainty estimate given how confident are you that this statement is true?

“Figure 1-b) and c) present the average PBE intensity and per capita CBE of country groups.” Please define intensity here. I’m assuming it is annual PBE divided by gross domestic product for the corresponding year in a specific currency? But it needs to be defined.

Please change “Per capita CBE has the opposite trend with emission intensity in most country groups.” to “Per capita CBE shows the opposite trend to emission intensity in most country groups.”

Decoupling of methane emissions and economic growth

Please change “In the last ten years from 2013 to 2023, there are 55 (33.5% of 164) countries that have reduced their PBE, ...” to “Between 2013 to 2023, 55 (33.5% of 164) reduced their PBE, ...”

Please change “...are developed ones at high economic levels...” to “...are developed at high economic levels...”

Please change “...PBE and CBE decline while GDP kept growing.” to “...PBE and CBE decline with GDP growth.” Note also that this is the first use of GDP and it has not be defined earlier.

Please change “...show the results for the most recent ten years (2013-2023).” to “...show the results for the period 2013-2023.”

I’d suggest including in this paragraph a fuller description of the decoupling index. Perhaps summarise the “A degree of strong decoupling ($DI \geq 1$ & $\frac{[GDP]_{t+1}}{[GDP]_t} \geq \frac{[GDP]_{t+1}}{[GDP]_t}$) implies that the country’s emissions decrease but the GDP keeps growing, weak decoupling ($0 \leq DI < 1$ & $\frac{[GDP]_{t+1}}{[GDP]_t} \geq \frac{[GDP]_{t+1}}{[GDP]_t}$) has both emissions and GDP increasing, but GDP is growing faster than emissions, while no decoupling ($DI < 0$ & $\frac{[GDP]_{t+1}}{[GDP]_t} \geq \frac{[GDP]_{t+1}}{[GDP]_t}$) means the country’s emissions are growing faster than its GDP.” sentence from the methods section?

“In contrast, the PBE intensities of strongly decoupled countries are significantly lower than those of weakly or non-decoupled countries (p-value = 0.000), mirroring the pattern seen in CO₂ emissions (Hubacek et al., 2021). This indicates that economic growth in weakly/non-decoupled countries is heavily driven by methane-intensive sectors such as agriculture and energy mining” How does this indicate this? What does the sectoral break down of these countries show?

Emissions embodied in trade and outsourcing

“In 2023, developing countries are the...” should this be “In 2023, developed countries are the...” ? That seems to be what Figure 3 is showing.

Drivers of consumption-based methane emissions

“Structural Decomposition Analysis on global emissions (as shown in Figure 5-a) and on each country group (as shown in Figure 5-b to f) at five-year intervals from 1998 to 2023.” Which does what? Assigns a contribution to emissions for five potential drivers?

“...while improvements in emission coefficients are the sole determinants of reduced emissions over the period...” please refer to earlier comments. This needs to be further explored. What exactly is driving changes in the emissions coefficients? Mitigation efforts? Which sectors?

“Improved production technology increased global methane emissions by 56.7 million tons (or 90%) from 2003 to 2018. However, there has been a reversal, leading to decreasing emissions in the last five years (2018-2023).” Considering the

focus of the paper (methane emissions) one would assume that “improved production” = less methane emissions so perhaps rephrase this? I’d suggest “Changes production technology between 2003 and 2018 increased global methane emissions by 56.7 million tons (or 90%), however, recent improvements have led to decreasing emissions in the last five years (2018-2023).”

Please change “The results show that population has the least degree of dispersion compared to the other four drivers, with a Standard Deviation of 0.04. The Standard Deviation of contributing effects of emission coefficient to total emission in each country are 0.54...” to “The results show that population has the least degree of dispersion compared to the other four drivers, with a SD of 0.04. The SD of contributing effects of emission coefficient to total emission in each country are 0.54...”

Conclusions

“Methane is the world’s second largest greenhouse gas emission source, contributing about 30% of global warming.”

Methane is not an emission source. Please re-word.

Please change “This study employs an environmental...” to “This study employs an environmental...”

Please change “...to analyze the most up-to-date emission changes...” to “...to analyze up-to-date emission changes...”

Please reword “It is essential to recognize that, although other emerging countries currently emit lower levels of methane...”

This reads as referring to USA and China as emerging countries.

“...they may become significant methane emitters in the future due to growth of agriculture and mining sectors...” Surely that’s dependent on the natural resources available to each country?

“When factoring in the dimension of economic growth, our decoupling analysis reveals that developed countries stand out as the sole group that has successfully achieved strong decoupling between emissions and GDP.” This is good news but how did they do it? Which sectors decreased? What policy/economic/technological mechanisms were used? This is the most interesting bit please expand.

“Our primary objective is to reduce emissions while sustaining economic growth, essentially striving for sustainable transitions” Our? Our = the authors? No, the individual countries.

“Brazil (15.7% of the global total import), India (11.9%), and the US (9.3%) are the top three emitting countries of fertilizers.”

Do you mean emitters of fertilisers or emitters of methane due to fertiliser production?

Please change “... more and more trade restrictions...” to “... additional trade restrictions...”

Please change “The EU and US may need...” to “As such, the EU and US may need...”

Please change “This indicates that Russia has relatively advanced production technology and efficiency for fertilizer production.” to “This suggests that Russia has relatively advanced production technology and efficiency for fertilizer production minimizing associated methane emissions.”

Please change “Mechanism (CBAM), that confines the import of high-emitting products...” to “Mechanism (CBAM), that aims to minimise the import of high-emitting products...” or “Mechanism (CBAM), that aims to reduce the import of high-emitting products...”

Please change “CBAM includes fertilizers, which are one of the top methane sources,...” to “CBAM includes fertilizers, the production of which is one of the largest methane sources,....”

“Although CBAM can effectively reduce consumption-based emissions in the EU by importing from cleaner-producing countries, it could be a potential trade barrier that obstructs globalization.” As the focus of the paper is the reduction of methane emissions, I would suggest either removing this sentence or linking changes in the level of globalization to methane emissions.

“...improvements in emission coefficients...” please refer to earlier comments.

“One possible explanation is that an upturn in living standards has led to increased consumption of red meat (e.g., beef and lamb), which is very methane-intensive.” Is this indicated by changes in the sectoral emissions distribution for these countries? You have the data to explore this.

“...it is imperative to allocate sufficient room for methane in the future carbon budget...” I’m not sure what the authors mean by this. Are they suggesting incorporating methane into a global carbon budget? A combined global GHG budget? Or are they suggesting focusing policy decisions/political effort etc. on reducing methane emissions? In which case recent efforts e.g. <https://www.globalmethanepledge.org> are already working that direction.

Figures

Figure 1 – Please add an extra panel or figure or table where the composition of the regions (ie. which country is where) is defined. In the supplement would be sufficient. Please correct the units. Emissions are amount per unit of time. E.g. million t/yr. The SI symbol for ton is t. Please use it consistently.

Figure 2 – What are the grey dots? I assume they are the individual countries but this is not specified. What is “delta CBE” and “delta PBE”? These need to be defined in the text or here in the caption.

Figure 4 – Are these sectoral distributions, like Fig 1, also from EDGAR?

Figure 5 – for b to f percentage of what? Is it contribution to emissions change? More information needed either here or in the text.

Reviewer #3

(Remarks to the Author)

See the PDF file attached.

Version 1:

Reviewer comments:

Reviewer #1

(Remarks to the Author)

The paper has been revised and is now much clearer in its analysis and methods used. However, there are areas which require further addressing before the manuscript can be considered for publication.

- There are still grammatical errors and incorrect tenses throughout the manuscript, as well as sections where the writing quality is poor. Given the journal the authors are trying to publish in, these should be improved and corrected e.g. line 139 'have' not 'has'; line 143 'kept' is unnecessary and 'increased' not 'increasing'; line 283 'was' or 'remained' not 'kept'
- Section 'Drivers of consumption-based methane emissions' is missing references in parts where it appears the authors are referring to the literature e.g. lines 342 to 348.
- Lines 345 to 348- can the authors expand on the reduction in emission coefficients? Especially for animal raising and cattle, how has the coefficient reduced by 42 and 20%?

Reviewer #2

(Remarks to the Author)

General comments

Changes made by the authors has significantly strengthened the introduction. Specifically, the increased discussion of existing studies has provided key context and justification for the study while the early explanation of the PBE and CBE concepts has improved the readability of the paper.

The increased detail provided in the "Methane emissions" section is useful and interesting. However, it could be expressed more concisely. For example,

"In terms of PBE, we can see that Asia and the developing Pacific has been dominating global methane PBE since the mid-1990s. Its PBE increased rapidly from 96.9 million tons/yr (36.4% of global emissions) in 1990 to 162.4 (42.4%) in 2023. In the meanwhile, developed countries, although being the 2nd largest contributor to global methane PBE, show a declining trend. Their PBE decreased from 74.7 million tons/yr (28.0%) in 1990 to 60.4 (15.8%) in 2023."

Could be expressed as,

"Since the mid-1990's the Asian and developing Pacific PBE has dominated global PBE, increasing from 96.9 million t/yr (36.4% of global PBE) in 1990 to 162.4 (42.4%) in 2023. Meanwhile, the PBE of Developed Countries, globally the 2nd largest contributor, decreased from 74.7 million t/yr (28.0%) in 1990 to 60.4 (15.8%) in 2023."

I remain concerned that the paper conflates a change in emissions coefficient with a change in actual emissions (not just a change in the estimates of these emissions).

Emissions coefficients are a best guess at relating sectoral emissions to the activity of that sector. However, their accuracy is limited and for many sectors they have very large uncertainties. This varies globally with some nations using country-specific derived emissions coefficients (nominally) reflective of their emissions portfolio while others use default estimates from the IPCC. Also of concern is that, for some sectors, reported emissions based on emissions coefficients differ significantly to those determined using Top-Down approaches (based on satellite and/or ground-based observations i.e. independent of emissions coefficients). E.g. Sadavarte et al. (2021) found coal mine methane emissions seven times higher than those estimated by the EDGAR approach.

As such, I am concerned by the emphasis in the Conclusion of increasing/focusing trade with countries with low emissions coefficients (e.g. page 14, line 31) as a method to reduce emissions as it embeds an unarticulated (and somewhat tenuous) assumption that the emissions coefficients are an accurate representation of actual emissions. Instead, trade should be focused on countries with low emissions (relative to the item being traded). I do not think an in-depth exploration of the limitations of emissions coefficients is within the scope of the paper. However, I do think that the Conclusion needs to include a statement acknowledging that:

1. The paper assumes that the emissions coefficients are an accurate representation of actual methane emissions but emissions coefficients can (and do) have large uncertainties,
2. Prioritising trade with low emissions coefficient partners will only reduce emissions if said coefficients are accurate representations of emissions.
3. Top-down (observation based) research is needed to independently verify that such changes in trading patterns leads to a decline in emissions.

Lastly, while I have suggested a number of small changes below, the paper needs to be copy edited to improve clarity and succinctness.

Specific comments

I would suggest that the words "satellite data" are not used in the paper unless referring to satellite (orbiting space object derived) data. As also evidenced in comments by another reviewer and considering the target audience of the paper the use of the word adds an unnecessary layer of confusion.

Location

P 2, In 12 Please change "air pollutant" to "air pollutants"

P 3, In 11 Please change "spanning from 2000" to "spanning 2000"

P 3, In 36 Please change "Based on our calculation" to "Based on our calculations"

P4, Ins 14-15 Please change "the Supporting Document" to "the supporting documentation" or "the supplementary".

P5, In 5 Please change "Supplementary document" to "Supplementary".

P5, In 7 Please remove "perspectives" it's superfluous.

P5, In 10 Here “developed countries” is referring to the specific category you have defined in your study. As such, it should be capitalised to “Developed Countries” to distinguish it from the general term. This should be changed throughout the paper when referring to this specific category.

P6, In 16-18 The way this is currently written it seems to imply that Figure 1 d) shows the global totals aggregated into the sectors. It’s also not clear whether PBE or CBE is under discussion. For clarity I’d suggest rewording it as “To illustrate the sectoral distribution of methane PBE, we aggregated 120 sectors into seven categories, while underlying calculations are done at the 120-sector resolution. These seven sectors are shown in Figure 1-d) for the ten nations with the highest PBE.”

P8, In 20 I’m assuming when you say “developing groups” you’re referring to countries not listed in the “Developed Countries” category? But it almost reads as if you’re referring to a new category here. Perhaps “Some countries not in the Developed Country category have already achieved strong decoupling in terms of both PBE and CBE. For example, Armenia...” would be clearer.

P8, In 26 Please change “time period” to either “a time” or “a period”.

P8, In 27 Please change “...decoupling results ... for every ten years’ interval...” to “...decoupling results ... at ten year intervals...”

P9, In 4-6 Please change “but their proportion in global total emission kept unchanged” to “but as a proportion of global total emission remained unchanged”

P9, In 6 Please change “In 2023, developed countries are the” to “In 2023, Developed Countries were the”

P11, In 10 Not all developed countries are in the North (e.g Australia and NZ). Perhaps instead of “(i.e. developed countries)” try “(i.e. dominated by developed countries)”

P11, In 13-14 Please change “A possible explanation of these patterns is that global North are outsourcing resource...” to “A possible explanation of these patterns is that the global North outsources resource...”

P15, In 3 Please change “...emissions.Countries...” to “...emissions. Countries...”

P15, In 16 Please change “coefficient” to “coefficients”

Version 2:

Reviewer comments:

Reviewer #1

(Remarks to the Author)

I am overall happy with the revisions made to the manuscript and the response to my comments. There are three minor punctuation/formatting issues but the manuscript is more or less in a state for acceptance for publication:

-lines 105 to 106: text has been accidentally superscript

-line 181: space between 0.05 and in

line 448: no space between 'heating' and the semicolon

Response letter to Reviewers' comments

Dear editor and reviewers,

Thank you very much for your consideration of our submission and your efforts in managing its review. We very much appreciate the comments we received, which are very constructive and will greatly improve our paper. Please find our point-by-point response to the comments listed below.

Reviewer #1

The manuscript titled 'Global methane footprints growth and drivers 1990-2023' presents an assessment of global methane emissions based on production and consumption for 164 countries. The study uses a mixture of satellite and inventory (EDGAR and World Bank) data for the analysis and finds that there is a relationship between consumption and production to methane emissions and decoupling methane from economic growth is tricky. The paper offers an interesting insight into methane emissions and a different approach to emissions accounting, but the paper requires major revisions before it can be considered for publication in Nature Communications. I outline my concerns below:

RE: Thanks for your recognition of our work.

The paper is too long and does not fit the requirements specified for articles published in Nature Communications

RE: According to the suggestion of the editor, we focused on the text's content in this round of revision. We will finalize the length and formatting issues with the publishing team at a later time.

The GLORIA data used in the study, what resolution and frequency are the data? Satellite measurements of methane are traditionally used on a regional or site specific basis as the resolution of satellites, bar GHGSat and Sentinel-5P (when paired with data from Sentinel-2), is too large for useful studies and the data is often patchy.

RE: "Satellite accounts" in input-output studies refer to environmental data (e.g., greenhouse gas emissions, land use, water use, material consumption), rather than measurements of emissions using satellite observations. We have added an explanation in the methods section.

We collect methane PBE from the satellite (environmental) accounts of GLORIA ¹, which provides emissions data by countries and products consistent with the GLORIA MRIO table. The emission data includes both biogenic methane, produced and released from living plants and animals, and emissions from fossil fuel use. GLORIA satellite accounts combine EDGAR, the Organization for Economic Cooperation and Development, and Eurostat emission datasets.

Also, in the descriptive note for Figure 1, where the "satellite data" first appears:

Data source: the PBE and CBE of countries used in sub-figure a) are calculated based on GLORIA satellite data (environmental accounts) and MRIO tables ¹.

The emission coefficient is stated as an important factor in reducing emissions, but this term is never explained.

RE: The emission coefficient refers to production-based methane emissions per unit of sectoral output. We have added explanations in the revised manuscript.

In Abstract:

It identifies that the reduction in emission coefficient, reflecting improvements in emission technology, is the main determinant for reducing emissions over the observation period and can offset the increasing effects from growth of final demand.

In Results:

Changes in emission coefficient (i.e., emissions per unit of output) are the main determinant to reduce emissions over the period, offsetting the increasing effects from demand growth and structure changes, which is consistent with previous studies²⁻⁴. This driver reflects improvement in the emission technology during production which are reducing methane emissions.

The paper talks in great detail about agriculture and waste methane emissions but does not mention oil and gas methane emissions. As many initiatives to tackle methane are related to the oil and gas sector, and globally the oil and gas sector is estimated to account for around 20% of anthropogenic emissions, this seems like an oversight.

RE: We have added more description of sectoral emissions from oil and gas, and redone figures 1 and 4 to highlight the contributions from the oil and gas sector.

Figure 1:

To illustrate the sectoral distribution of methane emissions, we aggregated 120 sectors into seven categories, as shown in Figure 1-d), the underlying calculations are done at the 120-sector resolution. Globally, agriculture and food production are the largest contributors to methane emissions, accounting for 47% in 2023, followed by waste and recovery (17%), coal and electricity (14%), and oil and gas (13%). Over time, the proportions of sectoral emissions have shifted. Agriculture and food production decreased from 54% in 1990 to 47% in 2023, and oil and gas declined from 18% in 2006 to 13% in 2023. In contrast, emissions from coal and electricity production rose from 9% in 2000 to 14% in 2023, reflecting sectoral structure changes in global methane

emissions. Countries have a huge difference in the sectoral structure of methane emissions, as shown in Figure 1-d), which is closely associated with the economic structure of the countries. For example, coal and electricity production, especially extraction of hard coal, is the largest emission source in China and Indonesia, contributing to 35% and 56% of their PBE, respectively. The two countries account for 59.3% of global coal production in 2023 and therefore emit a huge amount of methane from coal mining⁵. In contrast, Russia and Iran have higher emissions from oil and gas production (33% and 57% respectively). India and Brazil emit large shares of emissions from agriculture (67% and 73% respectively), e.g., pigs and cattle due to livestock farming.

Figure 4:

When examining emissions flows by sectors, we observe considerable differences across countries. For example, in 2023, developed countries have the largest share of emissions embodied in their agricultural and food-related exports (57.6%), while the major sectors for Africa and the Middle East's emissions embodied in exports are oil and gas (27.2%). Agricultural and food products account for 20.5% of Africa and the Middle East's exported emissions. Latin America and the Caribbean export substantial amounts of agricultural and food products, accounting for 73.4% of their exported embodied methane emissions. In contrast, Asia and the Developing Pacific region have 47.4% of their emissions embodied in exports associated with coal and electricity production. Figure 4 shows detailed trade embodied emissions by sectors in top-emitting countries.

When we classify countries into the global north (i.e., developed countries) and south, we observe that emissions embodied in North-to-South trade primarily originate from agriculture (63.6% in 2023). In contrast, South-to-North trade is predominantly focused on resource extraction (i.e., coal, oil, gas and mineral mining), accounting for 45.4% of emissions embodied in exports. A possible explanation of these patterns is that global North are outsourcing resource mining industries to the global South, with a flow of resources back through global supply chains^{6,7}. More importantly, these trade patterns have intensified over the past decades. Detailed sectoral emissions embodied in trade of countries are provided in **Error! Reference source not found.**

The International Energy Agency (IEA) have a methane tracker that is often used for methane data. Have the results of the study been compared to this (for 2022 to 2023)?

RE: We have compared GLORIA production-based emissions with the IEA methane tracker for the year 2023. IEA's emissions are 349.5 million tons, which is 9.7% lower than our emissions of 383.5 million tons in 2023. Meanwhile, we also compared the GLORIA emissions with

EDGAR v8.0_GHG 1970-2022 (CO₂, CH₄, N₂O, F-gases). We have added a paragraph in Methods – Data source to show the discrepancy.

In Methods-Data source:

To verify GLORIA satellite accounts, we utilize global methane emission data from EDGAR v8.0_GHG 1970-2022 (CO₂, CH₄, N₂O, F-gases) ⁸ and IEA Methane Tracker ⁹. Comparisons reveal that GLORIA methane emissions (379.5 million tons in 2022 and 383.5 in 2023) are 0.9% lower than EDGAR estimates in 2022 (388.3 million tons) and 9.7% higher than IEA Methane Tracker in 2023 (349.5 million tons). Detailed comparison of countries' emissions between GLORIA and EDGAR are shown in the Supplementary document.

There are some minor grammatical issues in the paper which can be addressed from detailed proof reading (but overall writing is of high quality but some minor issues with tenses)

RE: The revised manuscript has been proofread by a native speaker, and we have addressed various language issues throughout the text. We are confident that the remaining typos and language errors will be picked up during the final proofing stages.

The authors explain the general reasons for the changes in PBE and CBE, but more detailed explanations could be given to better explain why some countries and regions experience such large growths in emissions e.g. is China's big increase in methane entirely due to coal or did the improvement in diet (more meat and access to food) also have an impact?

RE: We have added further analyses of emission drivers for the five top-emitting countries (i.e., China, India, USA, Brazil and Russia) in the supplementary document Table S1.

Error! Reference source not found. *below presents the contributions of each emission driver in selected top-emitting countries, with global results provided for comparison. Significant variability is observed among countries, for instance, between 2018 and 2023, the 'emission coefficient' driver reduced China's emissions by 65% and India's by 47%, even though both countries belong to the Asia and Developing Pacific region. Overall, the 'emission coefficient' driver decreased emissions across countries, while drivers including 'production technology', 'final demand per capita', and 'population' generally increased emissions (with a few exceptions), that align with global trends. The driver of 'final demand structure' exhibited more significant variations across different countries and time periods.*

Table 1 Methane emission drivers of top-emitting countries, in million tons

Country	Period	Emission coefficient	Production technology	Demand structure	Demand/capita	Population	Emission changes
Global	1998-2003	-75.48	22.07	-1.21	53.29	16.43	15.11
	2003-2008	-349.14	62.30	60.03	237.04	18.13	28.35
	2008-2013	-412.09	145.82	154.95	110.95	18.73	18.37
	2013-2018	-116.16	18.27	71.50	23.97	18.61	16.20
	2018-2023	-272.61	-56.23	234.38	100.85	13.06	19.46
China	1998-2003	-6.40	-0.76	-8.93	19.67	1.51	5.08
	2003-2008	-32.66	-0.49	-6.00	46.60	1.33	8.78
	2008-2013	-43.70	4.60	-1.83	50.80	1.67	11.54
	2013-2018	-7.26	-0.96	-11.57	24.38	1.92	6.51
	2018-2023	-45.67	-2.20	24.60	34.94	0.50	12.17
India	1998-2003	-106.07	6.47	93.03	5.37	2.06	0.87
	2003-2008	-18.85	1.52	-1.84	19.64	1.89	2.37
	2008-2013	-5.77	4.26	-9.47	8.25	1.71	-1.01
	2013-2018	-4.72	1.39	-3.04	7.83	1.54	3.00
	2018-2023	-12.96	-0.38	4.55	7.85	1.15	0.21
USA	1998-2003	-11.74	5.23	0.00	7.02	1.63	2.14
	2003-2008	-23.73	11.43	2.49	7.35	1.56	-0.90
	2008-2013	-246.35	91.58	147.44	2.11	1.18	-4.05
	2013-2018	1.45	-1.90	-2.27	5.04	1.01	3.33
	2018-2023	-19.06	-5.23	10.03	8.31	0.68	-5.27
Brazil	1998-2003	1.22	1.09	5.76	-8.18	0.92	0.82
	2003-2008	-19.33	0.76	0.42	19.19	0.82	1.85
	2008-2013	-7.67	2.16	0.52	6.62	0.80	2.43
	2013-2018	9.95	-0.13	-6.16	-5.70	0.74	-1.30
	2018-2023	-9.50	-0.60	8.88	0.43	0.50	-0.28
Russia	1998-2003	110.27	-10.51	-104.80	3.42	-0.17	-1.80
	2003-2008	-64.36	8.80	43.79	14.78	-0.12	2.88
	2008-2013	-4.03	0.87	0.53	4.32	0.06	1.75
	2013-2018	5.50	1.34	-4.45	-4.31	0.08	-1.84
	2018-2023	-62.54	-0.19	64.85	1.49	-0.01	3.60

It is not clear from that paper what actions should be taken to reduce methane emissions and how countries can effectively decouple emissions from growth (especially in a permanent or long term way).

RE: We have added more suggestions on methane emission reduction in the conclusion section.

We find that developed countries stand out as the sole group that has consistently achieved emission reductions and strong decoupling between emissions (both PBE and CBE) and GDP. Their decoupling is mainly caused by a decline in their emission coefficient, and to a lesser degree due to outsourcing methane emissions to less developed regions. Among the 120 sectors classified in the GLORIA configuration in developed countries, 100 of them reduced their emission coefficient between 2003 and 2013. This is evident from the analysis of the drivers, where emission coefficient is the primary determinant in reducing emissions. Therefore, many sectors in developed countries provide successful examples in methane emission control for other countries around the world.

As sectors across countries differ in technological endowment, production efficiency, emission coefficient of products produced in different countries varies. This provides an opportunity to reduce a sector's methane footprint by supply chain management of upstream production through carefully selecting import partners with low emission coefficient. Taking the production and trade of fertilizers as an example, the production of which is one of the top methane sources, Russia is the world's leading exporter of fertilizers. It exported 18.7 billion USD worth in 2022, accounting for 13.1% of the global total export, followed by Canada (14.5 billion USD), China (12.7), US (8.19), and Morocco (7.92)¹⁰. Our results find that the emission coefficient in Russia's fertilizer production is 0.07 tons/million USD/yr in 2022. However, the emission coefficients of fertilizer production in Morocco (with 0.002 tons/million USD/yr), Canada (0.05), US (0.04) and China (0.06) are lower than in Russia, indicating these countries have more advanced production technologies with higher efficiency or alternative sources for fertilizer production, reducing associated methane emissions. Countries that currently import fertilizer from Russia can reduce their imported embodied emissions by shifting to other exporting countries with lower emission coefficient. However, it is important to consider that this shift could lead to other

environmental problems (e.g. in the case of Morocco, which has a larger share of phosphate fertilizers but lower methane emissions). It is encouraging to see the EU making efforts to reduce the import of high-emitting products from non-European countries by setting up the Carbon Border Adjustment Mechanism (CBAM), which will invite EU countries to choose cleaner trade partners and upstream partners to reduce their emissions to stay competitive. CBAM includes fertilizers as one of the initial goods and selected precursors whose production is emission-intensive and at most significant risk of carbon leakage. The EU accounts for 23% of global fertilizer imports. Future actions of CBAM should focus on expanding to a larger range of products, including methane-intensive ones ¹¹.

Reducing methane emission coefficient requires targeted strategies across sectors. For example, the energy sector can apply advanced detection technologies, such as infrared imaging and drones, in oil and gas extraction, storage, and transportation to identify and repair emission leaks ^{12,13}; replace coal and oil with clean energy across the entire energy system ¹⁴; improve ventilation systems in oil and coal mines to reduce uncontrolled methane emissions ¹⁵; and deploy equipment to capture associated methane emissions for recycling rather than flaring. In the agricultural sector, we can improve feed formulations to reduce methane production in ruminants by incorporating additives like oils, enzymes, or methane inhibitors into livestock diets ^{16,17}; increase biogas recovery by processing manure and organic waste into energy ¹⁸; and improve rice farming through irrigation control (e.g., alternate wetting and drying methods) and better fertilization management ¹⁹. In the waste management sector, we can enhance landfill management by sealing landfills and installing methane recovery systems to recycle methane for power generation or heating ; promote composting technologies by replacing landfilling with composting to effectively reduce methane emissions from organic waste ²⁰; and upgrade wastewater treatment by adopting advanced anaerobic digestion technologies to capture methane from wastewater and use it as an energy source ^{21,22}.

The measures above can reduce emission coefficients from the production side. However, unlike CO₂ emissions, which mainly stem from fossil fuels, a major source of methane emissions are biological activities, such as biodegradation in wastewater treatment, fertilizers production, and effusion from coal mining, livestock farming. We further call for smart demand-side emission reduction strategies to supplement the aforementioned reduction measures in mitigating methane emissions, particularly from biological activities.

Reviewer #2

General comments

The consumption/production analysis outlined within this paper is an interesting and pertinent approach to investigating the trends and drivers in anthropogenic methane emissions. Considering that earlier studies of this type are now over a decade old this will be a topic of interest to many and useful to the wider scientific community. However, I have a number of concerns which must be addressed before considering publication.

RE: Thanks for your recognition of our work.

The comparison of consumption/production emissions is a very interesting approach and useful to better understand the global scale flows and drivers of emissions. But its benefits need to be better emphasised and articulated within the paper. What does this approach tell you that the standard (typically production) doesn't? This should be your focus.

RE: The production and consumption-based approaches represent different methods of allocating global emissions. The former assigns emissions to the countries where they are produced (to answer where emissions arise), while the latter allocates the emissions to the countries where the goods are consumed (to answer the question of why we have these emissions or the ultimate drivers of emissions). In our revised paper, we have compared the two approaches emission in the following aspects.

Emission trends in the first result section and figure 1:

Figure 1-a) shows the emission trends by country groups in terms of production (solid lines) and consumption perspectives (dashed lines). In terms of PBE, we can see that Asia and the developing Pacific has been dominating global methane PBE since the mid-1990s. Its PBE increased rapidly from 96.9 million tons/yr (36.4% of global emissions) in 1990 to 162.4 (42.4%) in 2023. In the meanwhile, developed countries, although being the 2nd largest contributor to global methane PBE, show a declining trend. Their PBE decreased from 74.7 million tons/yr (28.0%) in 1990 to 60.4 (15.8%) in 2023. The PBE from other country groups kept slowly increasing over the past decades and reached 79.7 million tons/yr (20.8% of global emissions, Africa and Middle East), 53.5 (13.9%, Latin America and Caribbean) and 27.5 (7.2%, Eastern Europe and West-Central Asia) in 2023. The top ten emitting countries shown in Figure 1-d) contributed 56.8% of global PBE in 2023, reflecting an extremely uneven distribution of methane emissions among countries. In terms of CBE, Asia and developing Pacific surpassed the developed countries after 1999 and became the world's top CBE emitter. The CBE of Asia and developing Pacific increased from 32.6% (86.9 million tons/yr) in 1990 to 44.1% (169.2) in 2023. Developed countries have declined CBE from 38.8% (103.5 million tons/yr) to 21.5% (82.6) over the same period.

When comparing a country's PBE and CBE, the gap between them is known as net-emissions embodied in trade. A larger PBE than CBE implies that the country is a net exporter of emissions embodied in goods and services, while a net emission importer (i.e., CBE higher than PBE) causes more emissions embodied in goods and services than it produces. Developed countries are the only global net importer over the whole observed period. Their net emissions embodied in trade varied between 20.2 and 35.6 million tons/yr over the period, which accounted for 24.4%-34.0% of their CBE. Asia and Developing Pacific has seen a change in the role from a net exporter to importer between 2010 to 2014. After 2014, their imported embodied emissions increased from 0.2 (0.1%) to 6.8 million tons/yr (4.0% of CBE). All the other country groups are net emission exporters with 12.1% to 65.7% of their national PBE caused by export production. Our findings align with prior research ^{2,3,23}, underscoring the increasing complexity of methane emissions driven by international trade and changing patterns. More details will be discussed in the section below.

Decoupling of emissions and economic growth in the second result section, including figure 2 and table 1:

Between 2013 to 2023, 55 (33.5% of 164) countries reduced their PBE, 57 (34.8%) countries reduced their CBE, and 30 (18.3%) countries reduced both. 17 of the 30 countries that reduced both PBE and CBE are high-income countries. Among the five country groups (shown in solid dots in Figure 1 Methane emissions by income country groups and top ten countries.

Note: Sub-figures a) shows PBE (in solid lines) and CBE (in dashed lines) of country groups during 1990-2023; b) and c) show PBE intensity (i.e., PBE per unit of GDP) and per capita CBE of country groups, respectively; and d) shows sectoral PBE in top ten countries in 2023. Data source: the PBE and CBE of countries used in sub-figure a) are

calculated based on GLORIA satellite data (environmental accounts) and MRIO tables¹; PBE intensity and per capita CBE in sub-figure b) are calculated with World Bank dataset; the sectoral emissions in sub-figure d) are collected from EDGAR⁸. Detailed PBE and CBE of countries and aggregated groups are provided in **Error! Reference source not found.**. A global map for the five aggregated regions is shown in the Supplementary document.

Figure 2 Decoupling of methane emissions and economic growth in selected countries at ten-year intervals

Note: Sub-figures a) and b) show the relationship between PBE, CBE and GDP during 1993-2023 at ten-year intervals, the coloured dots present five country groups, and the coloured stars present selected seven countries, the grey dots present the remaining countries. Sub-figures c) and d) show the decoupling results of countries/groups in coordinate systems. Delta GDP and emissions refers to the changes in GDP and emissions over the given period. Detailed results of decoupling are shown in **Error! Reference source not found.**

Figure 3 Emissions embodied in trade 1998 and 2023

Note: The figure includes the emissions flow from one country to another (i.e., emissions produced in one country but consumed in another country), and exclude the emissions consumed in the same country where they are produced. The country groups on the left y-axis are producers while country groups on the right y-axis are consumers. The flows between the same country group present emissions embodied in trade between different countries within the group.

Figure 4 Sectoral emissions embodied in export of countries 1998 and 2023

Note: sectors distributions in the figure are aggregated based on GLORIA sectors according to their emissions.

Figure 5 Emission drivers of consumption-based methane emissions at five-year intervals

Note: sub-figure a) shows the contributing effects of each driver to global emissions in absolute values, sub-figures b) to f) shows contributing effects of each driver to five country groups in percentages relative to the group's total emissions.

-a and b), Developed Countries is the only group that have achieved both PBE and CBE decline and GDP growth. We also notice a decline in emissions not only in developed countries, but also in some developing countries. 13 developing countries have reduced both PBE and CBE.

We use the Tapio Decoupling Index to describe the dynamic relationship between emissions (both PBE and CBE) and economic growth (i.e., GDP) in each country and aggregated group during the years 2013 to 2023. Strong decoupling occurs when a country's emissions decrease while its GDP continues to grow. Weak decoupling happens when both emissions and GDP increase, but GDP grows at a faster rate than emissions. In contrast, no decoupling indicates that emissions are rising at a faster rate than GDP.

Figure 1 Methane emissions by income country groups and top ten countries.

Note: Sub-figures a) shows PBE (in solid lines) and CBE (in dashed lines) of country groups during 1990-2023; b) and c) show PBE intensity (i.e., PBE per unit of GDP) and per capita CBE of country groups, respectively; and d) shows sectoral PBE in top ten countries in 2023. Data source: the PBE and CBE of countries used in sub-figure a) are

calculated based on GLORIA satellite data (environmental accounts) and MRIO tables¹; PBE intensity and per capita CBE in sub-figure b) are calculated with World Bank dataset; the sectoral emissions in sub-figure d) are collected from EDGAR⁸. Detailed PBE and CBE of countries and aggregated groups are provided in **Error! Reference source not found.**. A global map for the five aggregated regions is shown in the Supplementary document.

Figure 2 Decoupling of methane emissions and economic growth in selected countries at ten-year intervals

Note: Sub-figures a) and b) show the relationship between PBE, CBE and GDP during 1993-2023 at ten-year intervals, the coloured dots present five country groups, and the coloured stars present selected seven countries, the grey dots present the remaining countries. Sub-figures c) and d) show the decoupling results of countries/groups in coordinate systems. Delta GDP and emissions refers to the changes in GDP and emissions over the given period. Detailed results of decoupling are shown in **Error! Reference source not found.**

Figure 3 Emissions embodied in trade 1998 and 2023

Note: The figure includes the emissions flow from one country to another (i.e., emissions produced in one country but consumed in another country), and exclude the emissions consumed in the same country where they are produced. The country groups on the left y-axis are producers while country groups on the right y-axis are consumers. The flows between the same country group present emissions embodied in trade between different countries within the group.

Figure 4 Sectoral emissions embodied in export of countries 1998 and 2023

Note: sectors distributions in the figure are aggregated based on GLORIA sectors according to their emissions.

Figure 5 Emission drivers of consumption-based methane emissions at five-year intervals

Note: sub-figure a) shows the contributing effects of each driver to global emissions in absolute values, sub-figures b) to f) shows contributing effects of each driver to five country groups in percentages relative to the group's total emissions.

-c) and d) show the results for the period 2013-2023. We find that among the 55 countries that have reduced their PBE, 48 countries have achieved strong decoupling of PBE and GDP, indicating that there are 7 countries reducing PBE due to economic recessions (i.e., belonging to the group of recessive-no-decoupling or recessive-weak-decoupling). 47 countries have achieved strong decoupling of CBE and GDP, among which 23 countries also achieved PBE decoupling.

And the whole third section on Emissions embodied in trade and outsourcing is based on comparisons between PBE and CBE as emissions embodied in trade accounts for the difference between PBE and CBE.

Although a number of developed countries have decoupled emissions from economic growth, this might be achieved by outsourcing emission-intensive industries to less developed regions²⁴. Previous studies have noted this trend in methane emissions, despite domestic reductions being achieved^{3,23}. Such outsourcing will largely increase the emissions from less developed regions with lagging levels of technology and lower production efficiency, which will lead to a possible rise in global emissions²⁵. This is especially the case for countries that have achieved strong decoupling of PBE and GDP,

where no such decoupling is observed between CBE and GDP. For example, the PBE of New Zealand decreased by 2.8% from 2013 to 2023 while its CBE increased by 49.3%.

The entire fourth section on emission drivers explains how the consumption structure and size have contributed to the growth of methane emissions, which cannot be analysed using typically production-based emissions.

There is a distinct lack of context. While earlier studies are mentioned (paragraph 4 of the introduction) at no point are the results of this study compared to these studies. Is there the same general pattern of emissions distribution between sectors (e.g. compared to the GCP CH4 budget)? Are the trends over time similar to earlier studies? The discussion does not need to be extensive, but the results need context.

RE: We have added comparisons with previous studies.

In the first result section of emission trends:

Compared to previous studies focusing on emission trends before 2014^{3,26}, our result further highlights a resurgence in methane emissions after 2015, particularly in Asia and rapidly developing countries, driven by ongoing economic and population growth. That is to say global methane emission have been growing faster in recent years⁴¹, albeit with some fluctuations, which is in contrast with the recent slowdown observed in CO₂ emission growth⁴².

When comparing a country's PBE and CBE, the gap between them is known as net-emissions embodied in trade. ... Our findings align with prior research^{2,3,23}, underscoring the increasing complexity of methane emissions driven by international trade and changing patterns. More details will be discussed in the section below.

In the third result section of trade embodied emissions:

Previous studies have noted this trend in methane emissions, despite domestic reductions being achieved^{3,23}. Such outsourcing will largely increase the emissions from less developed regions with lagging levels of technology and lower production efficiency, which will lead to a possible rise in global emissions²⁵. This is especially the case for countries that have achieved strong decoupling of PBE and GDP, where no such decoupling is observed between CBE and GDP. For example, the PBE of New Zealand decreased by 2.8% from 2013 to 2023 while its CBE increased by 49.3%. Sun, et al.² also noted emissions transfer from developed countries to emerging countries in agriculture and food manufacturing sectors align with global trade.

In the fourth result section of emission drivers:

The global results indicate that economic growth, represented by final demand per capita in this study, has been identified as a key driver of increased emissions, which has been discussed previously³. In addition, we find that changes in structure of final demand play an important role in the increase of emissions, particularly after 2008.

Changes in emission coefficient (i.e., emissions per unit of output) are the main determinant to reduce emissions over the period, offsetting the increasing effects from demand growth and structure changes, which is consistent with previous studies²⁻⁴.

There are no uncertainty estimates given on any of the data. How can a reader be sure that these are actual trends/differences and not just noise?

RE: We added an uncertainty analysis of our estimates in the Supplementary document.

Uncertainty and robust analysis of global methane footprints

Method

Uncertainty assessment is an important tool in emission accounting that helps assess the robustness of emission estimates and facilitates efforts to improve their accuracy^{28,29}. The uncertainties of production-based emissions (PBE) can easily be assessed with Monte Carlo simulations based on the variances of emission factors and activity data³⁰. However, estimating the uncertainties of consumption-based emissions (CBE) is a more complex task as the uncertainties arise not only from the PBE intensity but also from input-output data^{31,32}. There are a number of studies investigating uncertainties of CBE. For example, Schulte, et al.³³ applied Monte Carlo simulations to analyze how uncertainty from raw data propagates to PBE. They also showed that the uncertainty can further propagate to CBE via inter-country input-output linkages. Rodrigues, et al.³⁴ proposed that the propagated uncertainties from PBE to CBE can be analyzed using available individual multi-regional input-output (MRIO) datasets. Therefore, in this study, we use three alternative PBE accounts (EDGAR⁸, GLORIA and GTAP satellite dataset) and two alternative MRIO datasets (GLORIA¹ and GTAP³⁵) to check the uncertainties of CBE, and we conclude that we derive robust results. Detailed results of each country are shown in Table S4.

Uncertainty and robustness

We first compare the methane PBE estimates from EDGAR, GLORIA, and GTAP in Figure S1. The Pearson correlation coefficients (R) displayed in each plot provide a quantitative measure of the agreement between the estimates, where values closer to 1 indicate stronger correlations. The comparison between EDGAR and GLORIA datasets shows the highest Pearson R value of 1.000, indicating a strong correlation. This suggests that the GLORIA PBE are highly consistent with EDGAR data. In contrast, when comparing GTAP data with GLORIA and EDGAR, they exhibit a little lower Pearson R value of 0.934 and 0.936, respectively. These values still represent a high correlation but suggest a little greater uncertainty and variability in the GTAP data compared to EDGAR and GLORIA. The increased scatter of points around the 1:1 line in the middle and right plots, particularly in the mid-range values, also indicates more pronounced deviations between GTAP and the other two estimates.

Overall, Figure S1 demonstrates that GLORIA's PBE are of high quality, showing excellent consistency with the trusted EDGAR dataset. This highlights the reliability of using GLORIA as a robust source for PBE intensities in accounting for countries' methane CBE in this study.

Figure S1 Uncertainties of production-based methane emissions (PBE) in 2014. Each dot shows the total emission of a country.

Figure S2 compares our CBE (using GLORIA satellite emission data and MIRO table, on the x-axis) with alternative estimates from various sources including EDGAR, GTAP and different configurations of GLORIA (on the y-axis). Overall, Figure S2 suggests that our CBE are consistent with other estimates using alternative sources of PBE intensity and

MRIO tables. The Pearson R value ranges from 0.952 to 0.998 for different comparisons.

The three plots in the upper row of Figure S2 use the GLORIA MRIO table with different sources of PBE intensity, while the three plots in the bottom row use the GTAP MRIO table. Comparing the plots horizontally, we find that this study's estimates are highly consistent with those using GLORIA and EDGAR PBE intensities (with R values ranging from 0.993 to 0.998). Estimates using GTAP PBE intensities show relatively higher discrepancies with our estimates, with a noticeable reduction in the correlation (e.g., $R = 0.952$ and $R = 0.967$).

Using a different MRIO table will increase the discrepancy in CBE estimates, but to a lesser extent than using alternative PBE intensities. For example, using the GTAP MRIO table reduces the R value to 0.993, while using alternative PBE intensities reduces the R value to as low as 0.967, indicating that PBE intensities carry greater uncertainty than MRIO tables.

Figure S2 Uncertainties of consumption-based methane emissions (CBE) in 2014

Figure S3 further displays the time-series CBE for top countries, estimated from alternative data sources. The discrepancies between sources provide insights into the uncertainties associated with different datasets over the observation period. For most countries, such as China, India, Indonesia, Brazil, and Japan, the estimates using EDGAR PBE intensity (shown in red lines) tend to be slightly higher than those from GLORIA (blue lines). The inclusion of additional estimates from GTAP PBE intensity and MRIO tables (both using GLORIA and GTAP MRIO table configurations) adds further uncertainties, shown in dots. Despite these differences, the overall trend in emissions over time is consistent between the sources, suggesting that while there are uncertainties in the absolute values, the directional changes in emissions are comparable, indicating that our estimates and analysis of countries' emission trends in the main text are robust.

Figure S3 Range of methane emission footprints of top 9 countries from 1990 to 2023

While the main paper only requires minimal methodological information there are a number of key details and definitions which should be included to ensure that a general scientific audience can understand the paper, specifically:

- A clear definition of a production-based emission vs. a consumption based emission.
- A brief (one sentence) description of how the GLORIA model works and what input data it relies on (UNFCCC? EDGAR? Country specific economic data?).
- A map showing which countries are in which region colour coded to match Fig. 1.
- A definition of PBE intensity.

RE: Thanks for your suggestions. We have added more details in the abovementioned points in the revised manuscript, for better understanding.

A definition of production-based and consumption-based emissions in the introduction

Production-based emissions (PBE) refer to emissions emitted within a specific country or region during the production of goods and services, encompassing all economic activities and sectors of a country (to answer where emissions arise). Meanwhile, the consumption-based emissions (CBE) allocate the emissions to the countries where the goods are consumed (to answer the question of why we have these emissions or the

Editorial Note: The basemap layer of the map on page 21 of this peer review file is derived from Runfola, D. et al. geoBoundaries: A global database of political administrative boundaries. PLoS one 15, e0231866 (2020), published under the CC BY 4.0 license (<https://creativecommons.org/licenses/by/4.0/>).

ultimate drivers of emissions). It offers a different insight into understanding countries' emissions along the entire global supply chain. It is essential to consider both PBE and CBE when designing policies to reduce emissions to prevent emission leakage and the outsourcing of emission-intensive production to other locations²⁴, which will result in a misleading decoupling of PBE from economic growth.

A description of GLORIA model and its input data in the method:

In this study, we use the GLORIA dataset¹ which is a global MRIO dataset that integrates global economic flows with environmental data to assess the impacts of consumption and production patterns across industries and regions, relying on a range of input data including sectoral economic accounts, international trade statistics, and environmental resource use data (e.g., energy, water, materials). GLORIA includes 120 sectors, 164 countries/regions, and four final demand categories (i.e., household final consumption, non-profit institutions serving households, government final consumption, gross fixed capital formation).

We collect methane PBE from the satellite (environmental) accounts of GLORIA¹, which provides emissions data by countries and products consistent with the GLORIA MRIO table. The emission data includes both biogenic methane, produced and released from living plants and animals, and emissions from fossil fuel use. GLORIA satellite accounts combine EDGAR, the Organization for Economic Cooperation and Development, and Eurostat emission datasets.

A map showing the five aggregated regions in the Supporting document:

A definition of PBE intensity in the first result section of emission trends:

Figure 1-b) and c) present the average PBE intensity (i.e., PBE per unit of Gross Domestic Product (GDP) in USD 2015 constant prices) and per capita CBE of country groups.

One of the key drivers of decreasing emissions is specifically identified as “improvement in emission coefficient[s]” in both the Abstract and Drivers of consumption-based methane emissions sections. But there’s no discussion as to whether these emissions coefficients are “improving” because of changes in production methods which are reducing methane emissions (i.e. they changes reflect an actual improvement/reduction in methane emissions for a given sector/process) or whether the emissions coefficients are being updated to reflect

emissions from the given sector more accurately (i.e. processes within the given sector have not changed just our estimate of the emissions). If it's the later than this isn't a true reduction in emissions merely a methodological artefact. If it's the former than more detail as to exactly which processes and which sectors would be of interest to the reader. This is touched on briefly in the final paragraph of the Conclusions but needs to be explored in further detail in the main body of the paper and the abstract.

RE: Thanks. This driver reflects changes in emission technology due to shifts in production methods over the observation period, rather than simple updates of parameters to reflect more accurate emissions. Therefore, in the revised manuscript, we provide more discussion of the driver, including examples of changes in the coefficient of various sectors.

Changes in emission coefficient (i.e., emissions per unit of output) are the main determinant to reduce emissions over the period, offsetting the increasing effects from demand growth and structure changes, which is consistent with previous studies²⁻⁴. This driver reflects improvement in the emission technology during production which are reducing methane emissions. Evidence shows that global average emission coefficient decreased by 66.7% during 1998 to 2023. Such decreases in emission coefficient have been observed across sectors. For example, between 2018 and 2023, 97 out of 120 sectors reduced their emission coefficient. In particular, three sectors with the highest emission coefficient—raising of animals and agricultural services; water collection, treatment, and supply, sewerage; and raising of cattle—reduced their emission coefficient by 42%, 32%, and 20%, respectively, during this period.

The language used is at times overly wordy, confusing and ambiguous. E.g. Brazil, India and Russia are described as “the top three emitting countries of fertilizers”. Does this mean that these countries have the highest methane emissions due to fertilizer production (or consumption?) or is it that they produce the most fertilizer? Similarly, methane is described as “the world’s second largest greenhouse gas emission source”. Is this perhaps meant to be the second most important contributor to radiative forcing? Specific suggestions for improvements are given below.

RE: These three countries have the highest emissions from fertilizer production. This sentence has been revised based on your and other reviewers' comments. Additionally, we have revised the sentence about methane being the second most important contributor and have carefully examined the entire manuscript.

Some acronyms are not defined while other are defined and then not used.

RE: We have carefully addressed the acronyms and corrected all of them. The acronyms are defined when they appeared for the first time. We also removed the acronyms that are not used.

Units – emissions are an amount per time. E.g. t/yr. Please correct throughout. Also, the text discusses emissions in terms of Gt CO₂-eq while the figures are in million tons, tons and kilotons of methane (I think – it is unclear). Please be consistent and clear to allow easier comparisons.

RE: Thanks for your comment. We use consistent units across the text and figures. We use million tons for methane emissions, use ton/million USD for emission intensity, use tons for per capita emissions, as described in Figure 1. The unit "Gt CO₂-eq" is used only in addition to "million tons of methane" by converting to the CO₂ equivalent value. The text also provides the value in million tons, which is consistent with the figure. For example:

global methane emissions have increased slightly from 266.4 million tons/yr (equivalent to 7.5 gigatons of CO₂ emissions/yr, Gt CO₂-eq/yr) in 1990 to 292.3 million tons/yr (8.2 Gt CO₂-eq/yr) in 2002⁸, with an annual growth rate of 0.8%/yr.

We agree that emissions are an amount per time. We have changed the units to million tons/yr for methane emissions, tons/yr for per capita emissions, and tons/million USD/yr for emission intensity.

Figure descriptions are lacking in key details.

RE: We have added more details to the figure descriptions to aid in understanding the figures. The information conveyed by the figures is described in the text.

Figure 1 Methane emissions by income country groups and top ten countries.

*Note: Sub-figures a) shows PBE (in solid lines) and CBE (in dashed lines) of country groups during 1990-2023; b) and c) show PBE intensity (i.e., PBE per unit of GDP) and per capita CBE of country groups, respectively; and d) shows sectoral PBE in top ten countries in 2023. Data source: the PBE and CBE of countries used in sub-figure a) are calculated based on GLORIA satellite data (environmental accounts) and MRIO tables¹; PBE intensity and per capita CBE in sub-figure b) are calculated with World Bank dataset; the sectoral emissions in sub-figure d) are collected from EDGAR⁸. Detailed PBE and CBE of countries and aggregated groups are provided in **Error! Reference source not found.**. A global map for the five aggregated regions is shown in the Supplementary document.*

Figure 2 Decoupling of methane emissions and economic growth in selected countries at ten-year intervals

*Note: Sub-figures a) and b) show the relationship between PBE, CBE and GDP during 1993-2023 at ten-year intervals, the coloured dots present five country groups, and the coloured stars present selected seven countries, the grey dots present the remaining countries. Sub-figures c) and d) show the decoupling results of countries/groups in coordinate systems. Delta GDP and emissions refers to the changes in GDP and emissions over the given period. Detailed results of decoupling are shown in **Error! Reference source not found.***

Figure 3 Emissions embodied in trade 1998 and 2023

Note: The figure includes the emissions flow from one country to another (i.e., emissions produced in one country but consumed in another country), and exclude the emissions consumed in the same country where they are produced. The country groups on the left y-axis are producers while country groups on the right y-axis are consumers. The flows between the same country group present emissions embodied in trade between different countries within the group.

Figure 4 Sectoral emissions embodied in export of countries 1998 and 2023

Note: sectors distributions in the figure are aggregated based on GLORIA sectors according to their emissions.

Figure 5 Emission drivers of consumption-based methane emissions at five-year intervals

Note: sub-figure a) shows the contributing effects of each driver to global emissions in absolute values, sub-figures b) to f) shows contributing effects of each driver to five country groups in percentages relative to the group's total emissions.

Specific comments

Abstract

“It also discusses the status of decoupling from economic growth for both methane emission accounts.” This is confusing. The decoupling of what from economic growth? What methane emission accounts? Do you mean “It also discusses the status of decoupling both production and consumption driven methane emissions from economic growth.”?

RE: Done.

It also discusses the status of decoupling of production- and consumption-based methane emissions from economic growth.

Please change “Developed countries are the only group that has managed to reduce both production- and consumption-based emissions...” to “Only developed countries have managed to reduce both production- and consumption-based emissions ...”

RE: Done.

Please change “It identifies that the improvement in emission coefficients is the main determinant for reducing emissions...” to “It identifies that changes in emission coefficients, reflecting [insert reason here] are the main determinant for reducing emissions...”

RE: Done.

It identifies that the reduction in emission coefficient, reflecting improvements in emission technology, is the main determinant for reducing emissions over the observation period and can offset the increasing effects from growth of final demand.

Introduction

In light of the recent global methane pledge please change “...CO₂, which is the main focus...” to “...CO₂, which has been the main focus...”.

RE: Done.

Please change “Global CH₄ emissions have been rising for decades and since 2001 the annual growth rate reached 1.22% on average, which was slightly below the growth rate of CO₂ emissions (1.80%)” to “Global CH₄ emissions have been rising for decades, with the average (since 2001) annual growth rate reaching 1.22%, slightly below the growth rate of CO₂ emissions (1.80%)”

RE: Done.

Please change “... specific global studies on methane are still scant...” to “... specific global studies on methane are not common...”. Considering the recent GCP methane budgets (e.g. Saunois et al. 2020) this seems a fairer assessment.

RE: Done.

“Some emission datasets include countries’ CH₄ emissions as a part of GHG emissions (Saunois et al., 2020; He et al., 2020; European Commission, 2024; Jones et al., 2023) ...” Of the studies given in these references two are focuses solely on methane while the other two while investigating multiple GHG’s do consider methane as distinct from other GHGs. In contrast, the sentence suggests that that methane emissions are provided as a mixed component of total GHGs emissions. Secondly the sentence implies that each of these studies outline a dataset. This is not true. While references 6 and 13 outline datasets the Saunois et al. paper (reference 11) pulls together multiple existing datasets (including that described in ref 6) and compares them and He et al. (reference 12) uses an optimisation (top-down) approach to match emissions transported via a global chemical transport model to global

methane observations. It relies on the CEDS and SSP2. Please rewrite this sentence and include further examples of bottom-up CH4 emission datasets (e.g. CEDS).

RE: Thanks for the suggestion. We have revised the text in the literature review.

Existing emission datasets and studies have presented long-term methane trends^{8,36-39}. He, et al.⁴⁰ employed a top-down approach to match methane emissions transported through a global chemical transport model with observation data. Saunio, et al.³⁶ combined multiple existing datasets to compare methane estimated derived from both bottom-up and top-down approaches. Some of the previous studies further looked into methane emission patterns of selected countries (e.g., China⁴¹, US⁴², UK⁴³) and several major sectors, including agriculture activities⁴⁴, such as livestock⁴⁵, and food system⁴⁶; wastewater treatment⁴⁷⁻⁴⁹ and waste management in landfills²⁰; energy sectors such as oil and gas production⁵⁰, coal mining⁵¹. Sun, et al.⁵² and Ari and Şentürk⁵³ also discussed the decoupled relationship between methane emissions and economic growth.

“Most of the existing studies focused on selected countries and a small number of sectors such as agricultural outputs in EU countries...” Does this refer only to studies of decoupling? If it refers to studies of methane emissions in general than I’d disagree e.g. the papers listed in above take a global approach. I’d also suggest Stavert et al. (Global Change Biology, 28, 182-200, 10.1111/gcb.15901, 2022).

RE: We have revised the literature review. Please refer to our response above.

“... lack comprehensive coverage of a wide range of countries and high sectoral resolution...” followed by “...accounted for consumption-based CH4 emissions of 26 sectors in 181 economies...”. 26 sectors and 181 economies doesn’t sound lacking in comprehensive coverage. Perhaps reword? E.g. “However, none of these studies were conducted in the last decade and many lack a comprehensive coverage of a wide range of countries and high sectoral resolution. For example, Zhang et al. (2018) while having a broad sectoral and economic focus (26 sectors in 181 economies) had only a narrow temporal range (2000 – 2012) ...”

RE: Done.

However, none of these studies were conducted in the last decade and many lack comprehensive coverage of a wide range of countries and high sectoral resolution. For example, Zhang, et al.⁵⁴ accounted for methane CBE using the Eora Multi-Region Input-Output (MRIO) database⁵⁵ and Emissions Database for Global Atmospheric Research (EDGAR). While their analysis covered a wide range of sectors and economies (26 sectors across 181 economies), it was limited to a narrow temporal scope, spanning from 2000 to 2012.

Please change “...to account for the most up-to-date time-series (1990 to 2023) consumption-based methane emissions of 164 countries/regions, which account for 98% of global methane emissions” to “...to examine consumption-based methane emissions of 164 countries/regions (accounting for 98% of global methane emissions) for the period 1990-2023.”

RE: Done.

Please expand on how the GLORIA model works and what input data is used. Also what are MRIO tables? How were they used?

RE: We introduced all the methods and input data in the Methods section. There was not enough space in the introduction for this technical information, so we added a sentence to guide readers to the Methods section.

More details about the input-output model, decoupling analysis, SDA, and input data are described in the methods section.

I would also suggest including here a sentence on where the production emissions estimates are coming from.

RE: Same as above. We also added a very brief introduction to the data source in the note of figure 1.

*Data source: the PBE and CBE of countries used in sub-figure a) are calculated based on GLORIA satellite data (environmental accounts) and MRIO tables ¹; PBE intensity and per capita CBE in sub-figure b) are calculated with World Bank dataset; the sectoral emissions in sub-figure d) are collected from EDGAR ⁸. Detailed PBE and CBE of countries and aggregated groups are provided in **Error! Reference source not found.** A global map for the five aggregated regions is shown in the Supplementary document.*

Please change “..discussion on..” to “...discussion of...”.

RE: Done.

Methane emission trends

“Global methane emissions...” are these production or consumption emissions? Or both? Where have they come from? Are you referring to those from this study or earlier studies? Are these annual emissions? Please specify.

RE: On the global scale, production-based emissions equal consumption-based emissions. So here we say ‘global methane emissions’. These values refer to annual emissions, we have changed the units to million tons/yr for better understanding. All these emissions are calculated in this study, as described in the note of figure 1. We have revised the text as well.

In the text:

Based on our calculation, global methane emissions have increased slightly from 266.4 million tons/yr (equivalent to 7.5 gigatons of CO₂ emissions/yr, Gt CO₂-eq/yr) in 1990 to 292.3 million tons/yr (8.2 Gt CO₂-eq/yr) in 2002 ⁸, with an annual growth rate of 0.8%/yr.

In the note of figure 1:

*Data source: the PBE and CBE of countries used in sub-figure a) are calculated based on GLORIA satellite data (environmental accounts) and MRIO tables ¹; PBE intensity and per capita CBE in sub-figure b) are calculated with World Bank dataset; the sectoral emissions in sub-figure d) are collected from EDGAR ⁸. Detailed PBE and CBE of countries and aggregated groups are provided in **Error! Reference source not found.** A global map for the five aggregated regions is shown in the Supplementary document.*

Please change “... at an annual growth rate...” to “... with an annual growth rate...”.

RE: Done.

Please change “Since 2002, global emissions increased at a faster rate of 2.0%/yr and reached 329.5 million tons (9.2 Gt CO₂-eq) in 2008. One major reason is that China joined the World Trade Organization (WTO) in 2002, and its emissions surged due to increasing production for both domestic use and exports. The increased emissions in China from 2002 to 2008 (12.1 million tons) accounted for 32.5% of the global total increment (37.2 million tons).” To “Since 2002, global emissions increased at the faster rate of 2.0%/yr reaching 329.5 million tons (9.2 Gt CO₂-eq) in 2008. Following its joining of the World Trade Organization (WTO) in 2002

Chinese methane emissions surged, with its emissions increasing by 12.1 million tons between 2002 to 2008 accounting for 32.5% of the global total increment (37.2 million tons).”

RE: We have revised this sentence based on you and other reviewers’ comments.

32.5% of the global total increment (37.2 million tons) over the period can be attributed to China, whose emissions increased by 12.1 million tons due to increasing production for both domestic use and exports.

Please change “After a decline in 2009 due to the global financial crisis, global methane emissions resumed its growth after 2009 at 1.1%/yr and finally reached 383.5 million tons (10.7 Gt CO₂-eq) in 2023. Unfortunately, the momentum of global methane emission growth shows no sign of slowing down, which contrasts with the recent slowdown observed in CO₂ emission growth” to “After a decline in 2009 due to the global financial crisis, global methane emission growth resumed at 1.1%/yr reaching 383.5 million tons (10.7 Gt CO₂-eq) in 2023. Unfortunately, the momentum of global methane emission growth shows no sign of slowing down, contrasting with the recent slowdown observed in CO₂ emission growth”

RE: Done.

Please change “The gaps between countries’ PBE and CBE are known as net-emissions embodied in trade.” to “The gap between a country’s PBE and CBE is known as its net-emissions embodied in trade.”

RE: Done.

When comparing a country’s PBE and CBE, the gap between them is known as net-emissions embodied in trade.

“Asia and Developing Pacific has seen a change in the role from a net exporter to importer between 2010 to 2014.” The lines shown in Figure 1 for this region are very close together. Considering that there is no uncertainty estimate given how confident are you that this statement is true?

RE: We compare the emissions of Asia and Developing Pacific countries with our newly added alternative CBE and PBE accounts in the uncertainty analysis. The results are shown in the figure below.

We find that the time-series results calculated using EDGAR data, the GLORIA satellite account, and the GLORIA MRIO table exhibit similar trends (sub-figures a and b). The CBE and PBE of this group are very close during 2009-2014 (sub-figure a) and 2011-2014 (sub-figure b). After 2014, the CBE surpasses the PBE significantly. Although other estimates using GTAP data (sub-figures c-f) still show that CBE is less than PBE, we observe that the gap between CBE and PBE is narrowing, indicating a trend similar to sub-figures a and b. Additionally, estimates using GTAP data only go up to 2014. We believe that after 2014, the CBE of this country group will surpass their PBE, following the observed trends. Therefore, we are confident in this statement.

“Figure 1-b) and c) present the average PBE intensity and per capita CBE of country groups.” Please define intensity here. I’m assuming it is annual PBE divided by gross domestic product for the corresponding year in a specific currency? But it needs to be defined.

RE: Done.

Figure 1-b) and c) present the average PBE intensity (i.e., PBE per unit of Gross Domestic Product (GDP) in USD 2015 constant prices) and per capita CBE of country groups.

Please change “Per capita CBE has the opposite trend with emission intensity in most country groups.” to “Per capita CBE shows the opposite trend to emission intensity in most country groups.”

RE: Done.

Decoupling of methane emissions and economic growth

Please change “In the last ten years from 2013 to 2023, there are 55 (33.5% of 164) countries that have reduced their PBE, ...” to “Between 2013 to 2023, 55 (33.5% of 164) reduced their PBE, ...”

RE: Done.

Please change “...are developed ones at high economic levels...” to “...are developed at high economic levels...”

RE: Done.

Please change "...PBE and CBE decline while GDP kept growing." to "...PBE and CBE decline with GDP growth." Note also that this is the first use of GDP and it has not be defined earlier.

RE: Done. GDP has been defined earlier in the first result section.

Please change "...show the results for the most recent ten years (2013-2023)." to "...show the results for the period 2013-2023."

RE: Done.

I'd suggest including in this paragraph a fuller description of the decoupling index. Perhaps summarise the "A degree of strong decoupling ($DI \geq 1$ & $[[GDP]]_1 \geq [[GDP]]_0$) implies that the country's emissions decrease but the GDP keeps growing, weak decoupling ($0 \leq DI < 1$ & $[[GDP]]_1 \geq [[GDP]]_0$) has both emissions and GDP increasing, but GDP is growing faster than emissions, while no decoupling ($DI < 0$ & $[[GDP]]_1 \geq [[GDP]]_0$) means the country's emissions are growing faster than its GDP." sentence from the methods section?

RE: Done.

We use the Tapio Decoupling Index to describe the dynamic relationship between emissions (both PBE and CBE) and economic growth (i.e., GDP) in each country and aggregated group during the years 2013 to 2023. Strong decoupling occurs when a country's emissions decrease while its GDP continues to grow. Weak decoupling happens when both emissions and GDP increase, but GDP grows at a faster rate than emissions. In contrast, no decoupling indicates that emissions are rising at a faster rate than GDP.

"In contrast, the PBE intensities of strongly decoupled countries are significantly lower than those of weakly or non-decoupled countries (p-value = 0.000), mirroring the pattern seen in CO₂ emissions (Hubacek et al., 2021). This indicates that economic growth in weakly/non-decoupled countries is heavily driven by methane-intensive sectors such as agriculture and energy mining" How does this indicate this? What does the sectoral break down of these countries show?

RE: Weakly/non-decoupled countries have higher sectoral PBE intensity than strongly decoupled countries. We have revised the text as follows.

In contrast, the PBE intensities of strongly decoupled countries are significantly lower than those of weakly or non-decoupled countries (p-value = 0.000), mirroring the pattern seen in CO₂ emissions⁴³. Among the top 10 emission-intensive sectors (e.g., agriculture, energy, mining, and wastewater treatment), the average sectoral emission intensity in weakly/non-decoupled countries is 35% to 460% higher across nine sectors than in strongly decoupled countries.

Emissions embodied in trade and outsourcing

"In 2023, developing countries are the..." should this be "In 2023, developed countries are the..." ? That seems to be what Figure 3 is showing.

RE: Corrected.

Drivers of consumption-based methane emissions

"Structural Decomposition Analysis on global emissions (as shown in Figure 5-a) and on each country group (as shown in Figure 5-b to f) at five-year intervals from 1998 to 2023." Which does what? Assigns a contribution to emissions for five potential drivers?

RE: Done.

We employed a Structural Decomposition Analysis (SDA) to investigate the contributions of five drivers behind the changes in global emissions (as shown in Figure 5-a) and each country group (as shown in Figure 5-b to f) at five-year intervals from 1998 to 2023.

“...while improvements in emission coefficients are the sole determinants of reduced emissions over the period...” please refer to earlier comments. This needs to be further explored. What exactly is driving changes in the emissions coefficients? Mitigation efforts? Which sectors?

RE: This driver refers to improvement in emission technology during production. We have revised the text to make it clear. We also added some examples to show the changes in emission coefficients in sectors. Potential efforts to reduce emission coefficients is discussed in the conclusion section.

Changes in emission coefficient (i.e., emissions per unit of output) are the main determinant to reduce emissions over the period, offsetting the increasing effects from demand growth and structure changes, which is consistent with previous studies²⁻⁴. This driver reflects improvement in the emission technology during production which are reducing methane emissions. Evidence shows that global average emission coefficient decreased by 66.7% during 1998 to 2023. Such decreases in emission coefficient have been observed across sectors. For example, between 2018 and 2023, 97 out of 120 sectors reduced their emission coefficient. In particular, three sectors with the highest emission coefficient—raising of animals and agricultural services; water collection, treatment, and supply, sewerage; and raising of cattle—reduced their emission coefficient by 42%, 32%, and 20%, respectively, during this period.

“Improved production technology increased global methane emissions by 56.7 million tons (or 90%) from 2003 to 2018. However, there has been a reversal, leading to decreasing emissions in the last five years (2018-2023).” Considering the focus of the paper (methane emissions) one would assume that “improved production” = less methane emissions so perhaps rephrase this? I’d suggest “Changes production technology between 2003 and 2018 increased global methane emissions by 56.7 million tons (or 90%), however, recent improvements have led to decreasing emissions in the last five years (2018-2023).”

RE: Done.

Please change “The results show that population has the least degree of dispersion compared to the other four drivers, with a Standard Deviation of 0.04. The Standard Deviation of contributing effects of emission coefficient to total emission in each country are 0.54...” to “The results show that population has the least degree of dispersion compared to the other four drivers, with a SD of 0.04. The SD of contributing effects of emission coefficient to total emission in each country are 0.54...”

RE: Done.

Conclusions

“Methane is the world’s second largest greenhouse gas emission source, contributing about 30% of global warming.” Methane is not an emission source. Please re-word.

RE: We have removed this sentence to keep the text short.

Please change “This study employes an environmental...” to “This study employs an environmental...”

RE: Done.

Please change "...to analyze the most up-to-date emission changes..." to "...to analyze up-to-date emission changes..."

RE: Done.

Please reword "It is essential to recognize that, although other emerging countries currently emit lower levels of methane..." This reads as referring to USA and China as emerging countries.

RE: Done.

It is essential to recognize that, although emerging and small countries currently emit lower levels of methane, they may become key methane emitters in the future due to economic growth.

"...they may become significant methane emitters in the future due to growth of agriculture and mining sectors..." Surely that's dependent on the natural resources available to each country?

RE: Agreed and corrected. Please see our response above.

"When factoring in the dimension of economic growth, our decoupling analysis reveals that developed countries stand out as the sole group that has successfully achieved strong decoupling between emissions and GDP." This is good news but how did they do it? Which sectors decreased? What policy/economic/technological mechanisms were used? This is the most interesting bit please expand.

RE: This is mainly caused by a decline in emission intensities across most sectors, which can be linked to the analysis of drivers below. We have added evidence at the sector level and revised the text.

We find that developed countries stand out as the sole group that has consistently achieved emission reductions and strong decoupling between emissions (both PBE and CBE) and GDP. Their decoupling is mainly caused by a decline in their emission coefficient, and to a lesser degree due to outsourcing methane emissions to less developed regions. Among the 120 sectors classified in the GLORIA configuration in developed countries, 100 of them reduced their emission coefficient between 2003 and 2013. This is evident from the analysis of the drivers, where emission coefficient is the primary determinant in reducing emissions.

"Our primary objective is to reduce emissions while sustaining economic growth, essentially striving for sustainable transitions" Our? Our = the authors? No, the individual countries.

RE: Done. It has been changed to 'Countries'.

"Brazil (15.7% of the global total import), India (11.9%), and the US (9.3%) are the top three emitting countries of fertilizers." Do you mean emitters of fertilisers or emitters of methane due to fertiliser production?

RE: The numbers refer to import value. This paragraph has been revised substantially according to reviewers' comments. This sentence has been deleted.

Please change "... more and more trade restrictions..." to "... additional trade restrictions..."

RE: Done.

Please change "The EU and US may need..." to "As such, the EU and US may need..."

RE: Done.

Please change “This indicates that Russia has relatively advanced production technology and efficiency for fertilizer production.” to “This suggests that Russia has relatively advanced production technology and efficiency for fertilizer production minimizing associated methane emissions.”

RE: Done.

Please change “Mechanism (CBAM), that confines the import of high-emitting products...” to “Mechanism (CBAM), that aims to minimise the import of high-emitting products...” or “Mechanism (CBAM), that aims to reduce the import of high-emitting products...”

RE: Done.

Please change “CBAM includes fertilizers, which are one of the top methane sources,...” to “CBAM includes fertilizers, the production of which is one of the largest methane sources,...”

RE: Done.

“Although CBAM can effectively reduce consumption-based emissions in the EU by importing from cleaner-producing countries, it could be a potential trade barrier that obstructs globalization.” As the focus of the paper is the reduction of methane emissions, I would suggest either removing this sentence or linking changes in the level of globalization to methane emissions.

RE: OK. We removed this sentence.

“...improvements in emission coefficients...” please refer to earlier comments.

RE: Done.

“One possible explanation is that an upturn in living standards has led to increased consumption of red meat (e.g., beef and lamb), which is very methane-intensive.” Is this indicated by changes in the sectoral emissions distribution for these countries? You have the data to explore this.

RE: Done. We have moved the text from conclusion to the result section “Drivers of consumption-based methane emissions”.

We find that changes in structure of final demand play an important role in the increase of emissions, particularly after 2008. Between 2008-2013 and 2018-2023, this factor contributed to a staggering 155 and 234 million tons (or 47% and 64%) of emission growth, respectively. This suggests that there is an increase in the consumption of methane-intensive products or a rise in the import of goods from countries with higher methane emission intensity, attributed to shifts in the global trade structure. For example, the share of red meats (beef, lamb, and pork), which are methane-intensive products ^{46,56}, in total global demand has increased from 0.6% to 0.8% between 2008 and 2023.

“...it is imperative to allocate sufficient room for methane in the future carbon budget...” I’m not sure what the authors mean by this. Are they suggesting incorporating methane into a global carbon budget? A combined global GHG budget? Or are they suggesting focusing policy decisions/political effort etc. on reducing methane emissions? In which case recent efforts e.g. <https://www.globalmethanepledge.org> are already working that direction.

RE: We removed this sentence as it is not the focus of this paper.

Figures

Figure 1 – Please add an extra panel or figure or table where the composition of the regions (ie. which country is where) is defined. In the supplement would be sufficient. Please correct

the units. Emissions are amount per unit of time. E.g. million t/yr. The SI symbol for ton is t. Please use it consistently.

RE: We have added a map in the SI to show the regions. We also corrected the units in figure 1.

Figure 2 – What are the grey dots? I assume they are the individual countries but this is not specified. What is “delta CBE” and “delta PBE”? These need to be defined in the text or here in the caption.

RE: We have revised the figure caption to explain these details.

Figure 6 Decoupling of methane emissions and economic growth in selected countries at ten-year intervals

*Note: Sub-figures a) and b) show the relationship between PBE, CBE and GDP during 1993-2023 at ten-year intervals, the coloured dots present five country groups, and the coloured stars present selected seven countries, the grey dots present the remaining countries. Sub-figures c) and d) show the decoupling results of countries/groups in coordinate systems. Delta GDP and emissions refers to the changes in GDP and emissions over the given period. Detailed results of decoupling are shown in **Error! Reference source not found.***

Figure 4 – Are these sectoral distributions, like Fig 1, also from EDGAR?

RE: These sectoral distributions are not from EDGAR. They are aggregated from GLORIA sectors. We have added a note to explain this.

Figure 7 Sectoral emissions embodied in export of countries 1998 and 2023

Note: sectors distributions in the figure are aggregated based on GLORIA sectors according to their emissions.

Figure 5 – for b to f percentage of what? Is it contribution to emissions change? More information needed either here or in the text.

RE: We have added explanations in the notes of figure 5.

Figure 8 Emission drivers of consumption-based methane emissions at five-year intervals

Note: sub-figure a) shows the contributing effects of each driver to global emissions in absolute values, sub-figures b) to f) shows contributing effects of each driver to five country groups in percentages relative to the group's total emissions.

Reviewer #3

This paper uses multiregional input-output analysis to calculate methane emissions embodied in production and consumption using data from 1990_2023 for a global dataset (GLORIA, 164 countries/regions and a high disaggregation at the sectoral level). The authors calculate the two inventories (i.e. based on production and consumption) and describe some stylized facts_i.e. methane trends and analysis of methane embodied in trade, decoupling based on the Tapio index, and structural decomposition analysis to study the drivers of methane consumption inventories.

Although the topic of research is an interesting topic which has relevance for climate change, I have concerns about some aspects of the paper - particularly, (i) about the novelty and value added of the paper, (ii) about the methodology used, and (iii) about some policy recommendations and conclusions made by the authors.

RE: Thanks for your comments. We have revised the manuscript and response point-by-point below.

Novelty of the paper

The exercise implemented by the authors, the calculation of MRIO-based consumption inventories of methane emissions and the descriptive analysis of methane emissions embodied in trade flows, as well as the decoupling and the structural analysis are not new, even though the authors claim novelty. The authors do not cite recent literature that has dealt with the same questions, and which provided calculations using similar methodologies and regression-based analysis. These articles can be easily found in any browser or Google Scholar.

RE: Thank you for your comments. We acknowledge that similar studies have been published previously; however, as Reviewer 2 mentioned, "Considering that earlier studies of this type are now over a decade old, this will be a topic of interest to many and useful to the wider scientific community", our study still offers significant novelty by covering 1) a longer time series (1990–2023); 2) more countries (164); and 3) more sectors (120) than previous studies. This improvement will enhance the accuracy of accounting for consumption-based emissions and provide detailed sectoral insights for a greater number of countries.

We have reiterated our study's novelty at the end of the introduction.

Aiming to address the research gaps concerning the lack of up-to-date analysis on methane emissions from the perspective of consumption, covering a wide range of countries and with high sectoral resolution, this study uses the latest GLORIA input-output dataset (Global Resource Input-Output Assessment)¹ to examine methane CBE of 164 countries/regions (accounting for 98% of global methane emissions) for the long-term period 1990-2023⁸.

Moreover, in this revised manuscript, we have updated the literature review, cited and discussed additional relevant studies, compared our results with previous findings, and included more results at the sectoral level. For example:

Updated literature review in the Introduction section:

Existing emission datasets and studies have presented long-term methane trends^{8,36-39}. He, et al.⁴⁰ employed a top-down approach to match methane emissions transported through a global chemical transport model with observation data. Saunio, et al.³⁶ combined multiple existing datasets to compare methane estimated derived from both bottom-up and top-down approaches. Some of the previous studies further looked into methane emission patterns of selected countries (e.g., China⁴¹, US⁴², UK⁴³) and several major sectors, including agriculture activities⁴⁴, such as livestock⁴⁵, and food system⁴⁶; wastewater treatment⁴⁷⁻⁴⁹ and waste management in landfills

²⁰; energy sectors such as oil and gas production ⁵⁰, coal mining ⁵¹. Sun, et al. ⁵² and Ari and Şentürk ⁵³ also discussed the decoupled relationship between methane emissions and economic growth.

However, these studies focused on production-based methane emissions and ignored consumption-based emissions. Production-based emissions (PBE) refer to emissions emitted within a specific country or region during the production of goods and services, encompassing all economic activities and sectors of a country (to answer where emissions arise). Meanwhile, the consumption-based emissions (CBE) allocate the emissions to the countries where the goods are consumed (to answer the question of why we have these emissions or the ultimate drivers of emissions). It offers a different insight into understanding countries' emissions along the entire global supply chain. It is essential to consider both PBE and CBE when designing policies to reduce emissions to prevent emission leakage and the outsourcing of emission-intensive production to other locations ²⁴, which will result in a misleading decoupling of PBE from economic growth.

Previous studies have employed the input-output approach to estimate methane CBE, emissions embodied in global supply chains, as well as their drivers. However, none of these studies were conducted in the last decade and many lack comprehensive coverage of a wide range of countries and high sectoral resolution. For example, Zhang, et al. ⁵⁴ accounted for methane CBE using the Eora Multi-Region Input-Output (MRIO) database ⁵⁵ and Emissions Database for Global Atmospheric Research (EDGAR). While their analysis covered a wide range of sectors and economies (26 sectors across 181 economies), it was limited to a narrow temporal scope, spanning from 2000 to 2012. Liu, et al. ⁵⁷ analyzed the trade embodied emissions of 56 sectors in 44 countries from 2000 to 2014 using the World Input-Output Dataset ⁵⁸. Fernández-Amador, et al. ³ analyzed the consumption-based methane emissions for 78 countries/regions in 1997, 2001, 2004, 2007, 2011 and 2014 using the Global Trade Analysis Project MRIO tables. Sun, et al. ² investigated the drivers of methane CBE in 43 countries from 2000 to 2014. Ma, et al. ⁵⁹ analyzed Chinese methane CBE and drivers for four years (2005, 2007, 2010 and 2012) and 20 sectors. Similar studies include Zhang, et al. ²³ and Wang, et al. ⁶⁰. In addition to the analysis of the whole economic system and supply chains, some studies have specifically focused on the Agriculture, Forestry and Other Land Use activities ^{61,62} and food production ⁶³, which constitute the major sources of methane emissions. Although sector-specific analyses offer detailed insights into emission patterns, they fall short of presenting the overall landscape of changes in emissions across the entire economic system influenced by human activities and interactions with other sectors.

We have added comparisons to previous studies in the results sections.

In the first result section of emission trends:

Compared to previous studies focusing on emission trends before 2014 ^{3,26}, our result further highlights a resurgence in methane emissions after 2015, particularly in Asia and rapidly developing countries, driven by ongoing economic and population growth. That is to say global methane emission have been growing faster in recent years ⁴¹, albeit with some fluctuations, which is in contrast with the recent slowdown observed in CO₂ emission growth ⁴².

When comparing a country's PBE and CBE, the gap between them is known as net-emissions embodied in trade. ... Our findings align with prior research ^{2,3,23}, underscoring the increasing complexity of methane emissions driven by international trade and changing patterns. More details will be discussed in the section below.

In the third result section of trade embodied emissions:

Previous studies have noted this trend in methane emissions, despite domestic reductions being achieved^{3,23}. Such outsourcing will largely increase the emissions from less developed regions with lagging levels of technology and lower production efficiency, which will lead to a possible rise in global emissions²⁵. This is especially the case for countries that have achieved strong decoupling of PBE and GDP, where no such decoupling is observed between CBE and GDP. For example, the PBE of New Zealand decreased by 2.8% from 2013 to 2023 while its CBE increased by 49.3%. Sun, et al.² also noted emissions transfer from developed countries to emerging countries in agriculture and food manufacturing sectors align with global trade.

In the fourth result section of emission drivers:

The global results indicate that economic growth, represented by final demand per capita in this study, has been identified as a key driver of increased emissions, which has been discussed previously³. In addition, we find that changes in structure of final demand play an important role in the increase of emissions, particularly after 2008.

Changes in emission coefficient (i.e., emissions per unit of output) are the main determinant to reduce emissions over the period, offsetting the increasing effects from demand growth and structure changes, which is consistent with previous studies²⁻⁴.

The analysis of methane production-, final production- and consumption-based inventories, as well as decomposition analysis (Kaya and LMDI decompositions) of those inventories has been carried out by Fernandez-Amador et al. (2020, Ecological Economics). This paper is not cited. This research also analyzes methane emissions embodied in trade. Although Fernandez-Amador et al.'s (2020) analysis goes until 2014, the authors have not made it clear their contribution with respect to that article. To my understanding, most of the findings are similar to those in that article.

RE: Thank you for your recommendations of this valuable literature. In the revised manuscript, we have cited and discussed this paper. We noticed that Fernandez-Amador et al. (2020, Ecological Economics) analysed the consumption-based emissions for 78 countries/regions in 1997, 2001, 2004, 2007, 2011 and 2014. Compared to Fernandez-Amador et al.'s study, we cover more countries (164) and consecutive time period 1990-2023. We have also compared the results from Fernandez-Amador et al. (2020, Ecological Economics) with our findings in the revised manuscript, as we mentioned in the previous response.

The authors implement a simple Tapio decoupling index to analyze decoupling between methane emissions and economic growth. Besides other methodological issues that I comment below, the authors again claim novelty in the analysis of decoupling. However, they miss two papers, Fernandez-Amador et al. (2018, Economics Letters) and Fernandez-Amador et al. (2022, Empirical Economics). Fernandez-Amador et al. (2018) present estimates of the income-elasticity of methane based on regression analysis, while Fernandez-Amador et al. (2022) present dynamic regressions of methane and analyze β -convergence trends. The authors cite Fernandez-Amador et al. (2020, Applied Economics), which implements regression analysis at the sectoral level to analyze the environmental Kuznets curve for methane emissions and estimates the income-elasticity of methane at the sectoral level, but do not compare the results with the ones in that article. Being the relation with that research so direct, this is a serious issue. Some of these references implement their analysis not only for emissions per capita but also for emissions intensity (emissions per value added) for production and consumption inventories.

Fernandez-Amador et al. have made a CEPR Voxeu column on the topic (see link).

RE: Thanks. We acknowledge the contributions of Fernandez-Amador et al. (2018, Economics Letters) and Fernandez-Amador et al. (2022, Empirical Economics), along with other papers by this author. Both studies focus on periods prior to 2014, while our analysis of decoupling covers the more recent decade from 2013 to 2023. Our study specifically examines the decoupling of total emissions and GDP, whereas the two cited studies analyse emissions per capita, emissions intensity, and income. However, due to the differing time periods, we are unable to incorporate many details from these studies into our paper. These studies have been referenced in the literature review section.

The authors should make a literature review and rethink the contribution of their research in relation to prior research. Moreover, they may show, for example, whether the high sectoral disaggregation of their dataset produces different results (i.e. whether other research suffers from aggregation bias).

RE: We rewrote the literature review paragraph and highlighted our contribution in comparison to prior studies. Please see our response to comments above.

Indeed, one of our contributions is the use of highly disaggregated sectoral datasets. The main benefit of conducting input-output research with high sectoral disaggregation is that it provides a more accurate accounting of consumption-based emissions, which has been verified in previous studies^{64,65}. In our revised manuscript, we have added more discussion of the results at the sectoral level. We also use the detailed sectoral information to explain some of the findings in the main text, which cannot be found in aggregated sectoral research. We published our sectoral CBE and trade embodied emission in the supplementary Table S1 and S3.

In section 'Methane emission trends':

To illustrate the sectoral distribution of methane emissions, we aggregated 120 sectors into seven categories, as shown in Figure 1-d), the underlying calculations are done at the 120-sector resolution. Globally, agriculture and food production are the largest contributors to methane emissions, accounting for 47% in 2023, followed by waste and recovery (17%), coal and electricity (14%), and oil and gas (13%). Over time, the proportions of sectoral emissions have shifted. Agriculture and food production decreased from 54% in 1990 to 47% in 2023, and oil and gas declined from 18% in 2006 to 13% in 2023. In contrast, emissions from coal and electricity production rose from 9% in 2000 to 14% in 2023, reflecting sectoral structure changes in global methane emissions. Countries have a huge difference in the sectoral structure of methane emissions, as shown in Figure 1-d), which is closely associated with the economic structure of the countries. For example, coal and electricity production, especially extraction of hard coal, is the largest emission source in China and Indonesia, contributing to 35% and 56% of their PBE, respectively. The two countries account for 59.3% of global coal production in 2023 and therefore emit a huge amount of methane from coal mining⁵. In contrast, Russia and Iran have higher emissions from oil and gas production (33% and 57% respectively). India and Brazil emit large shares of emissions from agriculture (67% and 73% respectively), e.g., pigs and cattle due to livestock farming.

In section 'Emissions embodied in trade and outsourcing':

When examining emissions flows by sectors, we observe considerable differences across countries. For example, in 2023, developed countries have the largest share of emissions embodied in their agricultural and food-related exports (57.6%), while the major sectors for Africa and the Middle East's emissions embodied in exports are oil and gas (27.2%). Agricultural and food products account for 20.5% of Africa and the

Middle East's exported emissions. Latin America and the Caribbean export substantial amounts of agricultural and food products, accounting for 73.4% of their exported embodied methane emissions. In contrast, Asia and the Developing Pacific region have 47.4% of their emissions embodied in exports associated with coal and electricity production. Figure 4 shows detailed trade embodied emissions by sectors in top-emitting countries.

*When we classify countries into the global North (i.e., developed countries) and South, we observe that emissions embodied in North-to-South trade primarily originate from agriculture (63.6% in 2023). In contrast, South-to-North trade is predominantly focused on resource extraction (i.e., coal, oil, gas and mineral mining), accounting for 45.4% of emissions embodied in exports. A possible explanation of these patterns is that global North are outsourcing resource mining industries to the global South, with a flow of resources back through global supply chains ^{6,7}. More importantly, these trade patterns have intensified over the past decades. Detailed sectoral emissions embodied in trade of countries are provided in **Error! Reference source not found.***

In section 'Drivers of consumption-based methane emissions':

This suggests that there is an increase in the consumption of methane-intensive products or a rise in the import of goods from countries with higher methane emission intensity, attributed to shifts in the global trade structure. For example, the share of red meats (beef, lamb, and pork), which are methane-intensive products ^{46,56}, in total global demand has increased from 0.6% to 0.8% between 2008 and 2023.

Such decreases in emission coefficient have been observed across sectors. For example, between 2018 and 2023, 97 out of 120 sectors reduced their emission coefficient. In particular, three sectors with the highest emission coefficient—raising of animals and agricultural services; water collection, treatment, and supply, sewerage; and raising of cattle—reduced their emission coefficient by 42%, 32%, and 20%, respectively, during this period.

In the conclusions:

Their decoupling is mainly caused by a decline in their emission coefficient, and to a lesser degree due to outsourcing methane emissions to less developed regions. Among the 120 sectors classified in the GLORIA configuration in developed countries, 100 of them reduced their emission coefficient between 2003 and 2013. This is evident from the analysis of the drivers, where emission coefficient is the primary determinant in reducing emissions.

Methodology

The analysis carried out is mostly descriptive (emission trends, analysis of emissions embodied in trade, as well as analyses based on a structural decomposition and the decoupling index). Therefore, it should be taken with caution. However, some results are reported as if they show causality.

RE: Thank you for your comment. SDA is used to decompose changes in emissions into independent drivers (or determinants), and it does not suffer from omitted variable bias or endogeneity issues, unlike regression analysis. Tapio decoupling index is used to identify the relationship between emissions and economic growth, we do not use the index to interpret causality between the variables. Both SDA and Tapio decoupling index are widely used in environmental studies. We have provided a more detailed clarification in response to your comments below.

The components of the structural analysis are not clearly explained, and their economic interpretation is not clear for the reader.

RE: Thank you very much for this suggestion. We have accordingly added the economic interpretation of the five drivers decomposed in our SDA analysis.

*The idea of an SDA is to consider the change in emissions due to the change in each driver, assuming the other drivers remain unchanged. Therefore, the five items in **Error! Reference source not found.** give the respective contribution of each driver to the change in CBE. (ΔCBE_{int} gives the impact of the emission coefficient, often interpreted as the contribution of emission technology. As emission technology improves, emissions per unit of output decrease directly. ΔCBE_L provides the contribution of the changes in the global Leontief inverse, often interpreted as changes in production technology. It reflects transformations in global production structures which encompass both domestic economic structure and inter-country input-output industrial linkages⁷⁵. ΔCBE_{FS} captures the impact of the final demand structure. For instance, a shift in final demand towards emission-intensive products will lead to an increase in overall emissions. ΔCBE_{FDper} and ΔCBE_P provide the contribution of final demand per capita and population, respectively. Final demand per capita often reflects a country's economic level, as wealthier nations tend to consume more.*

More important, the Tapio decoupling index cannot be read in terms of causality and may suffer from endogeneity. First of all, methane emissions and economic activity are simultaneously determined. Also, prior research shows that together with economic growth there are other determinants of methane emissions like population, sectoral composition, trade openness, institutional factors and lagged methane emissions. As long as these factors are not included in the analysis, the decoupling index calculated for methane emissions may suffer from omitted variable bias. This is a key issue. The authors should have addressed this issue and should have compared their results with those shown in the literature.

RE: We fully agree that the Tapio decoupling index cannot be used to interpret causality. In this study, we only use the index to describe the dynamic relationship between emissions (both PBE and CBE) and economic growth (i.e., GDP) in each country and aggregated group. That is, we use the decoupling index to do descriptive analysis, and we do not use it to interpret any causality relationship between methane emissions and economic growth.

Instead, we use SDA to analyse the determinants of the changes in the consumption-based methane emissions. SDA is a method used to analyse the drivers of change in a dependent variable by decomposing the aggregate change into contributions from various underlying drivers (or determinants). These drivers are seen as being independent of one another, and their combined contributions sum to 100%. Consequently, SDA does not suffer from any omitted variable bias or endogeneity problem. This is essentially different from regression analysis.

As explained above, in this study we decompose five drivers of the changes in the consumption-based methane emissions: emission intensity (reflecting emission technology), global Leontief inverse (reflecting overall production technology and intermediate trade), final demand structure, final demand per capita, and population.

We are aware of that determinants like trade openness and institutional factors may influence the drivers we have decomposed. For example, trade openness can impact trade of intermediate and final products, thereby affecting the global Leontief inverse and final demand. However, these determinants operate as secondary layers of influencing factors. SDA captures the contributions of first-layer drivers, with the effects of second-layer factors already embedded within these first-layer drivers⁶⁷. Due to this feature, SDA is widely

accepted in economics, environmental studies, and industrial ecology to investigate the drivers of changes in variables such as production, emissions, and energy use ⁶⁸⁻⁷⁰.

Results and policy recommendations

The results are mostly stylized facts which do not provide clearly significant new insights about the relationship of methane emissions and economic growth. Which new insights can be extracted? The authors should put these stylized facts in context, showing the contribution with respect to the existing literature. Also, they must emphasize that these trends detected are descriptive and no causal relation can be extracted from them.

RE: We have rewritten the conclusions to better summarize our findings, we also compared our results to existing literature in the result sections to show our contributions, as you can see our response to comments above.

We have emphasized that these trends detected are descriptive and no casual relation can be extracted from the results in the method section.

*Tapio decoupling index ⁷¹ is utilized to gauge the extent of decoupling between methane emissions and economic growth in individual countries, as shown in **Error! Reference source not found.** Such relationships between emissions and economic growth detected by the decoupling index are descriptive and no causality can be extracted.*

A claim made by the authors is that they base their calculations on a highly disaggregated database. Yet, they do not provide results based on such a disaggregation but at a level of aggregation that is already present in other research. They should take advantage of their data and show the advantage of having results disaggregated

RE: We have added more sector-specific details in the revised manuscript and provided the detailed sectoral results in the supplementary tables. Please refer to our response to your comment above.

Some of the statements are incorrect or miss enough scientific foundation. For example:

1. Line 111-112: Attributing the increase in emissions to WTO-membership of China is not grounded in any reference and is outside the scope of the paper.

RE: We removed China's WTO membership from the main text.

32.5% of the global total increment (37.2 million tons) over the period can be attributed to China, whose emissions increased by 12.1 million tons due to increasing production for both domestic use and exports.

- Line 118: “Unfortunately” is a subjective term that should be avoided in a scientific paper. Although they add to the problem of climate change, anthropogenic methane emissions increase linked to economic growth and development.

RE: We removed the word ‘unfortunately’.

That is to say global methane emission have been growing faster in recent years ⁴¹, albeit with some fluctuations, which is in contrast with the recent slowdown observed in CO₂ emission growth ⁴².

- Lines 282-283: There is not enough causal evidence provided by the authors to claim outsourcing of heavy-emitting industries.

RE: We changed the sentence to:

A possible explanation of these patterns is that global North are outsourcing resource mining industries to the global South, with a flow of resources back through global supply chains^{6,7}.

- Paragraph lines 378-298: This paragraph is not derived from the analysis and includes aspects from trade relations and geopolitics that are not well treated. Accordingly, the derivation of the policy recommendation is unclear or naive.

RE: We rewrote this section by removing the text on geopolitics but retaining the suggestions on selecting trade partners with low emission intensities, which is a key solution for reducing a country's consumption-based emissions.

We find that global trade accounts for approximately 30% of methane emissions, which underscores the key role of global trade in influencing emission patterns, providing valuable insights for shaping effective reduction policies. As sectors across countries differ in technological endowment, production efficiency, emission coefficient of products produced in different countries varies. This provides an opportunity to reduce a sector's methane footprint by supply chain management of upstream production through carefully selecting import partners with low emission coefficient. Taking the production and trade of fertilizers as an example, the production of which is one of the top methane sources, Russia is the world's leading exporter of fertilizers. It exported 18.7 billion USD worth in 2022, accounting for 13.1% of the global total export, followed by Canada (14.5 billion USD), China (12.7), US (8.19), and Morocco (7.92)¹⁰. Our results find that the emission coefficient in Russia's fertilizer production is 0.07 tons/million USD/yr in 2022. However, the emission coefficients of fertilizer production in Morocco (with 0.002 tons/million USD/yr), Canada (0.05), US (0.04) and China (0.06) are lower than in Russia, indicating these countries have more advanced production technologies with higher efficiency or alternative sources for fertilizer production, reducing associated methane emissions. Countries that currently import fertilizer from Russia can reduce their imported embodied emissions by shifting to other exporting countries with lower emission coefficient. However, it is important to consider that this shift could lead to other environmental problems (e.g. in the case of Morocco, which has a larger share of phosphate fertilizers but lower methane emissions). It is encouraging to see the EU making efforts to reduce the import of high-emitting products from non-European countries by setting up the Carbon Border Adjustment Mechanism (CBAM), which will invite EU countries to choose cleaner trade partners and upstream partners to reduce their emissions to stay competitive. CBAM includes fertilizers as one of the initial goods and selected precursors whose production is emission-intensive and at most significant risk of carbon leakage. The EU accounts for 23% of global fertilizer imports. Future actions of CBAM should focus on expanding to a larger range of products, including methane-intensive ones¹¹.

Therefore, some of the conclusions and policy implications derived by the authors require further justification and a more direct link to the results.

RE: We have rewritten the conclusions and policy implications.

This study employs an environmentally extended global MRIO approach, decoupling analysis, and SDA, to analyze emission changes in global and 164 countries' methane emissions from both production- and consumption-based perspectives during 1990 to 2023.

We find that developed countries stand out as the sole group that has consistently achieved emission reductions and strong decoupling between emissions (both PBE and

CBE) and GDP. Their decoupling is mainly caused by a decline in their emission coefficient, and to a lesser degree due to outsourcing methane emissions to less developed regions. Among the 120 sectors classified in the GLORIA configuration in developed countries, 100 of them reduced their emission coefficient between 2003 and 2013. This is evident from the analysis of the drivers, where emission coefficient is the primary determinant in reducing emissions. Therefore, many sectors in developed countries provide successful examples in methane emission control for other countries around the world.

Asia and developing countries, particularly due to the rapid growth of China, have been major contributors to the increase in methane emissions over the past decades, establishing themselves as leading global methane emitters. Consequently, we recommend directing our focus on addressing methane emissions in Asia and developing Pacific countries. While the joint announcement by the United States and China at COP28 is a positive initiative, further efforts are necessary to facilitate developed countries' support for developing countries in reducing emissions through financial and technical assistance. It is essential to recognize that, although emerging and small countries currently emit lower levels of methane, they may become key methane emitters in the future due to economic growth. Therefore, they should not be overlooked when designing global goals for GHG reduction, and policies need to be designed to prevent emissions from increasing rapidly at this stage.

We find that global trade accounts for approximately 30% of methane emissions, which underscores the key role of global trade in influencing emission patterns, providing valuable insights for shaping effective reduction policies. As sectors across countries differ in technological endowment, production efficiency, emission coefficient of products produced in different countries varies. This provides an opportunity to reduce a sector's methane footprint by supply chain management of upstream production through carefully selecting import partners with low emission coefficient. Taking the production and trade of fertilizers as an example, the production of which is one of the top methane sources, Russia is the world's leading exporter of fertilizers. It exported 18.7 billion USD worth in 2022, accounting for 13.1% of the global total export, followed by Canada (14.5 billion USD), China (12.7), US (8.19), and Morocco (7.92)¹⁰. Our results find that the emission coefficient in Russia's fertilizer production is 0.07 tons/million USD/yr in 2022. However, the emission coefficients of fertilizer production in Morocco (with 0.002 tons/million USD/yr), Canada (0.05), US (0.04) and China (0.06) are lower than in Russia, indicating these countries have more advanced production technologies with higher efficiency or alternative sources for fertilizer production, reducing associated methane emissions. Countries that currently import fertilizer from Russia can reduce their imported embodied emissions by shifting to other exporting countries with lower emission coefficient. However, it is important to consider that this shift could lead to other environmental problems (e.g. in the case of Morocco, which has a larger share of phosphate fertilizers but lower methane emissions). It is encouraging to see the EU making efforts to reduce the import of high-emitting products from non-European countries by setting up the Carbon Border Adjustment Mechanism (CBAM), which will invite EU countries to choose cleaner trade partners and upstream partners to reduce their emissions to stay competitive. CBAM includes fertilizers as one of the initial goods and selected precursors whose production is emission-intensive and at most significant risk of carbon leakage. The EU accounts for 23% of global fertilizer imports. Future actions of CBAM should focus on expanding to a larger range of products, including methane-intensive ones¹¹.

Our analysis of driving forces identifies that reduction in emission coefficient via emission technology improvement is the main determinant for reducing emissions and achieving strong decoupling over the observation period. Reducing methane emission coefficient requires targeted strategies across sectors. For example, the energy sector can apply advanced detection technologies, such as infrared imaging and drones, in oil and gas extraction, storage, and transportation to identify and repair emission leaks ^{12,13}; replace coal and oil with clean energy across the entire energy system ¹⁴; improve ventilation systems in oil and coal mines to reduce uncontrolled methane emissions ¹⁵; and deploy equipment to capture associated methane emissions for recycling rather than flaring. In the agricultural sector, we can improve feed formulations to reduce methane production in ruminants by incorporating additives like oils, enzymes, or methane inhibitors into livestock diets ^{16,17}; increase biogas recovery by processing manure and organic waste into energy ¹⁸; and improve rice farming through irrigation control (e.g., alternate wetting and drying methods) and better fertilization management ¹⁹. In the waste management sector, we can enhance landfill management by sealing landfills and installing methane recovery systems to recycle methane for power generation or heating ; promote composting technologies by replacing landfilling with composting to effectively reduce methane emissions from organic waste ²⁰; and upgrade wastewater treatment by adopting advanced anaerobic digestion technologies to capture methane from wastewater and use it as an energy source ^{21,22}.

The measures above can reduce emission coefficients from the production side. However, unlike CO₂ emissions, which mainly stem from fossil fuels, a major source of methane emissions are biological activities, such as biodegradation in wastewater treatment, fertilizers production, and effusion from coal mining, livestock farming. We further call for smart demand-side emission reduction strategies to supplement the aforementioned reduction measures in mitigating methane emissions, particularly from biological activities. For example, we can decrease dietary forage-to-concentrate ratio and promoting a shift towards consuming less red meat in our diets ^{56,72}. Since methane has a relatively short lifespan of 7 to 12 years in the atmosphere, our demand-side efforts today will have an immediate impact on its global warming effects.

References

- 1 Lenzen, M., Geschke, A., West, J., Fry, J., Malik, A., Giljum, S. *et al.* Implementing the material footprint to measure progress towards Sustainable Development Goals 8 and 12. *Nature Sustainability* **5**, 157-166 (2022).
- 2 Sun, X., Cheng, X., Guan, C., Wu, X. & Zhang, B. Economic drivers of global and regional CH₄ emission growth from the consumption perspective. *Energy Policy* **170**, 113242 (2022).
- 3 Fernández-Amador, O., Francois, J. F., Oberdabernig, D. A. & Tomberger, P. The methane footprint of nations: Stylized facts from a global panel dataset. *Ecological Economics* **170**, 106528 (2020).
- 4 Cheng, X., Wu, X., Guan, C., Sun, X. & Zhang, B. Impacts of production structure changes on global CH₄ emissions: Evidences from income-based accounting and decomposition analysis. *Ecological Economics* **213**, 107967 (2023).
- 5 Statista. Distribution of coal production worldwide in 2020, by major countries. <<https://www.statista.com/statistics/265638/distribution-of-coal-production-worldwide/>> (2021).
- 6 Yang, Y., Zhou, Y., Shan, Y. & Hubacek, K. The shift of embodied energy flows among the Global South and Global North in the post-globalisation era. *Energy Economics* **131**, 107408 (2024).
- 7 Parsons, L. *Carbon colonialism: How rich countries export climate breakdown*. (Manchester University Press, 2023).
- 8 European Commission. EDGAR (Emissions Database for Global Atmospheric Research) Community GHG Database-EDGAR CO₂, EDGAR CH₄, EDGAR N₂O, EDGAR F-GASES. *Joint Research Centre (JRC) the European Commission, the International Energy Agency (IEA), and comprising IEA*, <<https://edgar.jrc.ec.europa.eu/>> (2024).
- 9 IEA. Global Methane Tracker 2024. (IEA, Paris, 2024). <<https://www.iea.org/reports/global-methane-tracker-2024>>.
- 10 Observatory of Economic Complexity (OEC) World. Fertilizers. <<https://oec.world/en/profile/hs/fertilizers>> (2024).
- 11 Böhringer, C., Fischer, C., Rosendahl, K. E. & Rutherford, T. F. Potential impacts and challenges of border carbon adjustments. *Nature Climate Change* **12**, 22-29 (2022).
- 12 Yang, X., Kuru, E., Zhang, X., Zhang, S., Wang, R., Ye, J. *et al.* Direct measurement of methane emissions from the upstream oil and gas sector: Review of measurement results and technology advances (2018–2022). *Journal of Cleaner Production* **414**, 137693 (2023).
- 13 Shaw, J. T., Shah, A., Yong, H. & Allen, G. Methods for quantifying methane emissions using unmanned aerial vehicles: A review. *Philosophical Transactions of the Royal Society A* **379**, 20200450 (2021).

- 14 IEA. The Imperative of Cutting Methane from Fossil Fuels. (IEA, Paris, 2023). <<https://www.iea.org/reports/the-imperative-of-cutting-methane-from-fossil-fuels>>.
- 15 Mundra, I. & Lockley, A. Emergent methane mitigation and removal approaches: A review. *Atmospheric Environment: X* **21**, 100223 (2024).
- 16 Arndt, C., Hristov, A. N., Price, W. J., McClelland, S. C., Pelaez, A. M., Cueva, S. F. *et al.* Full adoption of the most effective strategies to mitigate methane emissions by ruminants can help meet the 1.5 C target by 2030 but not 2050. *Proceedings of the National Academy of Sciences* **119**, e2111294119 (2022).
- 17 Temple, J. *Cattle burping remedies: 10 Breakthrough Technologies 2025*, <<https://www.technologyreview.com/2025/01/03/1108827/cow-cattle-burping-methane-cutting-supplements-additives-carbon-emissions-breakthrough-technologies-2025/>> (Accessed in 2025).
- 18 Cucina, M. Integrating anaerobic digestion and composting to boost energy and material recovery from organic wastes in the circular economy framework in Europe: A review. *Bioresource Technology Reports*, 101642 (2023).
- 19 Yang, J., Zhou, Q. & Zhang, J. Moderate wetting and drying increases rice yield and reduces water use, grain arsenic level, and methane emission. *The Crop Journal* **5**, 151-158 (2017).
- 20 Maasackers, J. D., Varon, D. J., Elfarsdóttir, A., McKeever, J., Jervis, D., Mahapatra, G. *et al.* Using satellites to uncover large methane emissions from landfills. *Science Advances* **8**, eabn9683 (2022).
- 21 Archana, K., Visckram, A., Kumar, P. S., Manikandan, S., Saravanan, A. & Natrayan, L. A review on recent technological breakthroughs in anaerobic digestion of organic biowaste for biogas generation: Challenges towards sustainable development goals. *Fuel* **358**, 130298 (2024).
- 22 Almansa, X. F., Starostka, R., Raskin, L., Zeeman, G., De Los Reyes III, F., Waechter, J. *et al.* Anaerobic Digestion as a Core Technology in Addressing the Global Sanitation Crisis: Challenges and Opportunities. *Environmental Science & Technology* **57**, 19078-19087 (2023).
- 23 Zhang, Y., Wu, X., Guan, C. & Zhang, B. Methane emissions of major economies in 2014: A household-consumption-based perspective. *Science of the Total Environment* **768**, 144523 (2021).
- 24 Bruckner, B., Shan, Y., Prell, C., Zhou, Y., Zhong, H., Feng, K. *et al.* Ecologically unequal exchanges driven by EU consumption. *Nature Sustainability*, 1-12 (2023).
- 25 Hubacek, K., Chen, X., Feng, K., Wiedmann, T. & Shan, Y. Evidence of decoupling consumption-based CO₂ emissions from economic growth. *Advances in Applied Energy* **4**, 100074 (2021).
- 26 Fernández-Amador, O., Oberdabernig, D. A. & Tomberger, P. Do methane emissions converge? Evidence from global panel data on production-and consumption-based emissions. *Empirical Economics* **63**, 877-900 (2022).

- 27 Zhu, K. & Jiang, X. Slowing down of globalization and global CO₂ emissions—A causal or casual association? *Energy Economics* **84**, 104483 (2019).
- 28 Jonas, M., Marland, G., Krey, V., Wagner, F. & Nahorski, Z. Uncertainty in an emissions-constrained world. *Uncertainties in Greenhouse Gas Inventories: Expanding Our Perspective*, 9-26 (2015).
- 29 Intergovernmental Panel on Climate Change (IPCC). *Revised 1996 IPCC Guidelines for national greenhouse gas inventories*. (Intergovernmental Panel on Climate Change, 1996).
- 30 Shan, Y., Guan, D., Zheng, H., Ou, J., Li, Y., Meng, J. *et al.* Data Descriptor: China CO₂ emission accounts 1997-2015. *Scientific Data* **5** (2018).
- 31 Lenzen, M., Wood, R. & Wiedmann, T. Uncertainty analysis for multi-region input-output models - a case study of the UK's carbon footprint. *Economic Systems Research* **22**, 43-63 (2010).
- 32 Dhakal, S., Minx, J. C., Toth, F., Abdl-Aziz, A., Meza, M. F., Hubacek, K. *et al.* Emissions trends and drivers in *Climate Change 2021: Mitigation of Climate Change. Contribution of Working Group III to the Sixth Assessment Report of the Intergovernmental Panel on Climate Change* (2021).
- 33 Schulte, S., Jakobs, A. & Pauliuk, S. Estimating the uncertainty of the greenhouse gas emission accounts in global multi-regional input–output analysis. *Earth System Science Data* **16**, 2669-2700 (2024).
- 34 Rodrigues, J. F., Moran, D., Wood, R. & Behrens, P. Uncertainty of consumption-based carbon accounts. *Environmental Science & Technology* **52**, 7577-7586 (2018).
- 35 Aguiar, A., Chepeliev, M., Corong, E. & van der Mensbrugghe, D. The global trade analysis project (GTAP) data base: Version 11. *Journal of Global Economic Analysis* **7** (2022).
- 36 Saunio, M., Stavert, A. R., Poulter, B., Bousquet, P., Canadell, J. G., Jackson, R. B. *et al.* The global methane budget 2000–2017. *Earth System Science Data* **12**, 1561-1623 (2020).
- 37 Jones, M. W., Peters, G. P., Gasser, T., Andrew, R. M., Schwingshackl, C., Gütschow, J. *et al.* National contributions to climate change due to historical emissions of carbon dioxide, methane, and nitrous oxide since 1850. *Scientific Data* **10**, 155 (2023).
- 38 McDuffie, E. E., Smith, S. J., O'Rourke, P., Tibrewal, K., Venkataraman, C., Marais, E. A. *et al.* A global anthropogenic emission inventory of atmospheric pollutants from sector-and fuel-specific sources (1970–2017): an application of the Community Emissions Data System (CEDS). *Earth System Science Data* **12**, 3413-3442 (2020).
- 39 Stavert, A. R., Saunio, M., Canadell, J. G., Poulter, B., Jackson, R. B., Regnier, P. *et al.* Regional trends and drivers of the global methane budget. *Global Change Biology* **28**, 182-200 (2022).

- 40 He, J., Naik, V., Horowitz, L. W., Dlugokencky, E. & Thoning, K. Investigation of the global methane budget over 1980–2017 using GFDL-AM4. 1. *Atmospheric Chemistry and Physics* **20**, 805-827 (2020).
- 41 Zhang, B., Chen, G., Li, J. & Tao, L. Methane emissions of energy activities in China 1980–2007. *Renewable and Sustainable Energy Reviews* **29**, 11-21 (2014).
- 42 Turner, A. J., Jacob, D., Benmergui, J., Wofsy, S., Maasackers, J., Butz, A. *et al.* A large increase in US methane emissions over the past decade inferred from satellite data and surface observations. *Geophysical research letters* **43**, 2218-2224 (2016).
- 43 Ramsden, A. E., Ganesan, A. L., Western, L. M., Rigby, M., Manning, A. J., Foulds, A. *et al.* Quantifying fossil fuel methane emissions using observations of atmospheric ethane and an uncertain emission ratio. *Atmospheric Chemistry and Physics* **22**, 3911-3929 (2022).
- 44 Smith, P., Reay, D. & Smith, J. Agricultural methane emissions and the potential formitigation. *Philosophical Transactions of the Royal Society A* **379**, 20200451 (2021).
- 45 Bai, Y., Guo, C., Li, S., Degen, A. A., Ahmad, A. A., Wang, W. *et al.* Instability of decoupling livestock greenhouse gas emissions from economic growth in livestock products in the Tibetan highland. *Journal of Environmental Management* **287**, 112334 (2021).
- 46 Li, Y., Zhong, H., Shan, Y., Hang, Y., Wang, D., Zhou, Y. *et al.* Changes in global food consumption increase GHG emissions despite efficiency gains along global supply chains. *Nature Food* **4**, 483-495 (2023).
- 47 Du, M., Zhu, Q., Wang, X., Li, P., Yang, B., Chen, H. *et al.* Estimates and predictions of methane emissions from wastewater in China from 2000 to 2020. *Earth's Future* **6**, 252-263 (2018).
- 48 Wang, D., Ye, W., Wu, G., Li, R. Q., Guan, Y., Zhang, W. *et al.* Greenhouse gas emissions from municipal wastewater treatment facilities in China from 2006 to 2019. *Scientific Data* **9**, 317 (2022).
- 49 Song, C., Zhu, J.-J., Willis, J. L., Moore, D. P., Zondlo, M. A. & Ren, Z. J. Methane Emissions from Municipal Wastewater Collection and Treatment Systems. *Environmental Science & Technology* **57**, 2248-2261 (2023).
- 50 Höglund-Isaksson, L. Bottom-up simulations of methane and ethane emissions from global oil and gas systems 1980 to 2012. *Environmental Research Letters* **12**, 024007 (2017).
- 51 Gao, J., Guan, C. & Zhang, B. China's CH₄ emissions from coal mining: A review of current bottom-up inventories. *Science of the Total Environment* **725**, 138295 (2020).
- 52 Sun, X., Li, Z., Cheng, X., Guan, C., Han, M. & Zhang, B. Global anthropogenic CH₄ emissions from 1970 to 2018: Gravity movement and decoupling evolution. *Resources, Conservation and Recycling* **182**, 106335 (2022).

- 53 Ari, I. & Şentürk, H. The relationship between GDP and methane emissions from solid waste: A panel data analysis for the G7. *Sustainable Production and Consumption* **23**, 282-290 (2020).
- 54 Zhang, B., Zhao, X., Wu, X., Han, M., Guan, C. H. & Song, S. Consumption - based accounting of global anthropogenic CH₄ emissions. *Earth's Future* **6**, 1349-1363 (2018).
- 55 Lenzen, M., Kanemoto, K., Moran, D. & Geschke, A. Mapping the structure of the world economy. *Environmental Science and Technology* **46**, 8374-8381 (2012).
- 56 Li, Y., He, P., Shan, Y., Li, Y., Hang, Y., Shao, S. *et al.* Reducing climate change impacts from the global food system through diet shifts. *Nature Climate Change* **14**, 943-953 (2024).
- 57 Liu, Y., Yan, C., Gao, J., Wu, X. & Zhang, B. Mapping the changes of CH₄ emissions in global supply chains. *Science of the Total Environment* **832**, 155019 (2022).
- 58 Dietzenbacher, E., Los, B., Stehrer, R., Timmer, M. & De Vries, G. The construction of world input–output tables in the WIOD project. *Economic Systems Research* **25**, 71-98 (2013).
- 59 Ma, R., Chen, B., Guan, C., Meng, J. & Zhang, B. Socioeconomic determinants of China's growing CH₄ emissions. *Journal of Environmental Management* **228**, 103-116 (2018).
- 60 Wang, Y., Chen, B., Guan, C. & Zhang, B. Evolution of methane emissions in global supply chains during 2000–2012. *Resources, Conservation and Recycling* **150**, 104414 (2019).
- 61 Han, M., Zhang, B., Zhang, Y. & Guan, C. Agricultural CH₄ and N₂O emissions of major economies: consumption-vs. production-based perspectives. *Journal of Cleaner Production* **210**, 276-286 (2019).
- 62 Garsous, G. Developing consumption-based emissions indicators from Agriculture, Forestry and Land-use (AFOLU) activities. (2021).
- 63 Caro, D., LoPresti, A., Davis, S. J., Bastianoni, S. & Caldeira, K. CH₄ and N₂O emissions embodied in international trade of meat. *Environmental Research Letters* **9**, 114005 (2014).
- 64 Lenzen, M. Aggregation versus disaggregation in input–output analysis of the environment. *Economic Systems Research* **23**, 73-89 (2011).
- 65 Steen-Olsen, K., Owen, A., Hertwich, E. G. & Lenzen, M. Effects of sector aggregation on CO₂ multipliers in multiregional input–output analyses. *Economic Systems Research* **26**, 284-302 (2014).
- 66 Tian, K., Dietzenbacher, E., Yan, B. & Duan, Y. Upgrading or downgrading: China's regional carbon emission intensity evolution and its determinants. *Energy Economics* **91**, 104891 (2020).

- 67 Dietzenbacher, E. & Los, B. Structural decomposition techniques: sense and sensitivity. *Economic Systems Research* **10**, 307-324 (1998).
- 68 Hoekstra, R. & Van den Bergh, J. C. Comparing structural decomposition analysis and index. *Energy Economics* **25**, 39-64 (2003).
- 69 Su, B. & Ang, B. W. Structural decomposition analysis applied to energy and emissions: Some methodological developments. *Energy Economics* **34**, 177-188 (2012).
- 70 Feng, K., Davis, S. J., Sun, L. & Hubacek, K. Drivers of the US CO₂ emissions 1997–2013. *Nature Communications* **6**, 7714 (2015).
- 71 Tapio, P. Towards a theory of decoupling: Degrees of decoupling in the EU and the case of road traffic in Finland between 1970 and 2001. *Transport Policy* **12**, 137-151 (2005).
- 72 Pérez-Domínguez, I., Del Prado, A., Mittenzwei, K., Hristov, J., Frank, S., Tabeau, A. *et al.* Short-and long-term warming effects of methane may affect the cost-effectiveness of mitigation policies and benefits of low-meat diets. *Nature Food* **2**, 970-980 (2021).

Response letter to Reviewers' comments

Dear editor and reviewers,

Thank you very much for your consideration of our revision and your efforts in managing its review. We very much appreciate the comments we received. Please find our point-by-point response to the comments below.

Reviewer #1

The paper has been revised and is now much clearer in its analysis and methods used. However, there are areas which require further addressing before the manuscript can be considered for publication.

- There are still grammatical errors and incorrect tenses throughout the manuscript, as well as sections where the writing quality is poor.

Given the journal the authors are trying to publish in, these should be improved and corrected e.g. line 139 'have' not 'has'; line 143 'kept' is unnecessary and 'increased' not 'increasing'; line 283 'was' or 'remained' not 'kept'

RE: Thanks. We have revised the mistakes you pointed out and corrected the language issues throughout the manuscript. We are confident that any remaining typos will be caught at the proofing stage.

- Section 'Drivers of consumption-based methane emissions' is missing references in parts where it appears the authors are referring to the literature e.g. lines 342 to 348.

RE: The text and numbers in lines 342 to 348 are based on our own calculations; therefore, no references are cited. We have revised the text to prevent any misunderstanding.

Our results show that global average emission coefficients decreased by 66.7% between 1998 and 2023. Such decreases in emission coefficients have been observed across sectors. For example, between 2018 and 2023, 97 out of 120 sectors reduced their emission coefficient.

- Lines 345 to 348- can the authors expand on the reduction in emission coefficients? Especially for animal raising and cattle, how has the coefficient reduced by 42 and 20%?

RE: We added discussions on potential practices in these three sectors to reduce emission coefficients.

In particular, three sectors with the highest emission coefficient—raising of animals and agricultural services; water collection, treatment, and supply, sewerage; and raising of cattle—reduced their emission coefficient by 42%, 32%, and 20%, respectively, during this period. These reductions can be attributed to various advances in production practices, such as new irrigation control systems and improved fertilization management in the agricultural sector⁵³; enhanced livestock, manure management, and breeding efficiency in animal husbandry^{53,54}; the recycling and reuse of waste, along with emerging anaerobic digestion technologies in sewage treatment^{55,56}.

Reviewer #2

General comments

Changes made by the authors has significantly strengthened the introduction. Specifically, the increased discussion of existing studies has provided key context and justification for the study

while the early explanation of the PBE and CBE concepts has improved the readability of the paper.

The increased detail provided in the “Methane emissions” section is useful and interesting. However, it could be expressed more concisely. For example, “In terms of PBE, we can see that Asia and the developing Pacific has been dominating global methane PBE since the mid-1990s. Its PBE increased rapidly from 96.9 million tons/yr (36.4% of global emissions) in 1990 to 162.4 (42.4%) in 2023. In the meanwhile, developed countries, although being the 2nd largest contributor to global methane PBE, show a declining trend. Their PBE decreased from 74.7 million tons/yr (28.0%) in 1990 to 60.4 (15.8%) in 2023.” could be expressed as, “Since the mid-1990’s the Asian and developing pacific PBE has dominated global PBE, increasing from 96.9 million t/yr (36.4% of global PBE) in 1990 to 162.4 (42.4%) in 2023. Meanwhile, the PBE of Developed Countries, globally the 2nd largest contributor, decreased from 74.7 million t/yr (28.0%) in 1990 to 60.4 (15.8%) in 2023.”

RE: Done. Thanks for the suggestion.

I remain concerned that the paper conflates a change in emissions coefficient with a change in actual emissions (not just a change in the estimates of these emissions).

Emissions coefficients are a best guess at relating sectoral emissions to the activity of that sector. However, their accuracy is limited and for many sectors they have very large uncertainties. This varies globally with some nations using country-specific derived emissions coefficients (nominally) reflective of their emissions portfolio while others use default estimates from the IPCC. Also of concern is that, for some sectors, reported emissions based on emissions coefficients differ significantly to those determined using Top-Down approaches (based on satellite and/or ground-based observations i.e. independent of emissions coefficients). E.g. Sadavarte et al. (2021) found coal mine methane emissions seven times higher than those estimated by the EDGAR approach.

As such, I am concerned by the emphasis in the Conclusion of increasing/focusing trade with countries with low emissions coefficients (e.g. page 14, line 31) as a method to reduce emissions as it embeds an unarticulated (and somewhat tenuous) assumption that the emissions coefficients are an accurate representation of actual emissions. Instead, trade should be focused on countries with low emissions (relative to the item being traded). I do not think an in-depth exploration of the limitations of emissions coefficients is within the scope of the paper. However, I do think that the Conclusion needs to include a statement acknowledging that:

1. The paper assumes that the emissions coefficients are an accurate representation of actual methane emissions but emissions coefficients can (and do) have large uncertainties, 2. Prioritising trade with low emissions coefficient partners will only reduce emissions if said coefficients are accurate representations of emissions. 3. Top-down (observation based) research is needed to independently verify that such changes in trading patterns leads to a decline in emissions.

RE: Thanks for your comments. There appears to be a misunderstanding regarding emission coefficients versus emission factors.

- Emission Coefficients: emissions per unit of output, which is conceptually similar to Emission Intensity, typically defined as emissions per unit of GDP. The sectoral emission coefficient is calculated by us using sectoral PBE and sectoral outputs from the MRIO table.
- Emission Factors: emissions per unit of activity data (e.g., energy consumption, industrial production, etc.) and are commonly used in the bottom-up accounting of emissions.

The uncertainties raised by the reviewer arise from the emission factors, not from the emission coefficients. As the reviewer rightly points out, some countries use country-specific emission factors, while others rely on default values provided by the IPCC. This variation introduces uncertainty into emission estimates.

However, these uncertainties in emission factors are beyond the scope of our study. We did not calculate emission estimates using emission factors ourselves. Instead, we used emission data directly sourced from the GLORIA environmental accounts, which integrate datasets from EDGAR, the OECD, and Eurostat.

When we suggest in the conclusion that trade should increase or focus on countries with low emission coefficients, we are exactly referring to countries with low emissions relative to the economic value or volume of the items being traded, in alignment with the reviewer's comment.

To clarify the statement, we have defined the emission coefficient in the section of Drivers:

Changes in emission coefficients (i.e., emissions per unit of output) are the main determinant to reduce emissions over the period, offsetting the increasing effects from demand growth and structure changes, which is consistent with previous studies ^{31,32,52}.

and also in the method section:

$int_{i,j}$ is the emission coefficient in sector i of country j , which is PBE per unit of output in each country sector.

To acknowledge the limitations and uncertainties in the emission estimates, we have added a section at the end of the Methods section.

We acknowledge that there are uncertainties in the accounting of methane emissions, from both production- and consumption-based approaches. The uncertainties of PBE come from emission factors and activity data (e.g., energy consumption, industrial outputs, etc.) ⁷⁴. Top-down approaches based on satellite and ground-based observations can be used to enhance the accuracy of PBE accounts. For example, Sadavarte, et al. ⁷⁵ found that the methane emissions from coal mines estimated using satellite observations are considerably higher than those estimated by bottom-up approaches. Meanwhile, the uncertainties of CBE arise from the PBE coefficients and also from input-output data ^{76,77}.

We have performed uncertainty and robust analysis of countries' PBE and CBE as shown in the Supplementary and Table S3. In brief, we compare our PBE (i.e., GLORIA environmental accounts) with emission data from EDGAR v8.0_GHG 1970-2022 (CO₂, CH₄, N₂O, F-gases) ⁶ and IEA Methane Tracker ⁷⁸. Comparisons reveal that GLORIA methane emissions (379.5 million tons in 2022 and 383.5 in 2023) are 0.9% lower than EDGAR estimates in 2022 (388.3 million tons) and 9.7% higher than IEA Methane Tracker in 2023 (349.5 million tons). Regarding CBE, we compare our estimates with countries' CBE calculated using an alternative GTAP MRIO dataset ⁷⁹ and abovementioned PBE sources. The Pearson R value ranges from 0.952 to 0.998 for different comparisons, indicating the robustness of our calculations.

Lastly, while I have suggested a number of small changes below, the paper needs to be copy edited to improve clarity and succinctness.

RE: Thanks. We have addressed the mistakes you pointed out and thoroughly revised the manuscript.

Specific comments

I would suggest that the words “satellite data” are not used in the paper unless referring to satellite (orbiting space object derived) data. As also evidenced in comments by another reviewer and considering the target audience of the paper the use of the word adds an unnecessary layer of confusion.

RE: We have replaced the worlds ‘satellite data’ with ‘environmental accounts’.

We collect methane PBE from the environmental accounts of GLORIA ³⁹, which provides emissions data by countries and products consistent with the GLORIA MRIO table. The emission data includes both biogenic methane, produced and released from living plants and animals, and emissions from fossil fuel use. GLORIA environmental accounts combine EDGAR, the Organization for Economic Cooperation and Development, and Eurostat emission datasets.

P 2, In 12 Please change “air pollutant” to “air pollutants”

RE: Done.

P 3, In 11 Please change “spanning from 2000” to “spanning 2000”

RE: Done.

P 3, In 36 Please change “Based on our calculation” to “Based on our calculations”

RE: Done.

P4, Ins 14-15 Please change “the Supporting Document” to “the supporting documentation” or “the supplementary”.

RE: Done.

P5, In 5 Please change “Supplementary document” to “Supplementary”.

RE: Done.

P5, In 7 Please remove “perspectives” it’s superfluous.

RE: Done.

P5, In 10 Here “developed countries” is referring to the specific category you have defined in your study. As such, it should be capitalised to “Developed Countries” to distinguish it from the general term. This should be changed throughout the paper when referring to this specific category.

RE: Done

P6, In 16-18 The way this is currently written it seems to imply that Figure 1 d) shows the global totals aggregated into the sectors. It’s also not clear whether PBE or CBE is under discussion. For clarity I’d suggest rewording it as “To illustrate the sectoral distribution of methane PBE, we aggregated 120 sectors into seven categories, while underlying calculations are done at the 120-sector resolution. These seven sectors are shown in Figure 1-d) for the ten nations with the highest PBE.”

RE: Done.

P8, In 20 I’m assuming when you say “developing groups” you’re referring to countries not listed in the “Developed Countries” category? But it almost reads as if you’re referring to a new category here. Perhaps “Some countries not in the Developed Country category have already achieved strong decoupling in terms of both PBE and CBE. For example, Armenia...” would be clearer.

RE: Done. We changed it to ‘Some developing countries ...’.

P8, In 26 Please change “time period” to either “a time” or “a period”.

RE: Done.

P8, In 27 Please change “...decoupling results ... for every ten years’ interval...” to “...decoupling results ... at ten year intervals...”

RE: Done.

P9, In 4-6 Please change “but their proportion in global total emission kept unchanged” to “but as a proportion of global total emission remained unchanged”

RE: Done.

P9, In 6 Please change “In 2023, developed countries are the” to “In 2023, Developed Countries were the”

RE: Done.

P11, In 10 Not all developed countries are in the North (e.g Australia and NZ). Perhaps instead of “(i.e. developed countries)” try “(i.e. dominated by developed countries)”

RE: Thanks for your comments. The terms “Global North” and “Global South” are widely used to refer to developed countries (including Australia and New Zealand) and developing countries, respectively, particularly in trade-related analyses, such as in UN Trade and Development (UNCTAD) reports. These terms are not strictly geographical.

[Figure Redacted]

https://en.wikipedia.org/wiki/Global_North_and_Global_South

We have added a brief explanation in the main text to clarify the terms and their scope.

*With the rapid growth of developing countries, trade within the **Global South (encompassing four groups of developing countries)** has considerably increased, now playing a dominate role in the world trade patterns.*

*Emissions embodied in trade from **Global North (i.e., Developed Countries)** to Global South, also known as North-South trade, have declined from 49.4% (41.6 million tons/yr) of global trade embodied emissions in 1998 to 39.6% (47.1) in 2023.*

P11, In 13-14 Please change “A possible explanation of these patterns is that global North are outsourcing resource...” to A possible explanation of these patterns is that the global North outsources resource...”

RE: Done.

P15, In 3 Please change “...emissions.Countries...” to “...emissions. Countries...”

RE: Done.

P15, In 16 Please change “coefficient” to “coefficients”

RE: Done.

August 31st, 2024.

Re.: Referee report Nature Communications NCOMMS-24-32245-T
Global methane footprints growth and drivers 1990–2023

This paper uses multiregional input-output analysis to calculate methane emissions embodied in production and consumption using data from 1990–2023 for a global dataset (GLORIA, 164 countries/regions and a high disaggregation at the sectoral level). The authors calculate the two inventories (i.e. based on production and consumption) and describe some stylized facts—i.e. methane trends and analysis of methane embodied in trade, decoupling based on the Tapio index, and structural decomposition analysis to study the drivers of methane consumption inventories.

Although the topic of research is an interesting topic which has relevance for climate change, I have concerns about some aspects of the paper—particularly, (i) about the novelty and value added of the paper, (ii) about the methodology used, and (iii) about some policy recommendations and conclusions made by the authors.

- **Novelty of the paper:** The exercise implemented by the authors, the calculation of MRIO-based consumption inventories of methane emissions and the descriptive analysis of methane emissions embodied in trade flows, as well as the decoupling and the structural analysis are not new, even though the authors claim novelty. The authors do not cite recent literature that has dealt with the same questions, and which provided calculations using similar methodologies and regression-based analysis. These articles can be easily found in any browser or Google Scholar.

The analysis of methane production-, final production- and consumption-based inventories, as well as decomposition analysis (Kaya and LMDI decompositions) of those inventories has been carried out by Fernandez-Amador et al. (2020, *Ecological Economics*). This paper is not cited. This research also analyzes methane emissions embodied in trade. Although Fernandez-Amador et al.'s (2020) analysis goes until 2014, the authors have not made it clear their contribution with respect to that article. To my understanding, most of the findings are similar to those in that article.

The authors implement a simple Tapio decoupling index to analyze decoupling between methane emissions and economic growth. Besides other methodological issues that I comment below, the authors again claim novelty in the analysis of decoupling. However, they miss two papers, Fernandez-Amador et al. (2018, *Economics Letters*) and Fernandez-Amador et al. (2022, *Empirical Economics*). Fernandez-Amador et al. (2018) present estimates of the income-elasticity of methane based on regression analysis, while Fernandez-Amador et al. (2022) present dynamic regressions of methane and analyze β -convergence trends. The authors cite Fernandez-Amador et al. (2020, *Applied Economics*), which implements regression analysis at the sectoral level to analyze the environmental Kuznets curve for methane emissions and estimates the income-elasticity of methane at the sectoral level, but do not compare the results with the ones in that article. Being the relation with that research so direct, this is a serious issue. Some of these references implement their analysis not only for emissions per capita but also for emissions intensity (emissions per value added) for production and consumption inventories.

Fernandez-Amador et al. have made a CEPR Voxeu column on the topic (see link).

The authors should make a literature review and rethink the contribution of their research in relation to prior research. Moreover, they may show, for example, whether the high sectoral disaggregation of their dataset produces different results (i.e. whether other research suffers from aggregation bias).

- **Methodology:** The analysis carried out is mostly descriptive (emission trends, analysis of emissions embodied in trade, as well as analyses based on a structural decomposition and the decoupling index). Therefore, it should be taken with caution. However, some results are reported as if they show causality.

The components of the structural analysis are not clearly explained and their economic interpretation is not clear for the reader.

More important, the Tapio decoupling index cannot be read in terms of causality and may suffer from endogeneity. First of all, methane emissions and economic activity are simultaneously determined. Also, prior research shows that together with economic growth there are other determinants of methane emissions like population, sectoral composition, trade openness, institutional factors and lagged methane emissions. As long as these factors are not included in the analysis, the decoupling index calculated for methane emissions may suffer from omitted variable bias. This is a key issue. The authors should have addressed this issue and should have compared their results with those shown in the literature.

- **Results and policy recommendations:** The results are mostly stylized facts which do not provide clearly significant new insights about the relationship of methane emissions and economic growth. Which new insights can be extracted? The authors should put these stylized facts in context, showing the contribution with respect to the existing literature. Also, they must emphasize that these trends detected are descriptive and no causal relation can be extracted from them.

A claim made by the authors is that they base their calculations on a highly disaggregated database. Yet, they do not provide results based on such a disaggregation but at a level of aggregation that is already present in other research. They should take advantage of their data and show the advantage of having results disaggregated.

Some of the statements are incorrect or miss enough scientific foundation. For example:

Line 111–112: Attributing the increase in emissions to WTO-membership of China is not grounded in any reference and is outside the scope of the paper.

Line 118: "Unfortunately" is a subjective term that should be avoided in a scientific paper. Although they add to the problem of climate change, anthropogenic methane emissions increase linked to economic growth and development.

Lines 282–283: There is not enough causal evidence provided by the authors to claim outsourcing of heavy-emitting industries.

Paragraph lines 378–298: This paragraph is not derived from the analysis and includes aspects from trade relations and geopolitics that are not well treated. Accordingly, the derivation of the policy recommendation is unclear or naive.

Therefore, some of the conclusions and policy implications derived by the authors require further justification and a more direct link to the results.

All in all, I consider the paper falls short for the standard of *Nature Communications* and should be rejected.